# Single-cell transcriptomics reveal cellular diversity of aortic valve and the immuno-modulation by PPARγ during hyperlipidemia

Seung Hyun Lee [1,19], Nayoung Kim [2,3,19], Minkyu Kim [1,19], Sang-Ho Woo[4], Inhee Han[1], Jisu Park[1], Kyeongdae Kim[1], Kyu Seong Park[1], Kibyeong Kim[1], Dahee Shim[1], Sang-eun Park[1], Jing Yu Zhang[1], Du-Min Go[4], Dae-Yong Kim[4], Won Kee Yoon[5], Seung-Pyo Lee [6], Jongsuk Chung [7], Ki-Wook Kim [8], Jung Hwan Park[9], Seung Hyun Lee[10], Sak Lee[10], Soo-jin Ann[11], Sang-Hak Lee[12], Hyo-Suk Ahn[13,14], Seong Cheol Jeong[15], Tae Kyeong Kim[16], Goo Taeg Oh [16], Woong-Yang Park [7,17,18], Hae-Ock Lee [2,3,20] ✉ & Jae-Hoon Choi [1,20] ✉

Valvular inflammation triggered by hyperlipidemia has been considered as an important initial process of aortic valve disease; however, cellular and molecular evidence remains unclear. Here, we assess the relationship between plasma lipids and valvular inflammation, and identify association of low-density lipoprotein with increased valvular lipid and macrophage accumulation. Single-cell RNA sequencing analysis reveals the cellular heterogeneity of leukocytes, valvular interstitial cells, and valvular endothelial cells, and their phenotypic changes during hyperlipidemia leading to recruitment of monocyte-derived MHC-II[hi] macrophages. Interestingly, we find activated PPARγ pathway in $Cd36^+$ valvular endothelial cells increased in hyperlipidemic mice, and the conservation of PPARγ activation in non-calcified human aortic valves. While the PPARγ inhibition promotes inflammation, PPARγ activation using pioglitazone reduces valvular inflammation in hyperlipidemic mice. These results show that low-density lipoprotein is the main lipoprotein accumulated in the aortic valve during hyperlipidemia, leading to early-stage aortic valve disease, and PPARγ activation protects the aortic valve against inflammation.

Aortic valve disease emerges as a worldwide health problem as the population ages[1,2]. Unlike healthy, diseased aortic valve has a problem with its function of preventing blood from backflow. One of the most prevalent degenerative aortic valve diseases, calcific aortic valve disease (CAVD) is characterized by severe calcification of aortic valve leaflet, which gradually narrows the aortic orifice. This calcified lesion also causes a decrease in leaflet motion and obstruction of the left ventricular outflow tract, leading to aortic stenosis (AS)[3,4]. After the onset of AS symptoms, the mortality rate among those aged around 80 years is ~50% in 2 years and 80% in 5 years without surgical or transcatheter valve replacement[5,6]. However, there is no non-invasive therapeutic modality yet to prevent or suppress the initiation and progression of disease[4,7]. Thus, it is crucial to understand the molecular and cellular events in the early stage of aortic valve diseases to develop preventive therapeutics.

Inflammation is closely associated with the onset and progression of aortic valve disease. Various predisposing conditions including bicuspid valve, old age, male sex, hypertension, dyslipidemia, smoking, and obesity can induce the recruitment of leukocytes into aortic valve. The recruited leukocytes release pro-inflammatory cytokines

like TNF, IL1β, and IL6, which provoke a phenotypic change of valvular interstitial cells (VICs) into profibrotic and calcific phenotypes, leading to CAVD[8]. Similar to atherosclerosis, hyperlipidemia is an important risk factor for aortic valve inflammation and is known to drive sclerotic changes in the aortic valve. Familial hypercholesterolemia due to the inheritance of defective alleles of genes related to cholesterol homeostasis, such as low-density lipoprotein (LDL) receptor (LDLR), is associated with the prevalence of CAVD[9]. Recently, proprotein convertase subtilisin/kexin type 9 (PCSK9), which increases blood LDL level by binding and preventing LDLR recycling, is also reported as related to the CAVD prevalence[10], and PCSK9 loss-of-function mutation is reported to decrease the incidence of AS[11,12]. Although previous clinical trials on the effect of lipid-lowering treatments on progressed AS have been unsuccessful[13–15], a recent large-scale Mendelian randomization study demonstrated the causal association between high levels of LDL and AS, suggesting LDL-lowering treatments may be effective in mitigating aortic valve disease at an early time point[16]. However, the detailed mechanism of how leukocyte recruitment is initiated in the pre-calcific stage of aortic valve disease has remained elusive.

In the normal state, aortic valves contain various cell types, including VICs, valvular endothelial cells (VECs), and leukocytes including macrophages and dendritic cells (DCs)[17–19]. In the hyperlipidemic state, it is known that macrophages and T cells are accumulated in the lipid deposited region. But, the comprehensive immune cell profiling analysis is still lacking. In order to understand the initial disease mechanism, it is important to define the valvular cell components and dissect their phenotypic changes and cellular networks in hyperlipidemic mice. However, since the tissue is very small and composed of a few cells, it is challenging to analyze molecular and cellular events in murine aortic valves.

In this work, we first confirm the causal relationship between plasma lipids and valvular inflammation in *Ldlr*−/− and *Apoe*−/− mice having different plasma lipid profiles. We perform single-cell RNA sequencing (scRNA-seq) analysis of aortic valves from these mice. Our scRNA-seq analysis reveals the profound cellular heterogeneity of VICs, VECs, and leukocytes, and the increased pro-inflammatory gene expression in VECs and VICs during hyperlipidemia inducing the recruitment of monocyte-derived macrophages. Importantly, we find that PPARγ pathway is activated in aortic valves from *Ldlr*−/− and *Apoe*−/− mice and in human aortic valves of pre-calcified state but not in calcified valves. When *PPARG* is knockdown, VECs upregulate pro-inflammatory genes and cell adhesion molecules. While PPARγ inhibition promotes valvular inflammation, the activation of PPARγ protects the aortic valve from excessive inflammation during hyperlipidemia. Thus, we propose PPARγ as a key anti-inflammatory regulator in the pre-calcified stage of aortic valve disease.

## Results

### *Ldlr*−/− mice have more prominent valvular lipid accumulation than do *Apoe*−/− mice

To identify lesion formation in the aortic valve disease in hyperlipidemic conditions, we assessed lipid accumulation in the aortic valve, using the following two mouse models of hypercholesterolemia: *Apoe*−/− and *Ldlr*−/− mice (experimental group) and C57BL/6J mice (wild-type control group), fed either a chow or western diet (WD) for 10 weeks. We quantified valvular lipids and atherosclerotic lesions in the aortic sinus by staining frozen tissue sections with Oil Red O staining. Whereas C57BL/6J mice showed no significant difference in the valvular lipid content, *Apoe*−/− and *Ldlr*−/− mice that were fed with a WD had a higher valvular lipid deposition than did *Apoe*−/− and *Ldlr*−/− mice fed with a chow diet; however, *Ldlr*−/− mice showed a much higher increase in aortic valvular lipid accumulation than did *Apoe*−/− mice (Fig. 1a). In addition, whole-mount Oil Red O staining showed that lipid droplets mainly accumulated in the corpus arantii (C.A.)

region of the aortic valve in hyperlipidemic mice (*Apoe*−/− and *Ldlr*−/− mice fed a WD), but this effect was not observed in the valve of normal mice (C57BL/6J mice fed a chow diet) (Supplementary Figs. 1 and 2a). We further compared serum levels of total cholesterol and LDL in WD-fed and chow diet-fed mice. LDL, but not total cholesterol, was higher in *Ldlr*−/− mice than in *Apoe*−/− mice (Fig. 1b). In addition, in WD-fed hyperlipidemic mice (*Apoe*−/− and *Ldlr*−/− mice), LDL levels positively correlated with aortic valvular lipid accumulation, whereas total cholesterol levels and atherosclerotic lesions in the aortic sinus did not correlate with aortic valvular lipid accumulation (Fig. 1c, d). Further examination into lipid accumulation and lipid profiles of WD-fed *Apoe*−/− and *Ldlr*−/− mice, at the 4-, 8-, or 12-weeks[20], revealed similar patterns: lipid accumulation was greater in *Ldlr*−/− than in *Apoe*−/− mice at every time point, and a positive correlation between valvular lipids and LDL was observed (Supplementary Fig. 2b–d).

To identify whether the difference in valvular lipid accumulation between *Apoe*−/− and *Ldlr*−/− mice originated from differences in scavenger receptor expression levels, we examined the average expression levels of scavenger receptors and the level of each scavenger receptor (*Msr1*, *Scarb1*, *Orl1*, and *Cd36*) associated with LDL, via transcriptomic analysis of the whole aortic valve tissue, using five different experimental groups (C57BL/6J chow mice, *Apoe*−/− chow and WD mice, and *Ldlr*−/− chow and WD mice). We observed no significant differences among the groups (Supplementary Fig. 3).

### LDL is the main lipoprotein accumulated in the aortic valve

Next, we examined the uptake of VLDL and LDL in the ex vivo aortic valve culture system and VIC cultured in vitro using DiI-labeled lipoproteins. In the representative images and quantitative data of ex vivo aortic valve culture, uptake of LDL in the aortic valve was significantly higher than VLDL uptake in the three main cell types of the aortic valve, that is, macrophages, VECs, and VICs (Fig. 1e, f and Supplementary Movies 1–3). In addition, the accumulation of LDL in cultured VICs was higher than that of VLDL, and the deficiency of *Ldlr* or *Apoe* did not affect lipoprotein accumulation in VICs (Fig. 1g). The intravenously injected DiI-LDL accumulated more in the aortic valve than the injected DiI-VLDL (Fig. 1h). Overall, these results indicate that valvular accumulation of LDL occurs independently of the in situ protein expression of APOE or LDLR in the aortic valve.

Because LDL is the major accumulating lipoprotein in the aortic valve, we investigated the expression of SR-BI and CD36, known as receptors of LDL[21–23], in the aortic valve of C57BL/6J mice, to identify the mechanism of LDL accumulation. The expression of SR-BI and CD36 was mainly localized in the C.A. region of the valve (Fig. 1i). In the ex vivo LDL uptake assay, the treatment with salvianolic acid B (SAB, CD36 inhibitor) and block lipid transport-1 (BLT-1, SR-BI inhibitor) significantly diminished the uptake of LDL in VECs and VICs of C57BL/6J mice, whereas treatment with SAB or BLT-1 did not affect the macrophages (Fig. 1j).

### *Ldlr*−/− mice have a higher percentage of valvular leukocytes and associated subsets than do *Apoe*−/− mice, in hyperlipidemic conditions

To understand the inflammatory process in the aortic valve under conditions of hyperlipidemia, we performed a flow cytometric analysis of aortic valvular cells, using C57BL/6J, *Apoe*−/−, and *Ldlr*−/− mice fed either a chow or WD (Fig. 2a, b and Supplementary Fig. 4). *Apoe*−/− and *Ldlr*−/− mice fed with a WD showed a proportional increase in leukocytes and a decrease in VICs compared to *Apoe*−/− and *Ldlr*−/− mice fed with a chow diet, whereas C57BL/6J mice showed no difference between WD-fed and chow-fed mice (Fig. 2a, b and Supplementary Fig 5a). Similar to trends in lipid accumulation, *Ldlr*−/− mice showed a more prominent increase in total leukocytes than did *Apoe*−/− mice. Among leukocyte subsets, macrophages were significantly increased in both *Apoe*−/− and *Ldlr*−/− mice (Fig. 2a, b). Macrophages were classified into two different subsets: MHC-II^hiCD11c+CD206- and MHC-

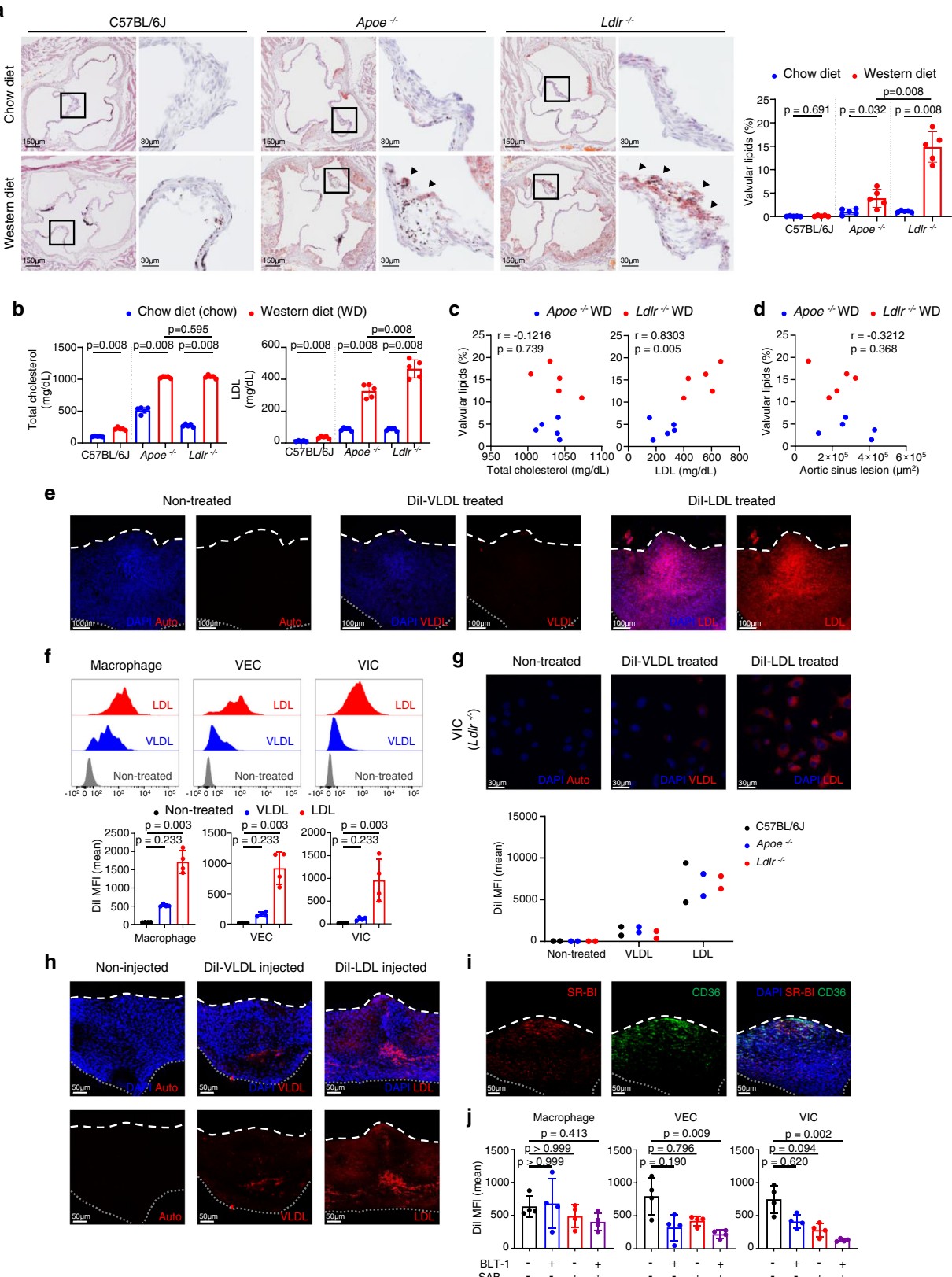

II^lo CD11c^-CD206^+ macrophages (Supplementary Fig. 4). Between the two subsets, only MHC-II^hi CD11c^+ CD206^- macrophages were significantly increased in WD-fed *Apoe^-/-* and *Ldlr^-/-* mice (Fig. 2b). Moreover, among the leukocyte subsets, MHC-II^hi CD11c^+ CD206^- macrophage percentage positively correlated with LDL levels, but not with total cholesterol levels (Fig. 2c and Supplementary Fig. 5b, c).

### High blood LDL level induced by hepatic PCSK9 overexpression increases the accumulation of lipids and macrophages in the aortic valve

To confirm the importance of blood LDL levels in the progression of aortic valve lesion formation, we induced the overexpression of gain-of-function PCSK9 mutant using adeno-associated virus (AAV)

**Fig. 1 | LDL is a primary lipoprotein accumulated in the aortic valve under hyperlipidemia. a–d** Comparison of lipid accumulation in the aortic valves and blood lipid profiles of mice (C57BL/6J, $Apoe^{-/-}$, and $Ldlr^{-/-}$) fed a chow diet (chow) versus a western diet (WD) for 10 weeks ($n = 5$). Representative lipid stain images and measurement of valvular lipid deposition. Black or dark brown spots are melanin pigments. Arrowhead: accumulated lipids. Scale bar: 150 µm (left), 30 µm (right) (**a**), total cholesterol and LDL levels in the blood plasma (**b**), correlations between aortic valvular lipid deposition and blood lipid profiles of WD-fed hyperlipidemic mice (**c**), and correlation between aortic valvular lipid accumulation and aortic sinus lesions in WD-fed hyperlipidemic mice (**d**). **e, f** DiI-lipoproteins (LDL or VLDL) uptake of aortic valves cultured ex vivo. Representative whole-mount images (repeated three times) (**e**) and flow cytometry analysis ($n = 4$) (**f**). Scale bar: 100 µm. **g** DiI-lipoprotein (LDL or VLDL) uptake levels of cultured VICs using C57BL/6J, $Ldlr^{-/-}$, and $Apoe^{-/-}$ mice. Representative

images (top) and mean fluorescence intensity (MFI) of DiI-lipoprotein detected by flow cytometry ($n = 2$. Each sample represents pooling of 10 mice). Scale bar: 30 µm. **h** Whole-mount images of the aortic valve from after intravenous injection of DiI-lipoproteins (LDL or VLDL) to C57BL/6J. Scale bar: 50 µm. **i** Whole-mount immunostaining of the aortic valve from C57BL/6J with SR-BI (red) and CD36 (green). Scale bar: 50 µm. **j** Flow cytometric analysis of DiI-lipoproteins (LDL or VLDL) uptake of aortic valves from C57BL/6 J mice, cultured with/without BLT-1 (SR-BI inhibitor) or SAB (CD36 inhibitor) ($n = 4$). Dashed line (white), outline of the free edge. Dotted line (gray), outline of the annulus-attached region. WD: western diet. Image data are representative of three independent experiments unless otherwise stated. Two-sided Mann–Whitney test (comparison of two groups) and Kruskal–Wallis test with post-hoc Dunn's test (comparison of three or more groups), were used for group comparisons. The Spearman correlation test was used for correlation analyses. Data are presented as mean ± SD.

serotype 8 (PCSK9-AAV). Injection of PCSK9-AAV induces a great elevation of blood LDL levels in mice[24]. Using Oil Red O staining, we found that PCSK9-AAV-injected $Apoe^{-/-}$ mice (24 weeks of WD) had more severe lipid accumulation, about 5.7-fold higher percentage, than control $Apoe^{-/-}$ mice. In addition, the overexpression of PCSK9 increased valve thickness in $Apoe^{-/-}$ mice and elevated plasma LDL levels (Fig. 2d–f). Furthermore, we checked the correlation between valvular lipid accumulation and each blood lipid profile, and only LDL presented a positive correlation with valvular lipid accumulation (Fig. 2g).

To verify the relationship between LDL levels and valvular inflammation, we performed immunostaining for CD68 and vimentin with lipid staining using BODIPY 493/503. Macrophages were localized in the lipid-accumulated C.A. region of the aortic valve and appeared as foamy cells containing many lipid droplets. Adjacent to the macrophages, vimentin⁺CD68⁻ VICs also showed a foamy phenotype (Fig. 2h). Next, we quantified the number of macrophages per aortic valve and found that macrophages were increased in PCSK9-AAV-infected mice, suggesting that the elevation of serum LDL levels worsened the aortic valvular inflammation (Fig. 2i). Altogether, these findings indicate that an increase in LDL enhances lipid accumulation and inflammation of the aortic valve, leading to the aortic valve sclerosis progression.

## Ezetimibe-induced lipid reduction ameliorates lipid accumulation and inflammation in the aortic valve

We assessed whether lipid-lowering treatment would alleviate lesion formations, caused by hyperlipidemia, during the early stages of aortic valve disease. To achieve this, we utilized $Ldlr^{-/-}$ mice, which were suitable to identify the change of valvular lesion, because they showed high levels of valvular lipid accumulation and inflammation when fed WD (Figs. 1a and 2a, b). $Ldlr^{-/-}$ mice were fed with WD containing a lipid-lowering drug (ezetimibe) and compared valvular lipid accumulation and immune cell proportions to control WD-fed $Ldlr^{-/-}$ mice (Fig. 2j–m). Ezetimibe clearly decreased total cholesterol and LDL levels in the blood of $Ldlr^{-/-}$ mice fed with an ezetimibe-based WD, and also decreased valvular lipid accumulation as well (Fig. 2j, k). Similarly, ezetimibe treatment significantly reduced the percentage of immune cells (leukocytes, macrophages, and MHC-II^hi CD11c⁺CD206⁻ macrophages) in aortic valves (Fig. 2l, m and Supplementary Fig. 5d). Collectively, we identified that lipid-lowering treatment was effective in reducing lipid accumulation and inflammation in hyperlipidemia-induced aortic valve disease.

## scRNA-seq analysis reveals cellular heterogeneity in aortic valves from normal and hyperlipidemic mice

To explore hyperlipidemia-associated cellular dynamics in aortic valves, we sorted total live single cells from the aortic valves of C57BL/6J (wild type), $Ldlr^{-/-}$, and $Apoe^{-/-}$ mice and performed single-cell transcriptome analysis using a droplet-based 10x Genomics

Chromium system (Supplementary Figs. 1b and 6). After quality filtering, we classified 6574 cells into 12 cell clusters using unsupervised cell clustering projected in a two-dimensional space through uniform manifold approximation and projection (UMAP) (Fig. 3a). Each cell cluster was assigned into three distinct cell lineages based on established canonical marker gene expression: VICs with stromal cells, VECs, and leukocytes (macrophages, DCs, T cells, and B cells) (Fig. 3b and Supplementary Data 1). All cell lineages were present in both normal and hyperlipidemic aortic valves, showing differences in cell distribution (Fig. 3c). VICs were the main cell population in aortic valves of C57BL/6J, and macrophages were the most abundant leukocytes in the aortic valves. Importantly the leukocytes, such as macrophages, DCs, and T cells were markedly increased in $Ldlr^{-/-}$ mice compared with C57BL/6J and $Apoe^{-/-}$ mice. Overall, the cellular proportion of scRNA-seq analysis sufficiently recapitulated the flow cytometric analysis (Fig. 2a–c and Supplementary Fig. 5a–c). The results indicated a tight association between macrophages and the pathogenesis of aortic valve disease. The extensive single-cell profiling enabled us to investigate cellular and molecular changes associated with hyperlipidemia in the aortic valve.

## The aortic valve contains two major macrophage populations, and monocyte-derived MHC-II^hi macrophages accumulate during hyperlipidemia

Based on the expression levels of $H2$-$Ab1$ (encoding a subunit of MHC-II), $Itgax$ (encoding CD11c), and $Mrc1$ (encoding CD206), valvular macrophages were subdivided into the following two main populations: $H2$-$Ab1^{hi}Itgax^{hi}$ and $Mrc1^{hi}$ macrophages (Fig. 3a, d), similar to the results of flow cytometric analyses (Fig. 2a, b and Supplementary Fig. 4). Using whole-mount immunofluorescence with MHC-II and CD206, we found that MHC-II^hi and CD206⁺ macrophages were spatially separated from each other. MHC-II^hi cells were mainly infiltrated into the fibrosa side of the C.A. region of the hyperlipidemic aortic valve, whereas CD206⁺ macrophages were lined in the ventricularis side of the area near the annulus (Fig. 3e, Supplementary Fig. 7 and Supplementary Movies 4–6).

To assess whether the MHC-II^hi CD11c⁺CD206⁻ macrophages are monocyte-derived, we induced hyperlipidemia by PCSK9-AAV infection and WD feeding to $Ccr2^{+/+}$ or $Ccr2^{-/-}$ mice. Although there was no difference in LDL levels of $Ccr2^{+/+}$ and $Ccr2^{-/-}$ mice, MHC-II^hi CD11c⁺ CD206⁻ macrophages were substantially decreased in $Ccr2^{-/-}$ mice, indicating that the valvular MHC-II^hi macrophages mainly originated from blood monocytes (Fig. 3f, g).

## MHC-II^hi valvular macrophages express pro-inflammatory genes whereas Lyve1⁺ macrophages show anti-inflammatory gene profiles

We re-classified leukocyte clusters to understand their dynamics and role in valvular inflammation. Sub-clustering of 3160 leukocytes identified them as macrophages (Macs), monocyte-derived dendritic cells

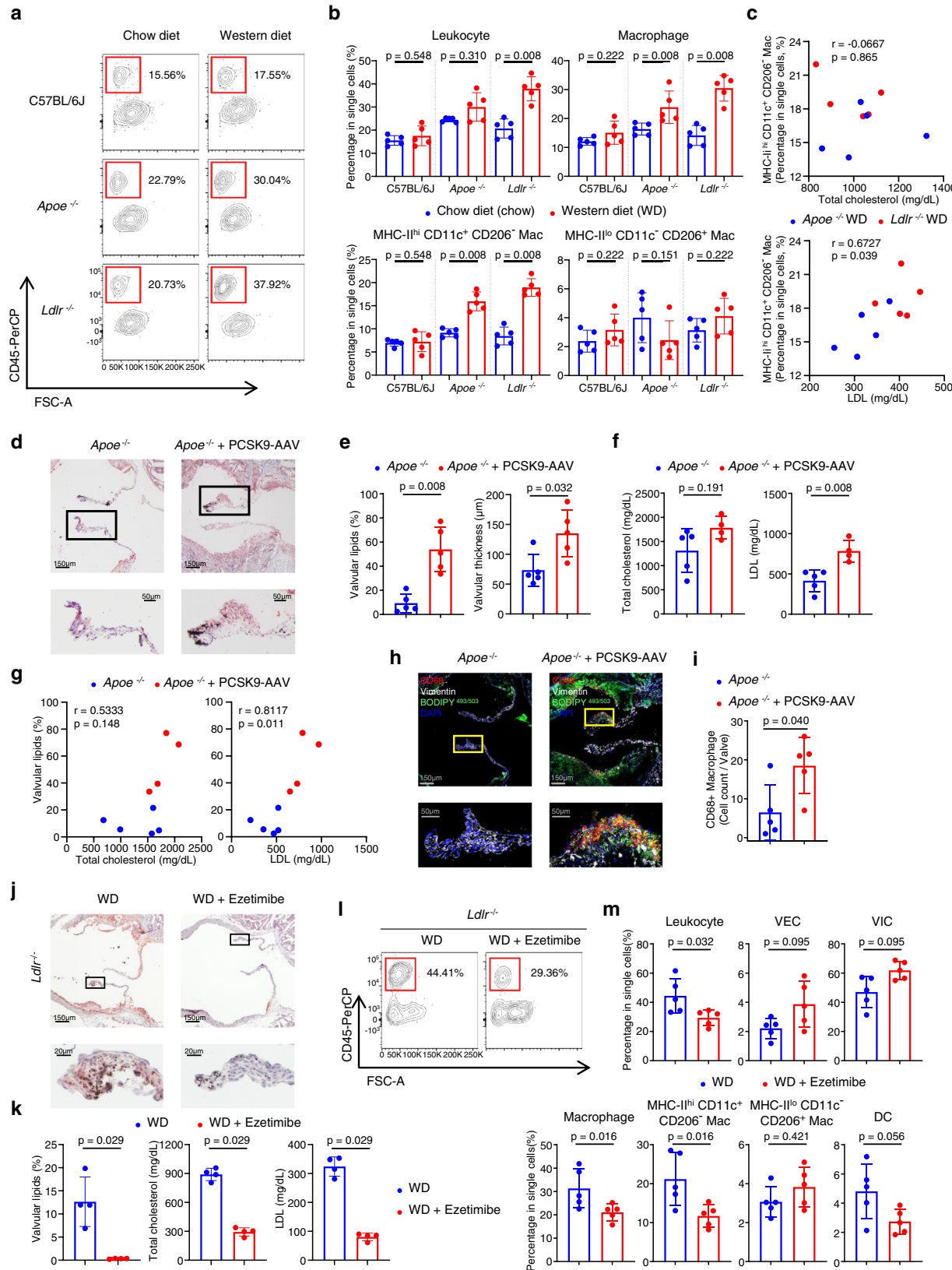

(moDCs), conventional type 1 dendritic cells (cDC1s), Cd8+ T cells, Cd4+ T cells, and B cells (Fig. 4a). A total of 11 clusters were assigned to 6 leukocyte lineages along with the expression of known canonical markers (Fig. 4b). The *Ldlr*[-/-] mice were enriched in leukocytes, with a 50% increase in cell numbers compared with those in *Apoe*[-/-] and C57BL/6J mice (Fig. 4c). The relative proportion of clusters within immune cells showed a similar distribution at this resolution. Notably, macrophages (LEU_C0, C1, C2, C4, C5, and C7) were the predominant leukocyte type in the aortic valve under all conditions. Each macrophage sub-cluster showed distinctive cellular pathways in functional categories (Supplementary Data 1), suggesting phenotypic and functional heterogeneity. LEU_C0 expressed *Ctss*, *Vcam1*, and genes related

**Fig. 2 | Plasma LDL levels positively correlate with inflammation and lipid accumulation in the aortic valve. a–c** Flow cytometry analyses of the aortic valves of C57BL/6 J, *Apoe*⁻/⁻, and *Ldlr*⁻/⁻ mice fed a chow diet (chow) or western diet (WD) for 10 weeks (*n* = 5). Representative plot of leukocytes (**a**), percentage of leukocytes, macrophages, and macrophage subsets (**b**), and correlations between the percentage of MHC-IIʰⁱ CD11c⁺ CD206⁻ macrophages and blood lipid profiles of WD-fed hyperlipidemic mice (**c**). **d–i** *Apoe*⁻/⁻ mice were injected with PCSK9-AAV to identify the effect of elevated serum LDL levels. PCSK9-AAV-injected *Apoe*⁻/⁻ mice were compared with non-injected *Apoe*⁻/⁻ mice (WD for 24 weeks). Representative Oil Red O stain images. Scale bar: 150 µm (top), 50 µm (bottom) (**d**), quantification of valvular lipid deposition and thickness of aortic valve (*n* = 5) (**e**), total cholesterol and LDL levels in blood plasma (*n* = 5 for *Apoe*⁻/⁻, *n* = 4 for *Apoe*⁻/⁻+PCSK9-AAV) (**f**), correlation between valvular lipid deposition and each lipid profile (total cholesterol and LDL) (*n* = 9 total; 5 from *Apoe*⁻/⁻; 4 from *Apoe*⁻/⁻+PCSK9-AAV) (**g**), immunostaining of CD68 (red) and Vimentin (white) along with the lipid stain using BODIPY 493/503 (green). Scale bar: 150 µm (top), 50 µm (bottom) (**h**), and quantification of CD68 + macrophages accumulated in the aortic valve (*n* = 5) (**i**). **j, k** Lipid-lowering effects of ezetimibe on lipid accumulation in the aortic valves of WD-fed (for 10 weeks) *Ldlr*⁻/⁻ mice (*n* = 4). Representative lipid stain images. Scale bar: 150 µm (top), 20µm (bottom) (**j**) and quantification of valvular lipid deposition with blood lipid profiles (**k**). **l, m** Effects of lipid-lowering by ezetimibe on the proportion of immune cells in aortic valves of WD-fed (for 10 weeks) *Ldlr*⁻/⁻ mice, through flow cytometry (*n* = 5). Representative plot of leukocytes (**l**) and percentages of each cell population in single cells (**m**). WD: western diet. For (**b**), (**e**), (**f**), (**i**), (**k**), and (**m**), two-sided Mann-Whitney test was used. For (**c**) and (**g**), the Spearman correlation test was used. Data are presented as mean ± SD.

to MHC-II (*Cd74*, *H2-Eb1*, and *H2-Aa*), and enriched with a lysosome pathway. LEU_C1 showed *Cd9*, *Actg1*, and *Cd14* as top genes, and upregulated with pathways (glycolysis, gluconeogenesis, and pentose phosphate pathway) related to glucose metabolism. LEU_C2 expressed marker genes of the M2-like macrophage such as *Folr2*, *Lyve1*, *Mrc1*, and *Cd163*, and enriched with the endocytosis, and the complement and coagulation cascade pathways. LEU_C4 was identified as the proliferating macrophage, showing a high level of genes (*Mki67*, *Ccna2*, and *Cdk1*) and pathways (DNA replication and spliceosome) associated with cell cycles. LEU_C5 contained the expression of genes related to the type I interferon (*Ifit1*, *Ifit2*, *Ifit3*, *Oasl1*, and *Irf7*) and MHC-I (*H2-D1*, *H2-T23*, and *H2-K1*), and involved with inflammatory pathways such as toll-like receptor signaling pathway and cell adhesion molecules. LEU_C7 expressed *Fabp5*, *Trem2*, *Plin2*, *Lpl*, and *Abca1*, which are previously reported as marker genes of foamy macrophage[20], and enriched with pathways (PPAR signaling pathway and glycerolipid metabolism) related to lipid handling.

As shown in Figs. 2 and 3e, we subcategorized valvular macrophages into two functional phenotypes: MHC-IIʰⁱCD11c⁺ macrophages and MHC-IIˡᵒCD206⁺ macrophages by flow cytometry and whole-mount immunofluorescence. In parallel, the expression pattern of *H2-Ab1*, *Itgax*, *Mrc1*, and *Lyve1* recapitulated two mutually exclusive populations (Fig. 4d). For a refined analysis, we constructed a transcriptional trajectory to infer changes in the functional status of macrophages (Fig. 4e). The individual cells of LEU_C2, LEU_C5, and LEU_C4 were located at the end of separate trajectory branches, while cells of LEU_C0, LEU_C1, and LEU_C7 spanned all branches. In accordance with previous findings[25], macrophages in the aortic valve demonstrated independent subpopulations similar to those in Lyve1⁻MHC-IIʰⁱ Mac (LEU_S1), Lyve1⁺MHC-IIˡᵒ Mac (LEU_S2), or proliferative Mac (LEU_S3) from the trajectory (Fig. 4f). Lyve1⁻MHC-IIʰⁱ Mac (LEU_S1) showed pro-inflammatory and Lyve1⁺MHC-IIˡᵒ Mac (LEU_S2) showed anti-inflammatory gene expression (Fig. 4f, g). Proportional cell counts demonstrated an increase in pro-inflammatory macrophages in *Apoe*⁻/⁻ and *Ldlr*⁻/⁻ mice but a decrease in anti-inflammatory macrophages (Fig. 4h).

## VICs contain heterogeneous cell populations with distinct gene expression profiles and increase pro-inflammatory gene expression during hyperlipidemic condition

Sub-clustering of non-immune, non-endothelial and non-*Clec3b*⁺ stromal cells revealed four clusters (Fig. 5a and Supplementary Fig. 8). Each cell cluster was constructed with an unbiased fraction of cells between normal and hyperlipidemic mice (Fig. 5b). Graph-based clustering showed heterogeneous subsets of VICs. The following four VIC types populated the aortic valve: *Meox1*⁺ (VIC_C0), *Id4*⁺ (VIC_C1), *Spp1*⁺ VICs (VIC_C2), and *Irf7*⁺ (VIC_C3); They presented with the expression of distinct marker genes and the activation of unique functional categories (Fig. 5c, d, Supplementary Data 1). We confirmed that VIC clusters were in the aortic valve leaflet—*Meox1*⁺ VICs on the inner side, *Id4*⁺ VICs on the outer side, and *Spp1*⁺ VICs on the root of aortic valve leaflet near the annulus (Fig. 5e). VICs from hyperlipidemic mice showed increased expression of genes related to myofibroblast activation and calcification than those of normal mice (C57BL/6 J) (Fig. 5f and Supplementary Data 2). The expression of genes associated with monocyte recruitment, such as *Csf1* and *Cx3cl1* were markedly increased in the VICs of hyperlipidemic mice compared to normal mice (Fig. 5g, h). Moreover, in the gene set enrichment analysis focused on inflammation-associated terms, VICs in hyperlipidemic mice showed definitive inflammatory characteristics (Fig. 5i). Thus, a subset of VICs in hyperlipidemic aortic valves induces molecular changes to accommodate inflammatory responses. Indeed, oxLDL-treated VICs showed enhanced monocyte migration than non-treated or LDL-treated VICs (Fig. 5j). In the monocyte adhesion assay using ex vivo cultured aortic valve, oxLDL-treated valves promoted a greater attachment of monocytes than non-treated valves (Fig. 5k).

## VECs contain three major populations and *Cd36*⁺ VECs are markedly increased in hyperlipidemic condition

Sub-clustering of the 536 VECs revealed six clusters (Fig. 6a). Fewer VECs than VICs and leukocytes were recovered from the aortic valve. Nonetheless, distinct *Cd36*⁺ ECs (VEC_C1) were identified in both the *Apoe*⁻/⁻ and *Ldlr*⁻/⁻ groups (Fig. 6b, c). Other cell clusters were assigned to vascular endothelial cell types, including *Fgfr3*⁺ ECs (VEC_C0), *Prox1*⁺ ECs (VEC_C2), *Pdpn*⁺ ECs (VEC_C3), and *Edn1*⁺ ECs (VEC_C5) (Fig. 6a–c, Supplementary Data 1). The clusters (VEC_C0, C1, and C2) populated more than 80% of the total VECs (Fig. 6c). In total, VECs in *Apoe*⁻/⁻ and *Ldlr*⁻/⁻ mice showed upregulation of pro-inflammatory genes and over-enrichment in functional gene sets related to monocyte chemotaxis when compared with the C57BL/6J group, suggesting that VECs are also related to valvular inflammation (Fig. 6d, e).

*Fgfr3*⁺ VECs (VEC_C0) and *Prox1*⁺ VECs (VEC_C2) were the main EC populations in the C57BL/6J mice. However, *Cd36*⁺ VECs (VEC_C1) were markedly increased in *Apoe*⁻/⁻ and *Ldlr*⁻/⁻ mice (Fig. 6a, c). In addition, combined analyses of VECs with public scRNA-seq for aortic ECs formed distinct clusters of *Cd36*⁺ ECs from normal ECs (Supplementary Fig. 9). Using RNA in situ hybridization and immunofluorescence, we validated three major VEC clusters. Among these, *Fgfr3*⁺ VECs (VEC_C0) and *Prox1*⁺ VECs (VEC_C2) exhibited distinct spatial patterns. *Fgfr3*⁺ VECs were regionally confined to the ventricularis side while *Prox1*⁺ VECs were only positioned on the fibrosa side (Fig. 6f). Unlike the two clusters, *Cd36*⁺ VECs (VEC_C1) did not have a specific spatial pattern. Consistent with the scRNA-seq data, more *Cd36*⁺ VECs were observed in the diseased valve (*Ldlr*⁻/⁻) than in the normal valve (C57BL/6J) (Fig. 6c, f). Previously, Kalluri et al. demonstrated that aortic ECs were classified into three subtypes—EC1, EC2, and EC3[26]. We found that the transcriptional characteristics of the three ECs were comparable to those of the VEC clusters (Fig. 6g). *Cd36*⁺ VECs expressed EC2-associated gene signatures, whereas *Fgfr3*⁺ was similar to EC1. *Pdpn*⁺ EC showed higher expression of the EC3-related genes. *Fgfr3*⁺ VECs showed significantly upregulated extracellular matrix organization and integrin signaling, consistent with EC1. We confirmed the enrichment of a *Cd36*⁺ VEC population with lipid transport and

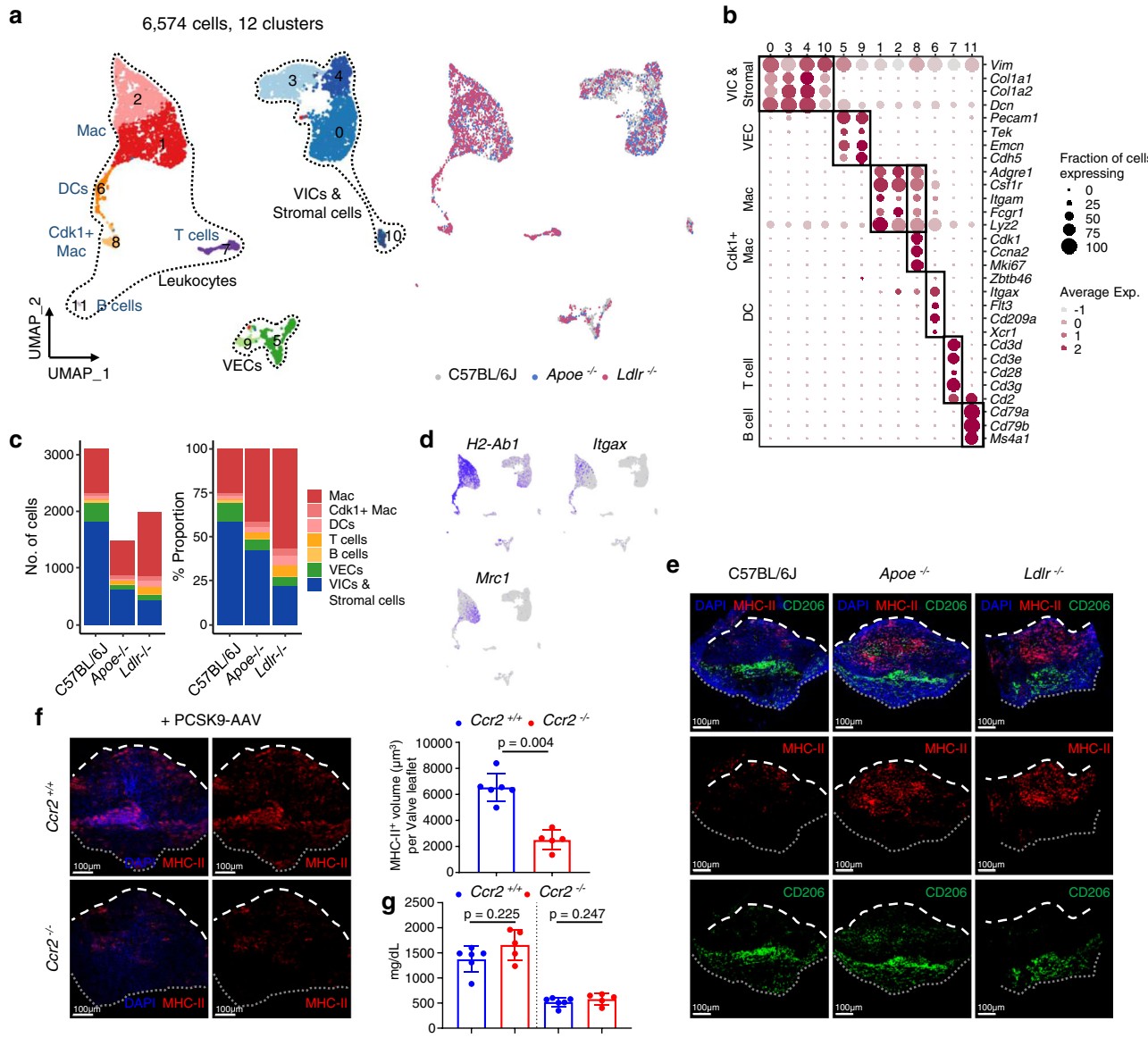

**Fig. 3 | Comprehensive single-cell profiling reveals that monocyte-derived macrophages are the major immune cells accumulated in aortic valve during hyperlipidemia. a** Uniform manifold approximation and projection (UMAP) plot of 6574 single cells colored by the clusters (left) and mouse models (right). *Apoe⁻/⁻* and *Ldlr⁻/⁻* mice were fed a WD for 16 weeks. **b** Average expression map of known cell-type marker genes for each cell cluster. Color represents average expression levels, which are scaled by z-transformation and limited to a scale from −2.5 to 2.5. Dot size represents the fraction of cells with the expression value of each marker gene for each cluster. **c** Absolute cell number (left) and relative proportion (right) of the major cell lineages from each mouse model. **d** UMAP plot of gene expression of *H2-Ab1*, *Itgax*, and *Mrc1* (gray to blue). **e** Whole-mount immunostaining of the aortic valve from C57BL/6J, *Apoe⁻/⁻* and *Ldlr⁻/⁻* mice for MHC-II (red) and CD206 (green). Scale bar: 100 μm. **f** Accumulated MHC-II⁺ cells in the aortic valve of PCSK9-AAV-injected *Ccr2⁺/⁺* or *-Ccr2⁻/⁻* mice (WD for 24 weeks). Representative whole-mount MHC-II immunofluorescence images (left) and measurement of MHC-II + volume in the aortic valve (right) (*n* = 6 for *Ccr2⁺/⁺*, *n* = 5 for *Ccr2⁻/⁻*). Scale bar: 100 μm. **g** Plasma total cholesterol and LDL levels in PCSK9-AAV-injected *Ccr2⁺/⁺* or *-Ccr2⁻/⁻* mice (WD for 24 weeks) (*n* = 6 for *Ccr2⁺/⁺*, *n* = 5 for *Ccr2⁻/⁻*). For (**f**) and (**g**), two-sided Mann−Whitney test was used. Dashed line (white), outline of the free edge. Dotted line (gray), outline of the annulus-attached region. Image data are representative of three independent experiments unless otherwise stated. Data are presented as mean ± SD.

angiogenesis, represented by the upregulation of genes in lipoprotein handling and the angiogenic tip cell gene set. Interestingly, EC2 type, *Cd36⁺* VECs (VEC_C1) exhibited enhanced gene expression associated with PPARγ signaling pathway including *Cd36* and *Fabp4* (Fig. 6b, g and Supplementary Data 1). Cluster-specific pathway analysis also showed enhanced PPAR signaling pathway in *Cd36⁺* VECs (Fig. 6h).

### PPARγ pathway is activated in *Cd36⁺* VECs

The upregulation of the lipoprotein handling pathway in *Cd36⁺* VECs was further supported by regulatory network inference, using Single-cell Regulatory Network Inference and Clustering (SCENIC) analysis[27]. The transcription factors activated in *Cd36⁺* VECs were related to lipid metabolisms such as *Nr1h3* (encoding LXRα) and *Pparg* (encoding PPARγ) (Supplementary Fig. 10). In particular, the PPARγ signaling pathway was significantly associated with *Cd36⁺* VECs in the gene expression level. We highlighted the activation score and expression level of PPARγ regulon (a gene module composed of the transcription factor and its direct target genes). As expected, the PPARγ regulon was highly activated in *Cd36⁺* VECs (Fig. 7a, b and Supplementary Fig. 11). Next, we classified aortic valvular cells into two classes (PPARγ high and low groups) according to the gene

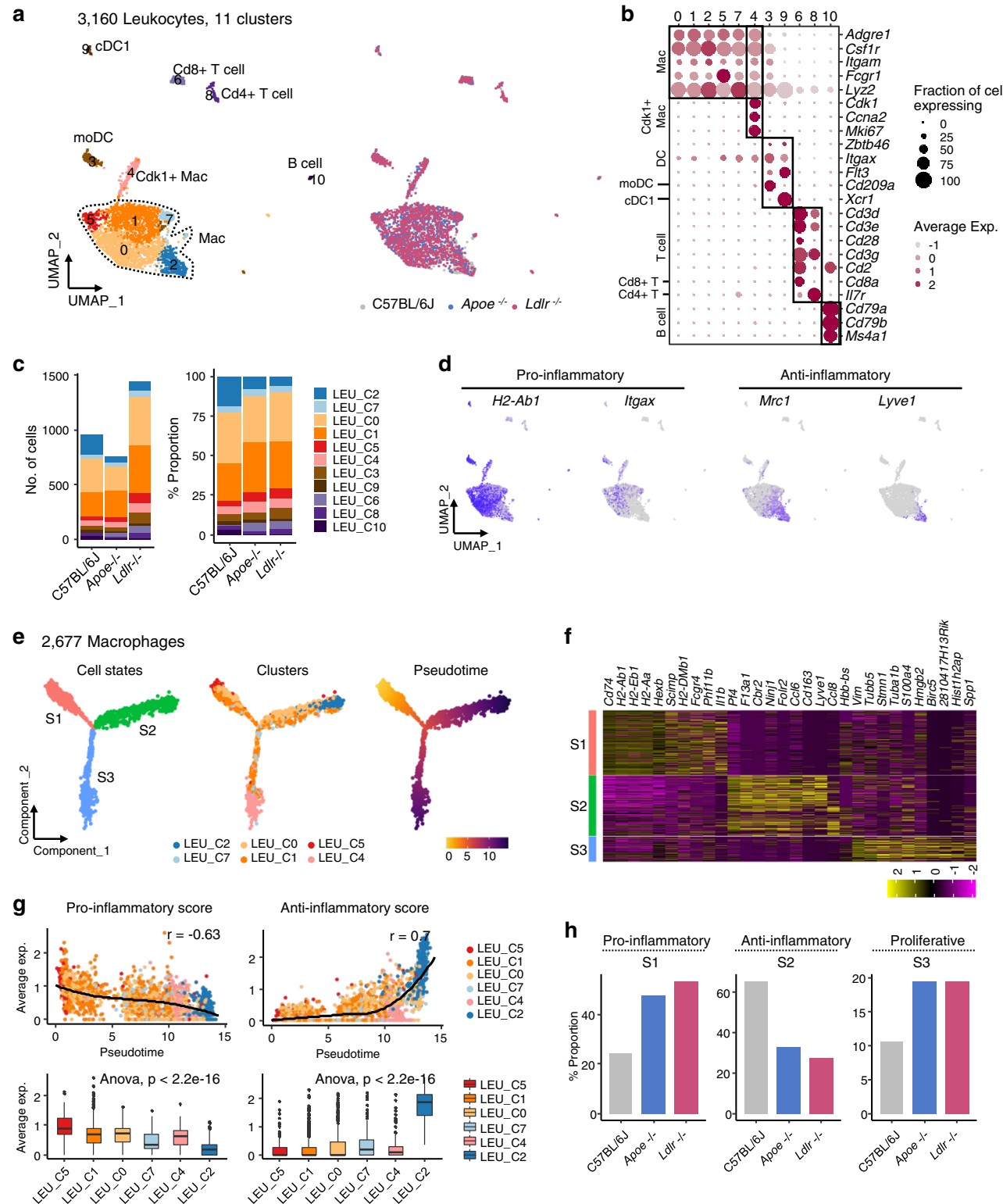

expression level of PPARγ regulon. Most of the valvular cells in PPARγ high group were VECs, accounting for a high proportion in *Cd36*+ VECs (VEC_C1) (Supplementary Fig. 12a). To identify genes associated with the expression level of PPARγ regulon, we selected differentially expressed genes specific for PPARγ high group compared to the low group in aortic valvular cells and VECs, respectively. Cells with high expression of PPARγ regulon showed significant upregulation of *Cd36*, *Fabp4*, and *Gpihbp1*, which belong to PPARγ dependent genes (Supplementary Fig. 12b and Supplementary Data 3). Interestingly,

they also represented upregulation of *Cxcl12*, which is known as a monocyte chemoattract and involved in the polarization of macro-phage to an anti-inflammatory state[28–31]. We also evaluated the average expression level of PPARγ target genes (49 genes, http://www.ppargene.org/)[32] in VECs, VICs, and macrophages. Only VECs in hyperlipidemic (*Apoe*−/− and *Ldlr*−/−) mice were increased when compared to control (C57BL/6J) mice (Supplementary Fig. 12c, d). Notably, even in C57BL/6J mice, the average expression level of PPARγ target genes was higher in VECs than in VICs and macrophages

**Fig. 4 | Pro-inflammatory valvular macrophages are markedly increased during hyperlipidemia. a** UMAP plot of 3160 leukocytes, colored by the clusters and mouse models as indicated. *Apoe*⁻/⁻ and *Ldlr*⁻/⁻ mice were fed a WD for 16 weeks. **b** Average expression map of known cell-type marker genes for each cell cluster. Color represents average expression levels which are scaled by z-transformation and limited to a scale from −2.5 to 2.5. Dot size represents the fraction of cells with the expression value of each marker gene for each cluster. **c** Absolute cell number (left) and relative proportion (right) of leukocyte subsets from each mouse model. **d** UMAP plot of expression level of marker genes (gray to blue). **e** Trajectory component plot of 2677 macrophages colored by the cell states, clusters, and mouse models. **f** Expression map of top10 significant genes for each cell state. Color represents the expression levels, which are scaled by z-transformation and limited

to a minimum scale of −2.5. (purple to yellow). **g** Correlation of Monocle pseudo-time with functional features of macrophages. Pro-inflammatory score represents mean expression of featured genes: *Il1b*, *Tnf*, *Ccl2*, *Cxcl10*, *Cxcl2*, *H2-Ab1*, and *Itgax*. Anti-inflammatory score represents mean expression of featured genes: *Mrc1*, *Lyve1*, *Folr2*, *Cbr2*, and *Il10*. Trend line and the top-right text (r) denote LOESS fit and Pearson's correlation, respectively (top). Boxplot of macrophage functional features in each cluster (bottom) ($n = 159$ cells for LEU_C5; 907 for LEU_C1; 973 for LEU_C0; 123 for LEU_C7; 182 for LEU_C4; 333 for LEU_C2). Each box depicts the interquartile range (IQR, the range between the 25th and 75th percentile) and median of each score, whiskers indicate 1.5 times the IQR. One-way ANOVA test p-value. **h** Cell proportion of functional cell states in macrophages for each mouse model.

(Supplementary Fig. 12c). To validate protein expression and nuclear localization in VECs, we performed immunostaining of PPARγ with an EC marker (endomucin, EMCN). The valves from hyperlipidemic mice (*Apoe*⁻/⁻ and *Ldlr*⁻/⁻) showed significantly higher PPARγ expression in nuclei of VECs, compared to normal valves (C57BL/6J) (Fig. 7c and Supplementary Fig. 13).

### PPARγ activity is conserved in non-calcified human aortic valves
To identify whether PPARγ-activated VECs are conserved in the human aortic valve, we re-analyzed a previous public scRNA-seq dataset of human aortic valves[33]. The single cell transcriptomes of two non-calcified and four calcified aortic valves were merged and visualized using UMAP (Fig. 7d). Notably, PPARγ-dependent genes were enriched in VECs (hAV_C4) and showed overexpression in both non-calcified and calcified aortic valves (Fig. 7e). There was no significant difference in *CD36* expression of VECs between non-calcified and calcified aortic valves, and the overall *CD36* expression was only barely detectable levels (Supplementary Fig. 14). Next, to elucidate the VEC-specific function of PPARγ, we isolated VECs from human aortic valves (Supplementary Fig. 15a, b), transfected siRNA to *PPARG* knockdown in human aortic VECs (Supplementary Fig. 15c), and performed RNA-seq analyses comparing *PPARG* knockdowns versus negative controls—with or without the stimulation of oxLDL (Fig. 7f, g and Supplementary Data 4). We found that PPARG knockdown upregulated pro-inflammatory genes and cell adhesion molecules, such as *CXCL1*, *CCL2*, *CXCL16*, *IL6*, *ICAM1*, and *ICAM2* in VECs (Fig. 7f, g). Moreover, immunohistochemistry (IHC) of PPARγ on the human aortic valves confirmed that PPARγ-activated valvular cells, especially VECs were also conserved in the human aortic valves and that PPARγ proteins were more abundant in non-calcified samples than in calcified samples (Fig. 7h, i and Supplementary Table 1). Interestingly, the percentage of PPARγ⁺ VEC was positively correlated with plasma total cholesterol and LDL in non-calcified aortic valves (Fig. 7j and Supplementary Table 1). Collectively, these results suggest that the PPARγ pathway in VECs is activated by increased plasma LDL. The PPARγ pathway in VECs appears to possess anti-inflammatory properties.

### PPARγ activation protects the aortic valve against hyperlipidemia-induced inflammation
To identify the role of PPARγ in aortic valvular inflammation, we first performed a monocyte adhesion assay in the ex vivo aortic valve culture system. Treatment of PPARγ antagonist (T0070907) with oxLDL enhanced the attachment of monocytes on valves (Fig. 8a). We next administered the PPARγ antagonist (T0070907) or vehicle daily to *Ldlr*⁻/⁻ mice fed the WD[34,35]. The proportion of leukocytes was increased in the T0070907-treated group than in the vehicle-treated group. In particular, monocyte-derived MHC-II^hiCD11c⁺CD206⁻ macrophages and DCs were significantly increased in the T0070907-treated group than in the vehicle-treated group, while MHC-II^loCD11c⁻CD206⁺ macrophages and T cell showed no differences (Fig. 8b, c). Conversely, when we administrated the PPARγ agonist pioglitazone

to hyperlipidemic mice, the percentage of leukocyte in aortic valve was markedly decreased compared to the control group. While there was no proportional change of MHC-II^loCD11c⁻CD206⁺ macrophages, DCs and T cell, the monocyte-derived MHC-II^hiCD11c⁺CD206⁻ macrophages were significantly decreased by the administration of pioglitazone (Fig. 8d). The treatment of T0070907 or pioglitazone did not affect the level of total cholesterol and LDL (Supplementary Fig. 16a, b). Also, PPARγ activation by pioglitazone did not change the number of circulating monocytes and associated subsets in the blood (Supplementary Fig. 16c, d). Altogether, these results suggest that PPARγ activation during hyperlipidemia protects the aortic valve against excessive inflammation (Fig. 8e).

## Discussion
AS is a progressive disease with its early asymptomatic stage longer than its symptomatic stage. In the context of AS prevention and considering the findings of statin trials[13–15], it is important to elucidate the mechanism of the early stage of AS progression, especially the mechanism of disease initiation. Here, we provided strong evidence that LDL cholesterol is the predisposing factor in the initiation of valvular inflammation leading to AS, demonstrated by a dramatic decrease in valvular lipid accumulation and inflammation following lipid-lowering treatment. Our data are supported by the recent clinical study on the association of LDL with AS[16]. Thus we postulate that lipid-lowering therapy with statins or PCSK9 inhibitors may be useful in preventing the initiation of valvular inflammation.

Previously, two research groups performed scRNA-seq analysis on the aortic valve. One group focused on the development mechanism of cardiac valves[18], and the other group analyzed single-cell transcriptome of the human CAVD[33]. Meanwhile, we performed scRNA-seq analysis primarily focusing on the early inflammatory stage in hyperlipidemia. Our scRNA-seq revealed that monocyte-derived MHC-II^hiCD11c⁺CD206⁻ macrophage is the main leukocyte infiltrated in aortic valvular inflammation by hyperlipidemia. In previous research, there have been some reports of the accumulation of MHC-II^hi macrophage in the aortic valve with *Notch1* haploinsufficiency[36], or monocyte-derived macrophages that invoke inflammation in the mitral valve with Marfan syndrome[37]. It might be worthwhile to compare the phenotypic similarity/dissimilarity among those infiltrated macrophages with the identical surface marker but in different kinds of aortic valve diseases. Another main valvular macrophage population was MHC-II^loCD206⁺Lyve1⁺ macrophages, which showed an anti-inflammatory gene expression profile compared to MHC-II^hiCD11c⁺CD206⁻ macrophages. Some of the MHC-II^loCD206⁺Lyve1⁺ macrophages might have originated from endocardial-derived resident macrophages, which have recently been reported to play a role in extracellular matrix (ECM) regulation for tissue remodeling during aortic valve development[19]. Also, according to a previous study of resident macrophages in the aorta, CD206⁺LYVE1⁺ aortic macrophages regulate the ECM to maintain the arterial tone[38]. Therefore, MHC-II^loCD206⁺Lyve1⁺ resident macrophages in the aortic valve may have a similar function of ECM regulation for tissue homeostasis.

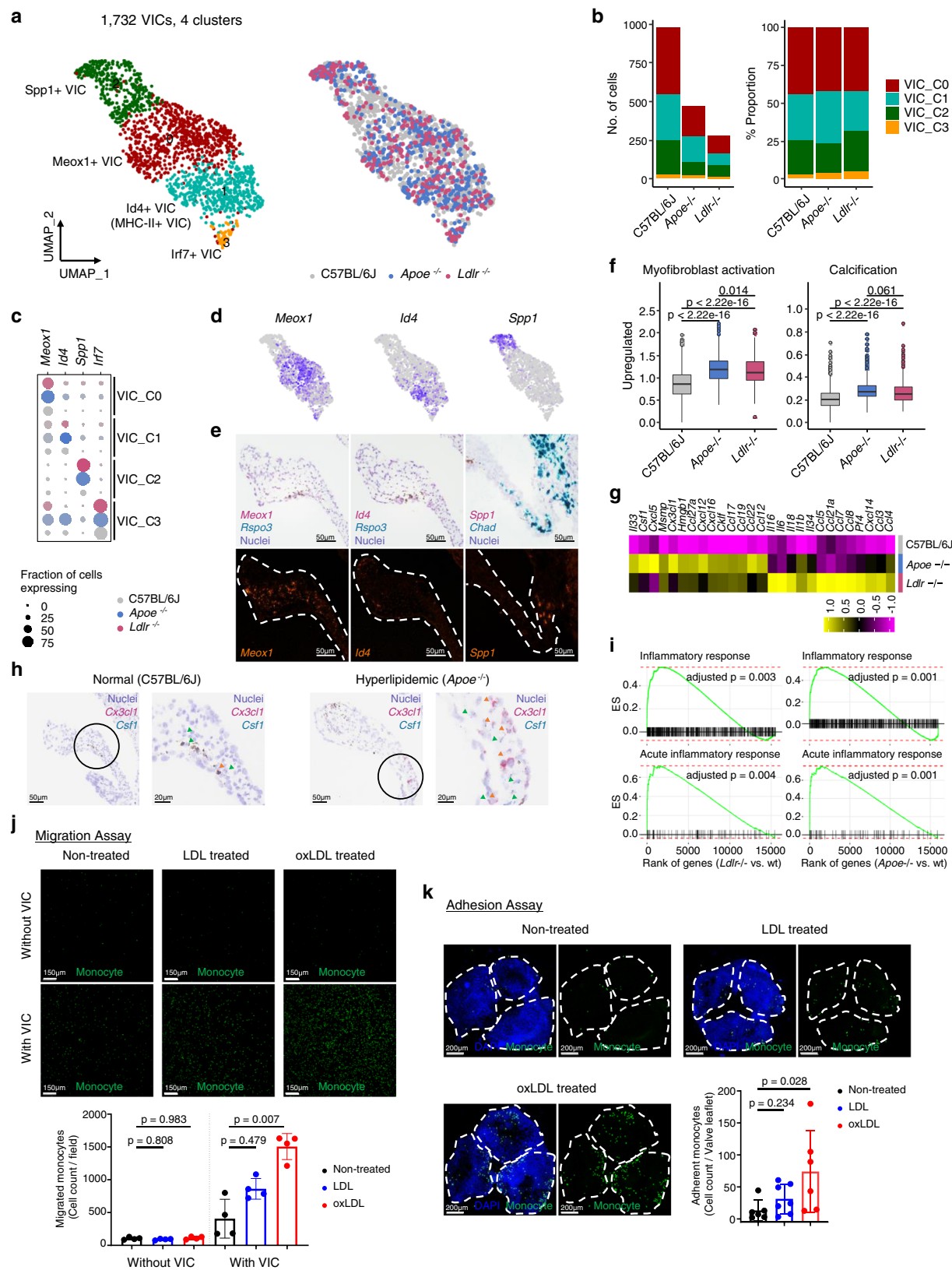

Notably, *Cd36*⁺ VEC was enriched with the PPARγ pathway gene expression, and its proportion was markedly increased in hyperlipidemic mice. *Cd36*⁺ VECs expressed high levels of *Cd36* (encoding CD36) and *Scarb1* (encoding SR-BI). These two scavenger receptors (CD36 and SR-BI) participate in LDL uptake; thus, VECs may transport LDL from the blood into the tissue via CD36- and SR-BI-related

mechanisms[21–23]. Although the distribution of CD36 or SR-BI is widespread (in other words, not confined to ECs only), direct exposure of ECs to blood functions to take circulating lipoproteins from the blood and provides them to peripheral tissues[39]. Therefore, the expression of CD36 and SR-BI in VECs might be crucial for lipid accumulation in the aortic valve during early-stage aortic valve disease. This observation is

**Fig. 5 | VICs show pro-inflammatory features during hyperlipidemia. a** UMAP plot of 1,732 VICs colored by clusters and mouse models as indicated. **b** Absolute cell numbers (left) and relative proportions (right) of the VIC subsets from each mouse model. **c** Average expression map of representative genes in each cell cluster. Color depends on the mouse model, and size represents the fraction of cells with the expression value of each gene for each group. **d, e** Representative gene expression of VIC subclusters. UMAP plot of gene expression (gray to blue) (**d**) and RNA in situ hybridization (**e**). Scale bar: 50 μm. **f** Boxplot of average expression level of genes related to myofibroblast activation (left) and calcification (right) (Gene list in Supplementary Data 2) ($n$ = 981 cells for C57BL/6J; 470 for $Apoe^{-/-}$; 281 for $Ldlr^{-/-}$). Each box depicts the IQR and median of each score, whiskers indicate 1.5 times the IQR. p, two-sided $T$-test p-value. **g, h** Expression map of monocyte chemoattractant genes. The average expression of genes for VICs (purple to yellow) in each mouse model was scaled by z-transformation and displayed on a scale of at least −2.5 (**g**) and RNA in situ hybridization of $Csf1$ and $Cx3cl1$ (**h**). Scale bar: 50 μm (left), 20 μm (right). Arrowheads indicate $Csf1$ (green) and $Cx3cl1$ (red) signal. **i** Enrichment plot of significant gene ontology (GO) terms related to inflammation. Genes were ranked by the fold change between knockout and wild-type models for VICs. **j** Transwell migration assay for evaluating monocyte chemotactic levels of VICs cultured with/without LDL or oxLDL. ($n$ = 4). Samples without VIC were used for control. Size of field: 1272.79 μm². Scale bar: 150 μm. **k** Adhesion assay of monocytes to ex vivo cultured aortic valves with/without LDL or oxLDL ($n$ = 6 for non-treated, $n$ = 7 for LDL, $n$ = 6 for oxLDL). Scale bar: 200 μm. Dashed line, outline of valve leaflets. Image data are representative of three independent experiments unless otherwise stated. For (**j**) and (**k**), Kruskal–Wallis test with post-hoc Dunn's test was used, and data are presented as mean ± SD.

supported by our data, which showed that CD36 and SR-BI blockages decreased LDL accumulation in the aortic valve. Meanwhile, a previous report showed that stenotic aortic valves expressed lower levels of *CD36* than did normal aortic valves[40]. We speculate that these differences may be attributed to distinct tissue microenvironments of the aortic valve in the early versus late stages of aortic valve disease.

Tissue-accumulated LDL can become oxLDL, which may trigger the inflammatory responses; however, oxLDL also acts as an endogenous ligand that activates PPARγ[41]. PPARγ is a transcription factor with pleiotropic functions. PPARγ is associated with glucose homeostasis, adipocyte differentiation, and lipid metabolism[42,43]. PPARγ activation upregulates the expression of liver X receptor α (LXRα), another important transcription factor involved in lipid metabolism[44]. Activation of PPARγ and LXRα induces the expression of various genes, such as *Cd36*, *Scarb1*, *Lpl*, and *Gpihbp1*, related to lipid metabolism. These genes facilitate lipid uptake, lipid efflux, and lipolysis[45,46]. In our scRNA-seq analysis, VECs that were high in PPARγ expressed notable levels of *Cd36*, *Scarb1*, *Lpl*, and *Gpihbp1*, implicating the lipoprotein-specific function of these genes in the hyperlipidemic state.

Furthermore, PPARγ-activated VECs are conserved in the human aortic valve, especially in the pre-calcified valvular state. In our RNA-seq analysis of human aortic VECs, knockdown of *PPARG* increased the expression of pro-inflammatory genes (*CXCL1*, *CCL2*, *CXCL2*, *IL6*, *CXCL8*, *CXCL16*, and *PTGS2*) and cell adhesion molecules (*ICAM1*, *ICAM2*, *VCAM1*, *PECAM1*, and *MCAM*), showing the anti-inflammatory role of PPARγ in a VEC-specific manner. Previously, it has been reported that PPARγ is expressed in the VECs of hyperlipidemic pigs, and pioglitazone attenuated valvular lipid deposition and calcification in hyperlipidemic mice[47,48]. It is also known that the activation of PPARγ induces anti-inflammatory functions in vascular ECs[49]. The mechanisms underlying the anti-inflammatory effects of PPARγ may be attributed to the negative regulation of NF-κB activation by PPARγ. Two molecular mechanisms of the negative regulation of NF-κB by PPARγ have been reported previously. First, PPARγ functions as an E3 ubiquitin ligase for p65 (also known as RELA, a subunit of NF-κB). Through the binding and ubiquitination of p65, PPARγ can induce p65 degradation via proteasomes[50]. Second, PPARγ can undergo sumoylation, where sumoylated PPARγ has an affinity for binding to the nuclear receptor coactivator 3 (NCOR3)-histone deacetylase 3 (HDAC3) complex. The NCOR3-HDAC3 complex interferes with NF-κB by binding to its target genes. The sumoylated PPARγ, bound to the NCOR3-HDAC3 complex, represses the ubiquitination and degradation of this complex, thereby eliciting transrepression of NF-κB[51].

In this study, we found that PPARγ inhibition aggravates valvular inflammation by infiltrated macrophages, and PPARγ agonist pioglitazone effectively mitigates aortic valvular inflammation. These results indicate that VECs exert anti-inflammatory feedback function via upregulation of the PPARγ pathway in the early stage of the disease. We suggest pioglitazone as the drug candidate for inhibiting aortic valvular inflammation via PPARγ activation. Since pioglitazone is currently prescribed for the treatment of diabetes mellitus, further

observational study of the effect of pioglitazone on aortic valve diseases such as CAVD, might be clinically meaningful.

In our results, ezetimibe reduced valvular lipid accumulation and inflammation via a lipid-lowering effect, and pioglitazone showed an anti-inflammatory effect on the aortic valve by PPARγ activation, without the change in blood total cholesterol and LDL levels. Previous studies showed the synergic beneficial effects of co-administration of lipid-lowering drugs and pioglitazone in patients with cardiovascular disease including atherosclerosis[52,53]. Therefore, it is needed to investigate the effect of co-treatment of lipid-lowering drugs and PPARγ agonists in the early aortic valve disease.

However, our study has several limitations. First, hyperlipidemic mouse models used in this study ($Apoe^{-/-}$ and $Ldlr^{-/-}$ mice) are less efficient to progress into aortic stenosis, although these models are sufficient to induce the early-stage lesion of aortic valve disease showing lipid accumulation and inflammation. It was reported that $Ldlr^{-/-}$ mice fed WD for 16 weeks only produced a scant amount of valvular calcification[54,55], and $Apoe^{-/-}$ mice fed WD for 20 weeks upregulated osteoblastic protein expression on the aortic valve but did not present microscopic valvular calcification[54,56]. In the case of the induction of severe calcification that may lead to aortic stenosis, utilizing a more suitable mouse model for valvular calcification such as $Notch1^{+/-}$ mice would be more appropriate[57,58]. Second, this study did not analyze the effect of lipoprotein(a) [Lp(a)], the LDL-like particle in humans. Assembly of Lp(a) needs apolipoprotein(a), coded by the gene named *LPA*, and this gene is missing in mice[59]. Lp(a) works as a carrier of oxidized phospholipid (OxPL) in human blood plasma[60], and it is reported that the Lp(a) and OxPL are associated with the incidence of AS by acceleration of aortic valvular inflammation and calcification[61–65]. Further investigation is required using Lp(a)-transgenic mice to dissect the role of Lp(a) in the early stage of aortic valve disease.

In summary, we demonstrated that lipid accumulation and inflammation of the aortic valve disease in the early stage is determined by LDL rather than total cholesterol, and identified the single-cell-based cellular characteristics among the normal and hyperlipidemic mouse models. Hyperlipidemic models showed a higher ratio of valvular leukocytes than the normal model, and leukocytes infiltrating the aortic valve were mainly monocyte-derived MHC-II^hi macrophages. These monocyte-derived MHC-II^hi macrophages were recruited by pro-inflammatory cytokines and chemokines, produced by VICs and VECs. Meanwhile, activated PPARγ exerted a protective role in the early inflammatory phase of aortic valve disease induced by hyperlipidemia, suggesting PPARγ as a putative drug target for aortic valve disease.

## Methods

### Human samples
The experimental protocols for the human study were reviewed and approved by the Institutional Review Board (IRB) of Yonsei Severance Hospital (Seoul, Korea, IRB No. 4-2018-0813), Seoul National University Hospital (Seoul, Korea, IRB No. 1104-122-360) and the Catholic

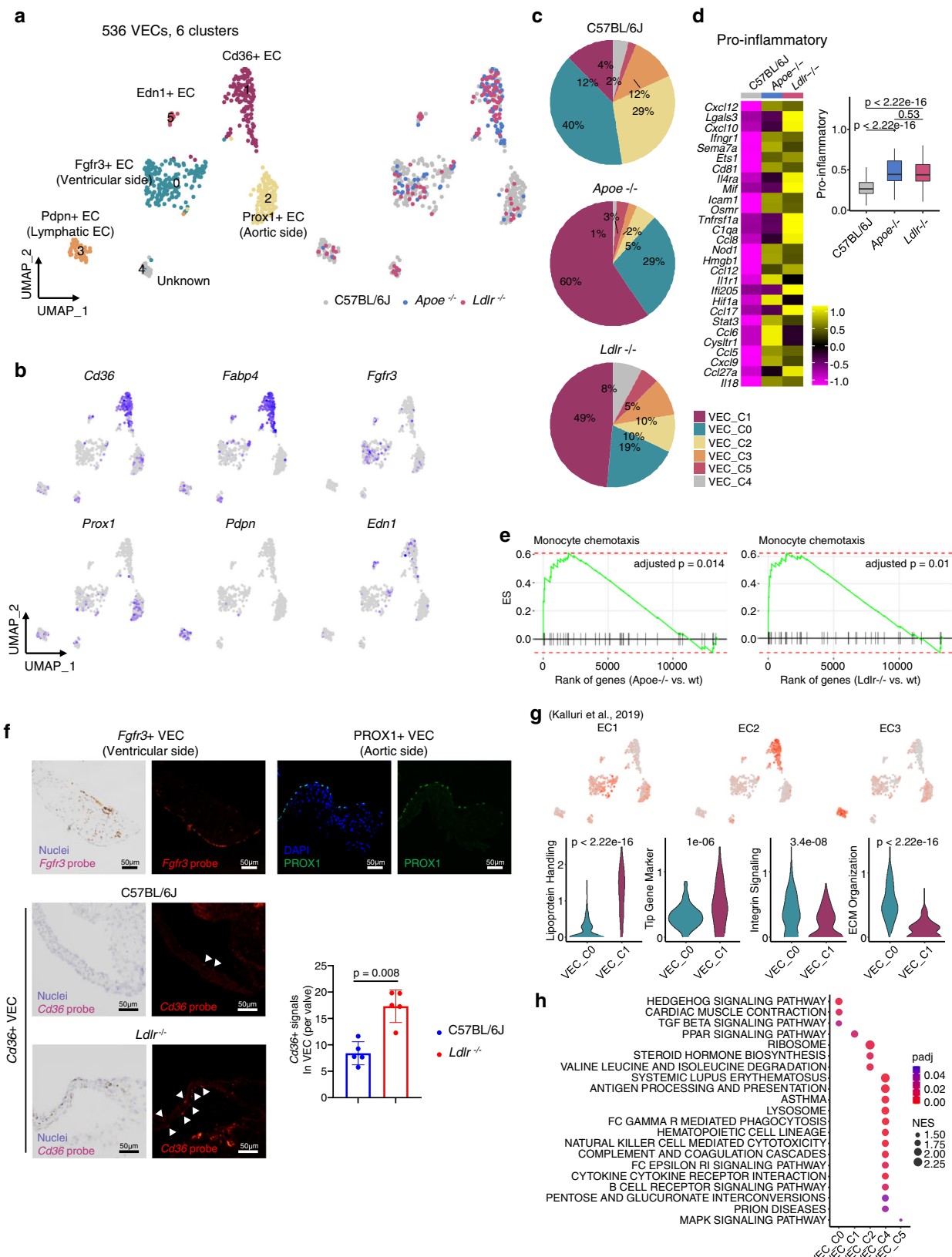

University of Korea, Uijeongbu St. Mary's Hospital (Uijeongbu, Korea, IRB No. UC19TIDE0142). All patients provided informed consent, including to the publication of information that identifies individuals. For IHC, human aortic valves ($n = 7$ for non-calcified and $n = 5$ for calcified group) were provided by Yonsei Severance Hospital (Seoul, Korea) and Seoul National University Hospital (Seoul, Korea). See

Supplementary Table 1 for the clinical information of patients. For human aortic VEC isolation, human aortic valves from two patients were provided by The Catholic University of Korea, Uijeongbu St. Mary's Hospital (Uijeongbu, Korea, IRB No. UC19TIDE0142). Patient information was as follows: patient #1 (age 73, male) was diagnosed with aortic aneurysm and underwent Bentall operation with aortic arch

**Fig. 6 | VECs contain three main subtypes and *Cd36*⁺ VECs are markedly increased in hyperlipidemic mice. a** UMAP plot of 536 VECs colored by the clusters and mouse models as indicated. **b** UMAP plot of VECs color-coded by expression (gray to blue) of marker genes. **c** Pie chart for the relative proportion of the VEC subsets from each mouse model. **d** Expression of pro-inflammatory genes in VECs (VEC_C0, C1, and C2). Heatmap (left) and boxplot (right) of the average expression level of genes listed in the heatmap. Heatmap are displayed as expression values scaled by z-transformation on a scale of at least-2.5. Cells having no expression for pro-inflammatory genes were excluded (*n* = 274 cells for C57BL/6J; 88 for *Apoe*⁻/⁻; 80 for *Ldlr*⁻/⁻). Each box depicts the IQR and median of each score, whiskers indicate 1.5 times the IQR. p, two-sided *T*-test *p*-value. **e** Enrichment plot for significant Gene Ontology (GO) terms related to monocyte chemotaxis. Genes were ranked by the fold changes between knockout and wild-type models for cells in VEC clusters (VEC_C0, C1, and C2). **f** Identification of the localization of VEC subclusters using RNA in situ hybridization (*Fgfr3* and *Cd36*) or immuno-fluorescence (PROX1). The graph indicates quantification of in situ hybridization using the *CD36* probe (*n* = 5). Two-sided Mann–Whitney test was used. Arrowhead: *Cd36*⁺ signals in aortic valve. Scale bar: 50 µm. **g** UMAP plot of VECs color-coded by average expression (gray to red) of genes specific to pre-defined EC subtypes (top). Expression score of genes in EC1 and EC2-associated pathways in VEC_C0 and VEC_C1 (bottom). p, two-sided *T*-test *p*-value. **h** Enrichment map of significant Kyoto Encyclopedia for Genes and Genomes (KEGG) gene sets for each cell cluster. Color represents the adjusted *p*-value (*p*adj) and size represents the normalized enrichment score (NES), calculated by *fgsea* R package. Image data are representative of three independent experiments unless otherwise stated. Data are presented as mean ± SD.

replacement, and patient #2 (age 76, male) was diagnosed with aortic aneurysm and aortic stenosis caused by a bicuspid aortic valve and underwent both aortic valve and ascending aorta replacement.

## Animal experiments
B6.129S7-*Ldlrtm1Her*/J (#002207, *Ldlr*⁻/⁻), B6.129P2-*Apoetm1Unc*/J (#002052, *Apoe*⁻/⁻), and B6.129S4-*Ccr2tm1Ifc*/J (#004999, *Ccr2*⁻/⁻) mice were obtained from The Jackson Laboratory. Male C57BL/6 J mice were purchased from SLC (Japan) or DBL (Korea). All mice were housed in the animal facility of Hanyang University under specific pathogen-free conditions in a 12-h light/12-h dark cycle with controlled temperature (20-24 °C) and humidity (40–60%), and supplied with a normal chow diet and water *ad libitum*. All animal procedures were approved by the Institutional Animal Care and Use Committee (IACUC) of Hanyang University (certification numbers: HY-IACUC-22-0027 and HY-IACUC-22-0029) and conformed to its regulations. To induce aortic valve lesions of lipid accumulation by hypercholesterolemia, male *Ldlr*⁻/⁻ and *Apoe*⁻/⁻ mice (8- to 10-week-old) fed a WD (Research Diets, #D12079B) for 10 or 16 weeks.

In some experiments, $1 \times 10^{11}$ vector genome copies of AAV8/D377Y-mPCSK9 (Sirion Biotech, Addgene plasmid: #58376) were intravenously injected to mice (8- to 10-week-old) to increase plasma LDL level[24,66]. *Apoe*⁻/⁻ mice, *Ccr2*⁻/⁻ or *Ccr2*⁺/⁺ (wild type) mice, and C57BL/6J (wild type) mice were used for the experiments, as shown in Figs. 2d–i, 3f, g and 8d, respectively. Immediately after the injection, the mice were switched from a chow diet to WD. A week after the injection, the serum lipid profile (total cholesterol, LDL, triglyceride, and HDL levels) was measured, and the WD diet was continued for 24 weeks (experiments using *Apoe*⁻/⁻ or *Ccr2*⁻/⁻ mice) or 6 weeks (pioglitazone administration experiment) in only the mice with elevated serum total cholesterol and LDL levels.

WD containing 0.015% of ezetimibe (Ezetrol, MSD; WD + Ezetimibe group) or WD without any supplement (WD group, control) was administered to *Ldlr*⁻/⁻ mice to evaluate the effect of the lipid-lowering treatment. T0070907 [Selleckchem, #S2871, 5 µg/body weight (g)/day, 1.75 mg/mL in the vehicle] or vehicle (5% DMSO and 45% PEG300 in double-distilled water) was used to inhibit PPARγ activity. WD containing pioglitazone [Actos (pioglitazone hydrochloride), Takeda, 250 mg Actos/1 kg WD] (pioglitazone group) or WD without any supplement (control group) was administered to increase the activity of PPARγ.

## Flow cytometry
After euthanizing the mice with $CO_2$, aortic valves were harvested and isolated into a single-cell suspension at 37 °C for 30 min by rotating incubation with $Ca^{2+}/Mg^{2+}$ Dulbecco's phosphate-buffered saline ($Ca^{2+}/Mg^{2+}$ DPBS, Welgene, #LB 001-01) containing collagenase II (1000 U/mL, Sigma-Aldrich, #C6885) and DNase I (90 U/mL, Roche, #10104159001).

For the analysis of blood monocytes, peripheral blood was obtained from the retro-orbital sinus of live mice, and 10 µL of blood was used for each sample. Red blood cells (RBCs) were lysed using RBC Lysis Buffer (eBioscience, #00-4333-57), according to the manufacturer's protocol.

After a brief wash step using Dulbecco's phosphate-buffered saline (DPBS, Welgene, #LB 001-02), the cells were stained with Zombie Aqua (1:200, BioLegend, #423102) for 20 min and soaked with Fc receptor blocking antibody (anti-mouse CD16/32, 1:400, BioLegend, #101320) for 10 min, according to the manufacturer's instructions. After blocking, the cells were incubated with a mixture of fluorochrome-conjugated anti-mouse antibodies in 2% fetal bovine serum (FBS, HyClone, #SH30084.03HI) containing DPBS for 30 min at 4 °C. [1:400 for all antibodies except PE anti-CD64 (BioLegend, #139304), 1:200 for PE anti-CD64, see Supplementary Data 5 for the antibody list].

To stain CD206, intracellular staining was performed using the Foxp3/Transcription Factor Staining Buffer Set (eBioscience, #00-5523-00) following the manufacturer's protocol. Cells were fixed and permeabilized with fixation/permeabilization solution for 30 min and then intracellularly stained with Alexa Fluor 488 anti-CD206 antibody (1:400, BioLegend, #141709) for 30 min.

After staining, the cells were briefly washed, resuspended in 2% FBS containing DPBS, and passed through a 70 µm cell strainer. Data of the treated cells were recorded using the BD FACSCanto II flow cytometer with FACSDiva (v6.1.3, BD Biosciences) and analyzed using the FlowJo software (v10.5.3, FlowJo, LLC).

## VIC isolation and culture conditions
C57BL/6J mice (2- to 4-month-old) were sacrificed in a $CO_2$ chamber, and the aortic valve and mitral valve were collected in Dulbecco's modified Eagle's medium (DMEM, Sigma-Aldrich, #D6429). The aortic valve and mitral valve were minced for a minute. For culture, tissue was digested for 30 minutes with 1000 U/mL collagenase II and 90 U/mL DNase I in $Ca^{2+}/Mg^{2+}$ DPBS at 37 °C. Cell suspensions were centrifuged at $635 \times g$ for 5 min, and the obtained pellets were resuspended in DMEM supplemented with 10% fetal bovine serum (FBS), 2 ng/mL epidermal growth factor (EGF, PeproTech, #315-09), 0.5 ng/mL basic fibroblast growth factor (bFGF, PeproTech, #450-33), 5 µg/mL insulin (Sigma-Aldrich, #I0516), 10 mM HEPES, and 50 µg/mL Primocin (InvivoGen, #ant-pm-05). Cell suspensions were aliquoted into 96 well plates or 48 well plates and incubated for 3 days at 37 °C and 5% $CO_2$.

## In vitro lipoproteins uptake assay
To measure lipoprotein uptake, VICs were incubated in media (DMEM supplemented with 10% FBS and 50 µg/mL Primocin) containing 50 µg/mL of DiI-LDL (Kalen Biomedical, #770230), DiI-oxLDL (Kalen Biomedical, #770232), or DiI-VLDL (Kalen Biomedical, #770130) for 24 hours at 37 °C and 5% $CO_2$. For flow cytometry, cells were harvested with 0.05% trypsin-EDTA (Gibco, #25300-062). For visualization, cells were fixed with 4% paraformaldehyde (Wako, #163-20145) and co-stained with DAPI (Invitrogen, #D1306).

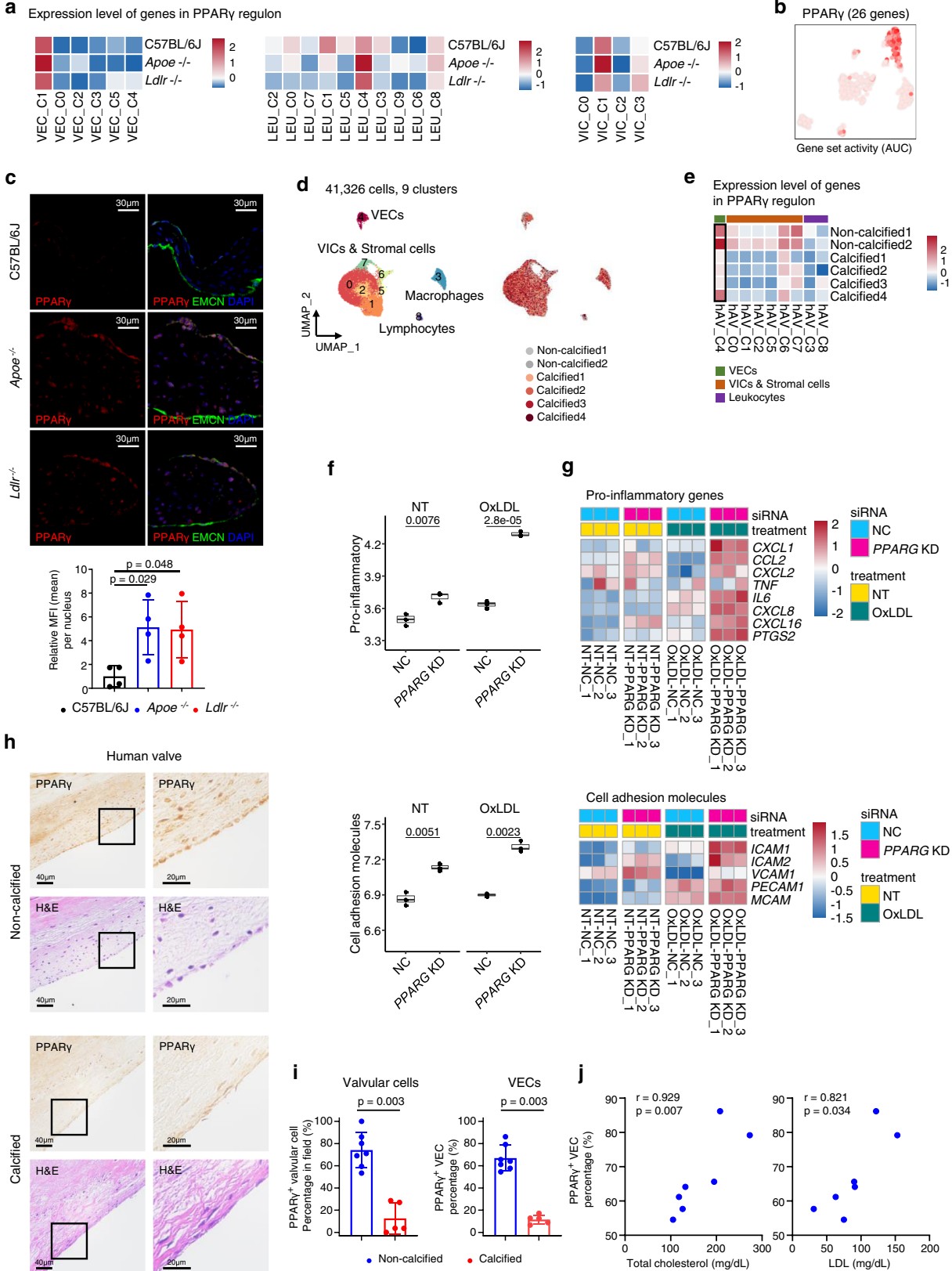

## CD36 and SR-BI Inhibition

C57BL/6J mice (2- to 4-month-old) were sacrificed in a $CO_2$ chamber, and the aortic valve attached to the aortic sinus was collected in DPBS containing 10 µM BLT-1 (SR-BI inhibitor, Sigma-Aldrich, #SML0059) or 200 µM SAB (CD36 inhibitor, Sigma-Aldrich, #SML0048) for 2 h on ice.

The aortic valve was transferred into DMEM supplemented with 10% FBS and 50 µg/mL Primocin containing 50 µg/mL of DiI-LDL and 10 µM BLT-1 or 200 µM SAB and incubated for 48 h at 37 °C and 5% $CO_2$. For flow cytometry, the tissue was digested for 30 min with 1000 U/mL collagenase II and 90 U/mL DNase I in $Ca^{2+}/Mg^{2+}$ DPBS at 37 °C.

**Fig. 7 | PPARγ pathway is activated in VECs of hyperlipidemic mice and conserved in human aortic valves. a** Average expression map of genes in PPARγ regulon, produced by Single-Cell Regulatory Network Inference and Clustering (SCENIC), for each cluster of leukocytes, VECs, and VICs. **b** UMAP plot of VECs color-coded by the activity of PPARγ regulon. Gene set activity was calculated by SCENIC. AUC: area under curve. **c** Immunostaining of PPARγ (red) and endomucin (EMCN, EC marker, green) in aortic valve with sinus from normal (chow diet) and hyperlipidemic mice ($Apoe^{-/-}$ and $Ldlr^{-/-}$ mice, WD for 16 weeks) ($n = 4$). DAPI (blue) was used to stain nuclei. The graph represents the relative MFI of PPARγ in the VECs. Kruskal–Wallis test with post-hoc Dunn's test was used. Scale bar: 30 μm. **d** UMAP plot of 41,326 single-cells derived from human aortic valve, colored by the clusters (left) and samples (right). **e** Average expression map of genes in PPARγ regulon for each cell cluster from human aortic valve. **f** Pro-inflammatory (top) and cell adhesion molecule (bottom) scores in non-targeting siRNA (NC) and *PPARG* targeting siRNA-treated (*PPARG* knockdown, KD) human VECs under no (NT) and oxLDL treatment conditions. Each score represents the average expression level of the genes, as shown in Fig. 7g ($n = 3$). Each box depicts the IQR and median of each score, whiskers indicate 1.5 times the IQR. p, two-sided *T*-test *p*-value. **g** Expression map of pro-inflammatory genes (top) and cell adhesion molecules (bottom). The expression of genes in all samples was scaled by z-transformation. **h–j** PPARγ IHC in human aortic valves ($n = 7$ for non-calcified, $n = 5$ for calcified). Representative image of PPARγ IHC (top) and H&E stain (bottom) (**h**), measurement of PPARγ$^+$ cellular proportion in valvular cells (left) and VECs (right) (**i**) and the positive correlation between PPARγ$^+$ VECs of non-calcified and the plasma levels of total cholesterol and LDL (**j**). For (**i**), two-sided Mann–Whitney test, and for (**j**), the Spearman correlation test were used. Scale bar: 40 μm (left), 20 μm (right). Image data are representative of three independent experiments unless otherwise stated. Data are presented as mean ± SD.

## Monocyte migration assay

To measure monocyte chemotaxis, VICs were incubated on a lower well of a transwell (Corning, #3388) in media (DMEM supplemented with 10% FBS and 50 μg/mL Primocin) containing 50 μg/mL of LDL (Kalen Biomedical, # 770200) or oxLDL (Kalen Biomedical, #770202) for 24 hours at 37 °C, 5% $CO_2$, and isolated monocytes ($5 \times 10^4$ cells) using the manufacturer's protocol [Mouse monocyte isolation kit (BM), Miltenyi Biotec, #130-100-629] were labeled with CFSE cell division tracker kit (BioLegend, #423801) and incubated on an upper well of transwell (Corning, #3388) for 10 h at 37 °C and 5% $CO_2$. Samples without VIC were used as control. The migrated monocytes were imaged using a confocal microscope (Nikon, C2 confocal microscope) and counted.

## Ex vivo lipoprotein uptake assay

C57BL/6J mice were sacrificed in a $CO_2$ chamber and the aortic valve attached to the aortic sinus was collected in DMEM supplemented with 10% FBS and 50 μg/mL Primocin containing 50 μg/mL DiI-LDL (Kalen Biomedical, #770230) or DiI-VLDL (Kalen Biomedical, #770130) and incubated for 48 hours at 37 °C and 5% $CO_2$. For imaging, after fixation with 4% paraformaldehyde and co-staining with DAPI, the aortic valve was imaged with a confocal microscope (Nikon). For flow cytometry, the tissue was digested for 30 min with 1000 U/mL collagenase II and 90 U/mL DNase I in $Ca^{2+}/Mg^{2+}$ DPBS at 37 °C.

## Ex vivo monocyte adhesion assay

C57BL/6J mice (2- to 4-month-old) were sacrificed in a $CO_2$ chamber and the aortic valve attached to the aortic sinus collected into media DMEM supplemented with 10% FBS and 50 μg/mL Primocin containing 50 μg/mL of LDL (Kalen Biomedical, # 770200), oxLDL (Kalen Biomedical, #770202), or 20 μM T0070907 (Sigma-Aldrich, #T8703) with 50 μg/mL of oxLDL (Kalen Biomedical, #770202). After 48 h of incubation at 37 °C and 5% $CO_2$, isolated monocytes ($1 \times 10^5$ cells) using the manufacturer's protocol [Mouse monocyte isolation kit (BM), Miltenyi Biotec, #130-100-629] were labeled with CFSE cell division tracker kit (BioLegend, #423801) and co-cultured with the aortic valve for an hour at 37 °C and 50 rpm. The attached monocytes were imaged using a confocal microscope (Nikon) and counted.

## Fluorescence-activated cell sorting (FACS)

Aortic valves were pooled and single cells were isolated as described above. The number of pooled male mice in each experimental group was as follows: C57BL/6J ($n = 30$), $Ldlr^{-/-}$ ($n = 24$), and $Apoe^{-/-}$ group ($n = 19$). Only live single-cells [propidium iodide (PI) negative single cells] were sorted using BD FACS Aria III and then aligned using the 10x Chromium pipeline to construct the single-cell cDNA library (Supplementary Fig. 6).

## scRNA-seq and read processing

The single-cell suspensions were subjected to 3′ single-cell RNA sequencing aiming for target recovery of 5,000 cells (C57BL/6J or $Ldlr^{-/-}$) and 3,000 cells ($Apoe^{-/-}$) using Single Cell A Chip Kit, Single Cell 3′ Library, and Gel Bead Kit V2, and i7 Multiplex Kit (10x Genomics). Solutions pertaining to 3′ chemistry and v2 of 10x Genomics were used to barcode individual cells. Libraries were sequenced on an Illumina HiSeq2500 and mapped to the mouse genome (build mm10) using the *Cell Ranger* toolkit (v3.0.2). Single-cell RNA sequencing parameters were summarized in Supplementary Table 2.

## Single-cell data quality control and normalization

We applied two quality measures on single cells, the *Cell Ranger* toolkit (v3.0.2): mitochondrial genes (<10%) and gene count (range from 200 to 5000) from the R package *Seurat* v3.1.1 (https://satijalab.org/seurat/)[67]. The UMI matrix on the selected cell barcodes was log-normalized using *the NormalizeData* function and then used in downstream analysis after z-transformation using *the ScaleData* function. To improve clustering of VICs, we further extracted cell clusters expressing canonical VIC markers (*Vim, Col1a1, Col1a2*, and *Dcn*) but not detecting the expression of other cell type markers. A total of well-defined 1732 VICs out of 2878 VICs and stromal cells were used in downstream analysis.

## Unsupervised dimensional reduction and clustering

Top-ranked 2000 variably expressed genes were selected using the 'vst' method implemented in *FindVariableFeatures* function of the *Seurat* package. These genes were then used to correct the effect of technical batches by applying the *FindIntegrationAnchors* and *IntegrateData* functions with a total of 15 or 20 anchors (all single cells, leukocytes, and VICs, 20 anchors; VECs, 15 anchors). Cell clustering and UMAP visualization were then performed using the *FindClusters* (resolution = 0.3) and *RunUMAP* functions. Cluster results were visualized using UMAP to verify that the graphically identified clusters were captured. Cell types were assigned based on known marker gene expression, considering the significantly expressed genes in each cell cluster.

## Differentially expressed marker gene analysis

We identified significantly expressed genes in each cell cluster using the *FindAllMarkers* function (default parameters) of the Seurat package. The significance of the difference was determined using the two-sided Wilcoxon rank sum test with Bonferroni correction. We selected marker genes based on the fraction of expressing cells (>25% of cells within either of the two cell groups, marked as pct) and the statistical threshold (log fold change > 0.25, *p*-value < 0.01, and adjusted *p*-value (Bonferroni) < 0.01).

## Trajectory analysis

The cell state transitions of macrophages were estimated using the *Monocle* v2 algorithm[68]. The UMI matrix for cells defined as macrophages (LEU_C0, C1, C2, C4, C5, and C7) was submitted to *Monocle*. An

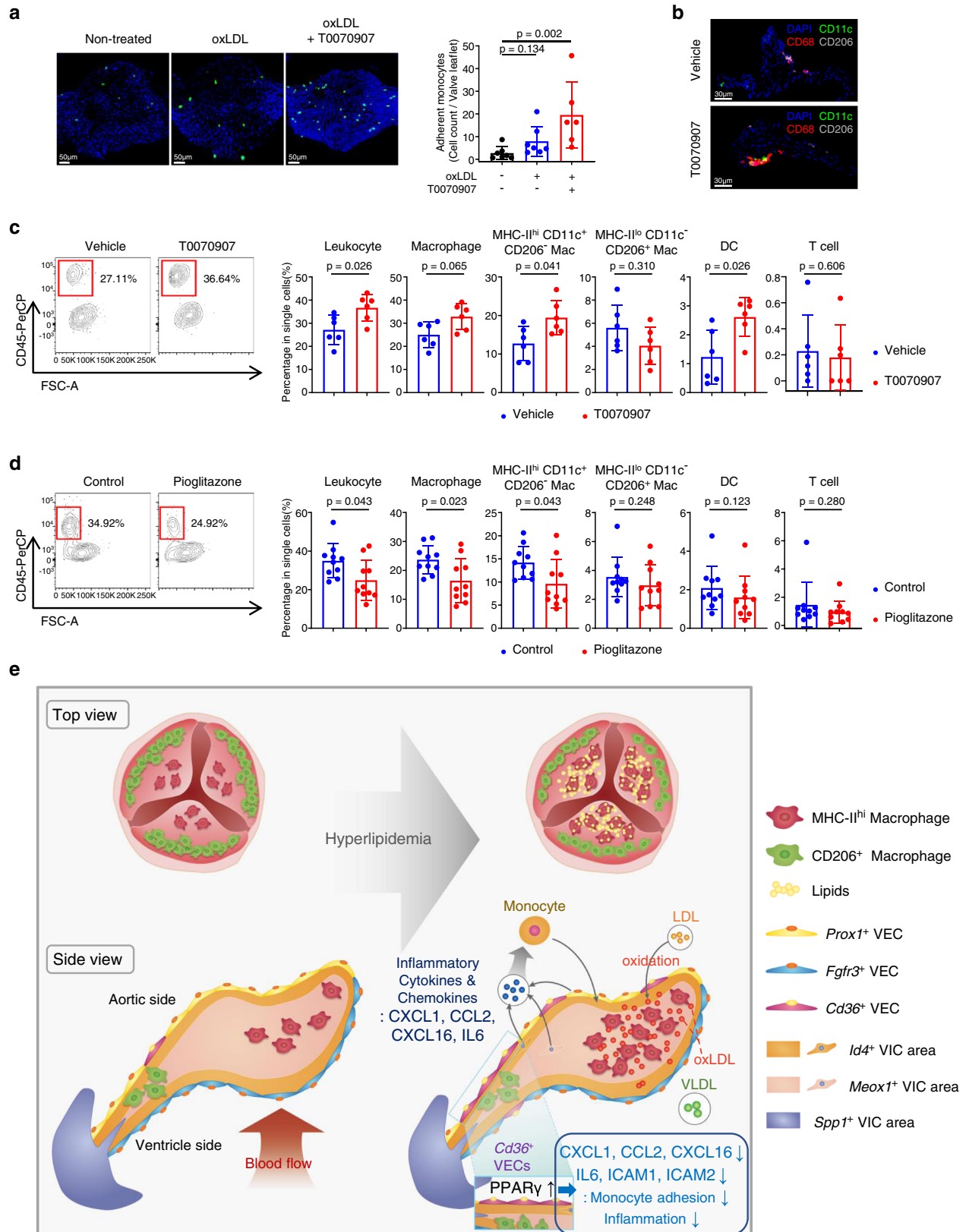

object was created using the *newCellDataSet* function with the parameter expressionFamily = negbinomial.size. Then, the *dispersionTable* function was used to select variably expressed genes based on the variance over the fit curve and an average expression >0.001. The cell trajectory was inferred through dimension reduction and cell ordering using default *Monocle* parameters.

## Projection of functional features of macrophage

To demonstrate the functional features of macrophages according to Monocle pseudotime and cell clusters, we used the *ggplot2* (v3.3.3) R package to visualize the expression pattern of pro- and anti-inflammatory scores. Pro-inflammatory represents mean expression of featured genes: *Il1b, Tnf, Ccl2, Cxcl10, Cxcl2, H2-Ab1*, and *Itgax*. Anti-

**Fig. 8 | PPARγ activation protects the aortic valve against the inflammation.**
**a** Adhesion assay of monocytes to ex vivo cultured aortic valve treated with oxLDL and/or inhibition of PPARγ by T0070907 ($n = 7$ for non-treated, $n = 7$ for oxLDL, $n = 6$ for oxLDL+T0070907). Scale bar: 50 μm. **b**, **c** Pro-inflammatory effect of T0070907 (PPARγ antagonist) on mouse aortic valve in vivo. $Ldlr^{-/-}$ mice were intraperitoneally injected daily with vehicle or T0070907 for 10 weeks with a WD feeding. Representative IHC images (**b**) and flow cytometry analysis presenting percentage of each immune cell subset ($n = 6$) (**c**). Scale bar: 30 μm. **d** Flow cytometry analysis showing anti-inflammatory effect of PPARγ activation by pioglitazone on mouse aortic valve in vivo. PCSK9-AAV-injected C57BL/6J mice were fed with pioglitazone-containing WD or normal WD for 6 weeks. Graphs present percentage of each immune cell subset ($n = 10$). **e** Proposed pathogenesis model of the early-stage aortic valve disease induced by hyperlipidemia. In hyperlipidemic states, oxidized LDL triggers aortic valvular inflammation by enhancing the productions of various cytokines and chemokines, leading to the recruitment of monocyte-derived MHC-II$^{hi}$ macrophages. Meanwhile, PPARγ activation during hyperlipidemia inhibits the accumulations of monocytes and macrophages in the aortic valve. Top view (left, top), side view (left, bottom), and legends (right). VIC: valvular interstitial cell. VEC: valvular endothelial cell. Image data are representative of three independent experiments unless otherwise stated. Two-sided Mann–Whitney test (comparison of two groups) and Kruskal-Wallis test with post-hoc Dunn's test (comparison of three or more groups) were used for group comparisons. Data are presented as mean ± SD.

inflammatory score represents mean expression of featured genes: *Mrc1, Lyve1, Folr2, Cbr2,* and *Il10*. In the dot plots for inflammatory score and pseudotime, we used *geom_smooth* function with the parameter method = loess to determine trend line and calculated Pearson's correlation using *cor.test* function from the R package *stats* (v3.6.3). In the boxplots of macrophage functional features in each cluster, we performed one-way ANOVA test using *stat_compare_means* function with the parameter method = anova from the *ggpubr* (v0.4.0) R package.

### Pathway analysis
Cluster-specific gene signatures were categorized according to the gene catalogs in the Kyoto Encyclopedia of Genes and Genomes (KEGG) database. For the differentially expressed genes between knockout and wild-type models, gene set analysis was performed using a set of inflammation-related and monocyte-related gene sets selected from Gene Ontology (GO) Biological Process ontology. Gene sets were collected from the MsigDB database using the *msigdbr* (v7.1.1) package. The gene list was ranked by the log fold change. The R package *fgsea* (v1.12.0)[69] was used to perform gene set enrichment analysis (default parameters) and selected the over-enriched pathways satisfying the adjusted *p*-value < 0.05.

### Transcription factor activity analysis
To investigate transcription factor activity, SCENIC[27] was used with cisTarget_databases: mm10_refseq-r80_500bp_up_and_100bp_down_tss.mc9nr.feather and mm10_refseq-r80_10kb_up_and_down_tss.mc9nr.feather (SCENIC v1.1.2, which corresponds to RcisTarget v1.6.0, AUCell v1.8.0, and GENIE3 v1.8.0). The input matrix was the log-normalized expression matrix from which genes passed the filtering (default parameters). GENIE3 was used to infer co-expression networks using runGenie3 function. Gene regulatory networks were constructed and scored using the following functions of SCENIC; runSCENIC_1_coexNetwork2modules, runSCENIC_2_createRegulons, and runSCENIC_3_scoreCells.

### Selection of cells with high PPARγ regulon expression and their specific genes
The valvular cells were divided into two classes (PPARγ high and low) along the 90th and 10th percentiles of the mean expression of the genes in PPARγ regulon. A total of 543 and 1,621 valvular cells (of these, 181 and 83 cells in VEC_C0, C1, and C2) were classified into PPARγ high and low groups. We calculated the differential expression levels of genes specific to PPARγ high cells compared to low cells using the *FindMarkers* function (default parameters) of the Seurat package. The significance of the difference was determined using the two-sided Wilcoxon rank sum test with Bonferroni correction. We selected significantly expressed genes based on the fraction of expressing cells (>25% of cells within either of the two cell groups, marked as pct) and the statistical threshold (log fold change > 0.25, *p*-value < 0.01, and adjusted *p*-value (Bonferroni) < 0.01).

### Sample preparation and bulk tissue RNA sequencing of mouse aortic valves
Male $Apoe^{-/-}$ and $Ldlr^{-/-}$ mice (6-week-old) fed a WD for 8 weeks (sacrificed at 14-week-old) to induce aortic valve disease with lipid accumulation by hypercholesterolemia. For the control group, chow diet-fed, 14-week-old male C57BL/6 J (wild type), $Apoe^{-/-}$, and $Ldlr^{-/-}$ mice were used. Mice were euthanized by $CO_2$ inhalation, and their aortic valves were collected into the cold DPBS. For each sample, isolated aortic valves from 10 mice were pooled into RNAlater Stabilization Solution (Invitrogen, # AM7021). After stabilization, RNA was extracted from aortic valves, using the TRIzol Reagent (Invitrogen, #15596026). cDNA libraries for sequencing were constructed from 0.5 μg of total RNA, using the HiSeq 3000/4000 SBS Kit (Illumina, # FC-410-1003) according to the manufacturer's protocol. Libraries were sequenced using the Illumina HiSeq 4000 system with 101-bp read length and paired-end reads.

### Human aortic VEC isolation and culture conditions
Human aortic valves from the patients were collected in a cold, sterile saline solution. After incubation with collagenase II solution [600 U/mL in EBM-2 Endothelial Cell Growth Basal Medium-2 (Lonza, #CC-3156) with 10% FBS and GA-1000 (1:1000, Lonza, # CC-4083)] for 10 min at 37 °C, human aortic VECs were isolated from the aortic valve tissue, by a gentle rolling of pre-soaked sterile cotton swabs, according to a previous report by Gould & Butcher[70]. Isolated VECs were cultured in EGM-2 Endothelial Cell Growth Medium-2 BulletKit (Lonza, #CC-3162), supplemented with 10% FBS in a humidified $CO_2$ incubator (5% $CO_2$, 37 °C). When 90 % confluency was reached, cells were trypsinized and then subcultured, at a 1:3 ratio. Gelatin-coated cell culture flasks (T25 for passage 0 and T75 for passage 1 or higher) were used for cell culture. VECs at passage 1-2 were used for further experiments.

### *PPARG* gene silencing in human aortic VECs by siRNA transfection
Human aortic VECs ($5 \times 10^5$ cells per sample) at passage 1–2, were transfected with siRNA targeting *PPARG* (AccuTarget Predesigned siRNA, Bioneer, pooling of 3 siRNA, #SDO-1001: siRNA #5468-1, -2, and -3) or negative control siRNA (Bioneer, #SN-1003) by electrophoresis (750 nM of final siRNA concentration, 1 pulse, voltage: 1300 V, width: 30 ms) using Neon Transfection System (Invitrogen). The human aortic VECs were then transferred into gelatin-coated six-well plates with EGM-2 medium supplemented with 10% FBS. 9 h after the transfer, media changes were performed. And 48 h after transfection (including serum starvation for 1 h), cells were treated with or without oxLDL (150 μg/mL, Kalen Biomedical, #770252), for 24 h and then subjected to RNA extraction for RNA sequencing.

### Library construction and RNA sequencing of human aortic VECs
Total RNA was extracted from human aortic VECs at passage 2–3, using the TRIzol Reagent (Invitrogen, #15596026). cDNA libraries for sequencing were constructed from 1 μg of total RNA, using a TruSeq

Stranded mRNA LT Sample Prep Kit (Illumina, #RS-122-2101 & RS-122-2102) according to the manufacturer's protocol. Libraries were sequenced using the Illumina NovaSeq 6000 system with paired-end (2 × 100 bp) reads.

## Bulk RNA sequencing data processing

The RNA reads were aligned to the reference sequences (mouse, GRCm38; human, GRCh38) and quantified as transcripts per million (TPM) using STAR (v2.7.8a) and RSEM (v1.3.1) using the function *rsem-calculate-expression* with the following parameters:–paired-end–star–estimate-rspd. The TPM count for the genes in each sample was log-normalized and used in the log2 scale TPM plus 1, for subsequent analyses.

## *PPARG* knockdown-specific gene analysis

Differentially expressed genes between the two sample groups (NT-PPARG KD versus NT-NC, and OxLDL-PPARG KD versus OxLDL-NC) were identified using the *DESeq* function (default parameters for Wald test) of the DESeq2 (v.1.26.0) package. We selected genes based on the statistical threshold (>1.5 fold change, $p$-value < 0.05, and adjusted $p$-value < 0.05).

## scRNA-seq analysis of human aortic valve

We obtained raw 3′ scRNA-seq data on human aortic valve leaflets from two healthy and four stenosis donors[33]. Sequencing libraries were mapped to the human genome (build GRCh38) using the Cell Ranger toolkit (v3.0.2). Then, we selected single cells satisfying the following two quality measures: mitochondrial genes (<10%) and gene count (range from 200 to 5,000) from the R package Seurat v3.1.1 (https://satijalab.org/seurat/)[67]. The UMI matrix on the selected cell barcodes was log-normalized using the *NormalizeData* function and used in downstream analysis after z-transformation using the *ScaleData* function. The top-ranked 2000 variably expressed genes were selected using the 'vst' methods implemented in *FindVariableFeatures* function of the Seurat package. These genes were then used to correct the effect of technical batches by applying the *FindIntegrationAnchors* and *IntegrateData* functions with a total of 20 anchors. Cell clustering and UMAP visualization were then performed using the *FindClusters* (resolution = 0.3) and *RunUMAP* functions. Cluster results were visualized using UMAP to verify that the graphically identified clusters were captured. Cell types were assigned based on known marker gene expression, considering the significantly expressed genes in each cell cluster.

## Meta-analysis of endothelial cell diversity

We obtained 3′ scRNA-seq data on normal aorta[71], and heart[72], from a C57BL/6J mouse model. UMI matrices were collected from published studies using the 10x Genomics platform. ECs were selected by in silico sorting based on the expression of *Pecam1* and *Cdh5* genes after cell quality control. We performed integrated dimensional reduction and clustering for aortic, heart, and valvular endothelial cells, including the data generated by us.

## Histology and plasma lipid measurement

Mouse hearts were perfused with 4% paraformaldehyde phosphate buffer solution, followed by tissue fixation for 2 h (O.C.T. compound embedded) or 16 h (paraffin-embedded) with the same solution as that used for perfusion. For histological analysis, Hematoxylin and Eosin (H&E) staining (BBC Biomedical, #MA0101035), PPARγ immunostaining, and RNA in situ hybridization were performed using paraffin sections. Standard tissue processing was performed to prepare the paraffin-embedded tissue. Processed heart tissue was embedded in paraffin and then sectioned into 4 μm sections. For the measurement of human PPARγ+ valvular cells, the average percentage of PPARγ+ valvular cells at least five random fields (22,500 μm² per field) was used. For the measurement of human PPARγ+ VECs, the percentage of total PPARγ+ VECs in total VEC count was used.

Frozen sections were used for Oil Red O staining and immunofluorescence. Fixed mouse hearts were embedded in O.C.T compound (Sakura Finetek, #4583) and frozen at −80 °C for 4 h. Frozen tissue blocks were serially sectioned into 7 μm sections from the beginning to the end of the aortic valve. To measure the aortic valve lesion area, frozen sections at 77 μm intervals were Oil Red O stained using a Fat stain kit (BBC Biomedical, #SSK5019), according to the manufacturer's protocol. The lesion area was quantified as a percentage of the Oil Red O-positive area in the total aortic valve area using NIS Elements (v4.30, Nikon) and Photoshop (v22.3.0, Adobe).

For whole-mount aortic valve Oil Red O staining, Fat stain kit (BBC Biomedical, #SSK5019) was used with a modified procedure. The aortic valve was rinsed with 100% propylene glycol and the tissue was stained with Oil Red O working solution for an hour at 37 °C. After brief washing with 85% propylene glycol, the stained tissue was mounted on a glass slide and imaged using brightfield microscopy (Nikon, Eclipse 50i with DS-Ri2).

To analyze the lipid profile, blood was obtained from the retro-orbital sinus. Plasma was collected from blood and centrifuged at 2000 × *g* for 20 min at RT. The levels of total cholesterol, triglycerides, HDL, and LDL were analyzed using an automated blood chemical analyzer (Hitachi). For an assessment of comparative analysis between *Ldlr*⁻/⁻ and *Apoe*⁻/⁻ mice fed WD feeding for 4-, 8-, or 12-weeks, the data and tissue-sections from Kim et al., were used[20].

## Immunostaining

Mouse hearts were fixed for 2 h in 4% paraformaldehyde phosphate buffer solution. Aortic valves were harvested from the fixed hearts and placed in a 96-well plate. For whole-mount immunostaining, aortic valves were permeabilized with 0.5% Triton X-100 for 30 min and then non-specific binding was blocked using 1% normal donkey serum for 1 hour. Subsequently, the aortic valves were incubated in primary antibodies (1:200, see Supplementary Data 5) in 1% normal donkey serum for 16 h at 4 °C with gentle shaking. The next day, valves were washed using DPBS with gentle shaking, and then incubated with secondary antibodies (1:400, see Supplementary Data 5) or Alexa Fluor 594-steptavidin (1:400, Invitrogen, #S11227), following the same procedure as that in primary antibody incubation. The valves were then washed with DPBS and mounted on slides using coverslips and mounting medium with DAPI (Vector Laboratories, #H-1200-10). Images were acquired as Z-stacks using a confocal microscope (Nikon; or Zeiss, LSM 780). The acquired images were analyzed using NIS Elements (v4.30, Nikon), ZEN (v8.1, Zeiss), ImageJ (v1.53c, National Institutes of Health, NIH), and Imaris (v9.0.2, Bitplane).

For immunostaining of CD68, vimentin or PROX1, frozen sections of aortic valves were rehydrated with DPBS, and endogenous peroxidase was quenched with 3% hydrogen peroxide and blocked with 1% normal donkey serum for 1 h. After blocking, the sections were incubated with a mixture of anti-CD68 (Bio-Rad, #MCA1957) and anti-vimentin (Abcam, #ab92547) antibodies or anti-PROX1 (R&D Systems, #AF2727) antibodies (all 1:200) in 1% normal donkey serum for 16 h at 4 °C. After washing, slides were incubated in a mixture of HRP anti-rat IgG (Jackson ImmunoResearch, #712-035-153) and Alexa Fluor 647 anti-rabbit IgG (Invitrogen, #A-31573) or Alexa Fluor 488 anti-goat IgG (Invitrogen, #A-11055) antibodies (all 1:400) in 1% normal donkey serum for 12 h at 4 °C. To detect CD68, the slides were treated with an Alexa Fluor 594-tyramide signal amplification kit (Invitrogen, #T20935) according to the manufacturer's protocol. In immunostaining of CD68 and vimentin, the slides were incubated for 1 h with 1 μM BODIPY 493/503 (Invitrogen, #D3922) solution to stain lipid droplets. Lastly, the slides were washed with DPBS and coverslipped using a mounting medium with DAPI (Vector Laboratories, #H-1200-10).

To immunostain PPARγ, the paraffin sections of aortic valves were deparaffinized, rehydrated, and double-boiled for 10 minutes in the Diva Decloaker (Biocare Medical, #DV2004) solution for antigen retrieval. After endogenous peroxidase quenching with 3% hydrogen peroxide and blocking with 1% normal donkey serum, the slides of mouse samples were incubated with anti-PPARγ (1:50, Cell Signaling Technology, #2435) and anti-EMCN antibodies (1:200, Abcam, # ab106100), whereas the slides of human samples were incubated only with the anti-PPARγ antibody (1:50, Cell Signaling Technology, #2435), in 1% normal donkey serum for 20 h at 4 °C. For the mouse samples, after primary antibody incubation, the slides were washed and then incubated with Cy3 anti-rabbit IgG (1:200, Jackson ImmunoResearch, #711-165-152) and Alexa Fluor 488 anti-rat IgG (1:200, Invitrogen, #A-21208) in 1% normal donkey serum for 4 h at room temperature. The slides were coverslipped using a mounting medium with DAPI (Vector Laboratories, #H-1200-10), after brief washing with DPBS. For human samples, VECTASTAIN Elite ABC-HRP Kit (Vector Laboratories, #PK-6101) and DAB Substrate (Vector Laboratories, #SK-4105) were used, according to the manufacturer's protocol.

For immunocytochemistry of CD31, human aortic VECs at passage 2–3 were cultured in gelatin-coated, four-well chamber slides (Thermo Scientific, # 154526PK). The cells in slides were briefly washed with DPBS, fixed with 4% paraformaldehyde, and then permeabilized with 0.5% Tween 20 in DPBS. After blocking with 1% normal donkey serum in PBST (DPBS with 0.1% Tween 20), the slides were incubated with anti-CD31 antibody (1:200, BioLegend, #303101) in 1% normal donkey serum in PBST for 16 h at 4 °C. After that, the slides were washed and then incubated with Alexa Fluor 488 anti-mouse IgG (1:200, Jackson ImmunoResearch, #715-545-151) in 1% normal donkey serum in PBST for 2 h at room temperature. After brief washing with DPBS, the slides were coverslipped using mounting medium with DAPI (Vector Laboratories, #H-1200-10).

Fluorescence images were captured using a confocal microscope (Nikon) or fluorescence microscope (Nikon, Eclipse 50i with DS-Ri2). Brightfield microscopy (Nikon) was used to obtain bright field images. Captured images were analyzed using NIS Elements (Nikon), ImageJ (NIH), and Imaris (Bitplane). For the quantification of PPARγ, ImageJ (NIH) was used.

## Western blotting analysis

Human aortic VECs, at passage 2–3, were lysed in RIPA buffer (GenDE-POT, #R4100-010) containing a protease inhibitor cocktail (GenDEPOT, #P3200-001). Protein samples were separated by SDS-PAGE and transferred into the polyvinylidene fluoride (PVDF) membrane (Millipore, #IPVH00010). After blocking with 5% skim milk in tris-buffered saline with 0.1% Tween 20 (TBST), the membrane was incubated with an anti-PPARγ antibody (1:500, Santa Cruz Biotechnology, #sc-7273) in 5% skim milk for 14 h at 4 °C, and then incubated with HRP anti-mouse IgG antibody (1:5000, BioLegend, #405306) in 5% skim milk, for 2 h at room temperature. After obtaining the PPARγ signal, antibodies were stripped from the membrane using BlotFresh Western Blot Stripping Reagent (SignaGen, #SL100324) according to the manufacturer's protocol. The membrane was blocked again with 5% skim milk and incubated with anti-GAPDH antibody (1:1000, Cell Signaling Technology, # 2118S, 14 h at 4 °C) as the primary antibody, and then incubated with HRP anti-rabbit IgG antibody (1:10,000, Jackson ImmunoResearch, #711-035-152, 1 h at room temperature). Immobilon Forte Western HRP substrate (Millipore, #WBLUF0100) was used for signal development, and a chemiluminescence imaging system (Fusion SL, Vilber Lourmat, France) with software (Evolution-Capt, v17.04a, Vilber Luormat) was used for signal capture and image processing. ImageJ (NIH) was used for image analysis.

## RNA in situ hybridization (RNAscope)

To detect in situ expression of marker genes, RNA in situ hybridization (RNAscope, ACDbio) was used, according to the manufacturer's instructions (RNAscope® 2.5 HD Duplex Reagent Kit, ACDbio, #322430 and ACDbio user manual document #322452 & 322500). Formalin-fixed paraffin sections of 4 μm thickness were used for the RNAscope assay. Amplified probes were detected by brightfield (C1, C2) or fluorescence (C2) microscopy (Nikon) and analyzed using the NIS Elements (Nikon) and ImageJ (NIH). For the quantification of *Cd36*, ImageJ (NIH) was used. The following probes were used in this study: Meox1 (ACDbio, #530641-C2), Id4 (ACDbio, #447861-C2), Rspo3 (ACDbio, #402011), Chad (ACDbio, #484881), Spp1 (ACDbio, #435191-C2), Fgfr3 (ACDbio, #440771-C2), Cd36 (ACDbio, #464431-C2), Cx3cl1 (ACDbio, #426211-C2), Csf1 (ACDbio, #315621), Clec3b (ACDbio, #539561-C2), Dpep1 (ACDbio, #480831), Polr2ra (ACDbio, #312471-C2), Ppib (ACDbio, #313911), and DapB (ACDbio, # 310043 & 310043-C2).

## Statistical analysis

Statistical methods used in the scRNA-seq analysis are addressed in each section of Materials and Methods. GraphPad Prism (v9.4.1, GraphPad) was used to perform statistical analysis and graph representation of all in vivo and in vitro data. Unless otherwise stated, data are shown as mean ± standard deviation (SD), and the nonparametric analysis was used because of the small sample size ($n < 30$). Two-sided Mann–Whitney test was used for the comparison between two groups, and Kruskal-Wallis test with post-hoc Dunn's test was used for comparing three or more groups. For correlation analysis, the Spearman correlation test was used. The $p$ value of two-sided Mann–Whitney test, post-hoc Dunn's test, and Spearman correlation test with correlation coefficient (r) is presented on the respective graph. Statistical significance was set at $p < 0.05$.

## Reporting summary

Further information on research design is available in the Nature Research Reporting Summary linked to this article.

## Data availability

All data associated with this study are present in the main text or the supplementary materials. The RNA-seq data generated in this study have been deposited in the GEO database under accession code "GSE180278", "GSE205587", and "GSE206927". The publicly available RNA-seq data used in this study are available in the BioProject database under accession code "PRJNA562645"[33], and in the ArrayExpress database under accession code "E-MTAB-7149"[71], and "E-MTAB-8077"[72]. "GRCh38" and "GRCm38" were used for the reference genome. All other relevant data supporting the key findings of this study are available within the article and its Supplementary Information files or from the corresponding author upon reasonable request. Source data are provided with this paper.

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

## Acknowledgements

We acknowledge Taek Chang Lee from Korea Research Institute of Bioscience and Biotechnology (KRIBB) and Sookyoung Kim from Samsung Genome Institute, for their technical supports. We acknowledge Jia Park from D-Lab of Hanyang Institute of Technology for designing the graphical summary of the proposed pathogenesis model. We also acknowledge the KREONET/GLORIAD service provided by the Korea Institute of Science and Technology Information. This work was supported by the National Research Foundation of the Korea grant (NRF 2016M3A9D5A01952413; J.-H.C., 2021R1A2C3004586; J.-H.C., and 2016R1A5A1011974; H.-O.L.) and the National Institutes of Health (R01 DK126753; K.-W.K.).

## Author contributions

Conceptualization, S.H.L.-1, N.K., M.K., H.-O.L., and J.-H.C.; Methodology, S.H.L.-1, N.K., M.K., H.-O.L., and J.-H.C.; Formal analysis, S.H.L.-1, N.K., and M.K.; Investigation, S.H.L.-1, N.K., M.K., S.-H.W., I.H., J.P., K.K.-1, K.S.P., K.K.-2, D.S., S.P., J.Y.Z., and T.K.K.; Resources, D.-M.G., D.-Y.K., W.K.Y., S.-P.L., J.C., K.-W.K., J.H.P., S.H.L.-2, S.L., S.A., S.-H.L., H.-S.A., S.C.J., G.T.O., and W.-Y.P.; Writing—Original draft, S.H.L.-1, N.K., M.K., H.-O.L., and J.-H.C.; Writing—Review and editing, S.H.L.-1, N.K., M.K., H.-O.L., and J.-H.C.; Visualization, S.H.L.-1, N.K., and M.K.; Supervision, H.-O.L., and J.-H.C.; Funding acquisition, K.-W.K., H.-O.L., and J.-H.C.

## Competing interests

The authors declare no competing interests.

## Additional information

[1]Department of Life Science, College of Natural Sciences, Hanyang Institute of Bioscience and Biotechnology, Research Institute for Natural Sciences, Hanyang University, Seoul 04763, Republic of Korea. [2]Department of Microbiology, College of Medicine, The Catholic University of Korea, Seoul 06591, Republic of Korea. [3]Department of Biomedicine and Health Sciences, Graduate School,  The Catholic University of Korea, Seoul 06591, Republic of Korea. [4]Department of Veterinary Pathology, College of Veterinary Medicine, Seoul National University, Seoul 08826, Republic of Korea. [5]Laboratory Animal Resource Center, Korea Research Institute of Bioscience and Biotechnology (KRIBB), Cheongju 28116, Republic of Korea. [6]Cardiovascular Center and Department of Internal Medicine, Seoul National University Hospital, Seoul National University College of Medicine, Seoul 03080, Republic of Korea. [7]Samsung Genome Institute, Samsung Medical Center, Seoul 06351, Republic of Korea. [8]Department of Pharmacology and Regenerative Medicine, The University of Illinois College of Medicine, Chicago, IL 60612, USA. [9]Division of Endocrinology and Metabolism, Department of Internal Medicine, Hanyang University College of Medicine, Seoul 04763, Republic of Korea. [10]Division of Cardiovascular Surgery, Department of Thoracic and Cardiovascular Surgery, Severance Hospital, Yonsei University College of Medicine, Seoul 03722, Republic of Korea. [11]Integrative Research Center for Cerebrovascular and Cardiovascular Diseases, Yonsei University College of Medicine, Seoul 03722, Republic of Korea. [12]Division of Cardiology, Department of Internal Medicine, Severance Hospital, Yonsei University College of Medicine, Seoul 03722, Republic of Korea. [13]Division of Cardiology, Department of Internal Medicine, Uijeongbu St. Mary's Hospital, College of Medicine, The Catholic University of Korea, Uijeongbu 11765, Republic of Korea. [14]Catholic Research Institute for Intractable Cardiovascular Disease (CRID), College of Medicine, The Catholic University of Korea, Seoul 06591, Republic of Korea. [15]Department of Thoracic and Cardiovascular Surgery, Uijeongbu St. Mary's Hospital, College of Medicine, The Catholic University of Korea, Uijeongbu 11765, Republic of Korea. [16]Heart-Immune-Brain Network Research Center, Department of Life Science, Ewha Womans University, Seoul 03760, Republic of Korea. [17]Department of Molecular Cell Biology, Sungkyunkwan University School of Medicine, Suwon 16419, Republic of Korea. [18]Department of Health Sciences and Technology, Samsung Advanced Institute for Health Sciences &Technology, Sungkyunkwan University, Seoul 06355, Republic of Korea. [19]These authors contributed equally: Seung Hyun Lee, Nayoung Kim, Minkyu Kim. [20]These authors jointly supervised this work: Hae-Ock Lee, Jae-Hoon Choi. ✉e-mail: haeocklee@catholic.ac.kr; jchoi75@hanyang.ac.kr

