## [Peer Review File · Nature Communications]

REVIEWER COMMENTS

Reviewer #1 (Remarks to the Author):

This manuscript describes the regulation of valvular disease in mice models. Using sophisticated analysis of cellular transcriptomes, the authors report that a detailed analysis of changes in compositions of the valve during disease onset. Among the regulated genes, the PPAR γ pathway was predominantly regulated in one cluster of VEC of diseased mice. While inhibition of PPAR γ further increased valve inflammation, pharmacological activation of PPAR γ reduced valve disease. The study comprises extensive data and is generally well performed. However, the findings are not very surprising and the role of PPARs in valve disease has been previously reported (see review www.ncbi.nlm.nih.gov/pmc/articles/PMC6719701/).

Specific comments

1. Previous studies showed a down-regulation of CD36 in diseased human aortic valves (DOI:<https://doi.org/10.1016/j.atherosclerosis.2014.05.933>). In Figure 6e, the authors showed a representative image of increased CD36 expression. However, these data are not quantified. An analysis of CD36 in early and later stages of the disease should be done to determine if the induction is indeed an early protective mechanism, which may be lost at later stages of the disease.
2. Please provide more controls for the single cell seq data sets (e.g. number of reads, mapped reads, genes/cell, UMI/cell, total genes per sample) to make sure that the quality is similar among all studied samples.
3. Mean values and group comparisons should be provided for Suppl. Table 1.

Reviewer #2 (Remarks to the Author):

The study by Lee SH et al., deciphers the cellular mechanisms underlying aortic valvular inflammation, the initial process of aortic valve diseases, in hyperlipidemic mouse models. First, the authors showed an association between LDL and increased valvular inflammation in ApoE $^{-/-}$ and Ldlr $^{-/-}$ mice. Next, single cell RNA sequencing analysis of aortic valves at steady state and under hyperlipidemic conditions revealed a marked cell heterogeneity among valvular interstitial cells (VIC), valvular endothelial cells (VEC) and leukocytes. Moreover, the authors showed that VECs control the recruitment of monocyte-derived macrophage through a PPAR γ dependent pathway. Based on these findings, the authors propose PPAR γ as a target to control valvular inflammation in the pre-calcified stage of aortic valve disease.

The study is well designed and comprehensive. It provides some mechanistic insight in aortic valvular inflammation and thus improves our knowledge on the pathophysiology of aortic valve disease and may offer potential therapeutics target. Nevertheless, some points should be addressed to further strengthen the conclusions of this study.

1. In figure 1, the authors analyzed lipid accumulation in blood and aortic valves in WT, Apoe^{-/-} and Ldlr^{-/-} mice. Despite similar plasma levels of total cholesterol and LDL in Apoe^{-/-} and Ldlr^{-/-}, Ldlr^{-/-} showed increased accumulation of lipids and LDL in aortic valve. What about macrophage accumulation, is it also different in aortic valve from Apoe^{-/-} and Ldlr^{-/-} mice? This should be addressed since the authors use this parameter to evaluate inflammation. Explanation for the difference in LDL accumulation in aortic valves between Apoe^{-/-} and Ldlr^{-/-} mice should be provided. Is this decreased in lipid accumulation in aortic valve from Apoe^{-/-} mice explained by differences in expression of LDL scavenger receptors. It is not clear which mice were used in Fig1h and j, the information is missing from the result and figure legend.
2. The authors showed that increasing plasma LDL levels in Apoe^{-/-} mice enhanced lipid and macrophage accumulation in aortic valves. To further support this causal relationship between lipid accumulation and aortic valvular inflammation, the authors should consider analyzing the effect of reducing LDL levels in Ldlr^{-/-} mice on aortic valve inflammation.
3. Single cell RNA sequencing analysis revealed heterogeneity in VIC, VEC and leukocytes. VICs show pro-inflammatory features. What about the VECs?. Among the VEC subpopulations, one express Prox-1. Are these cells lymphatic endothelial cells?
4. Single cell RNA sequencing analysis also reveal the activation of PPAR-g pathway in VECs. Is this also observed in VICs and macrophages?
5. The staining of PPAR α is not obvious and costaining with a marker of VECs should be performed in Fig 7e. For this experiment, the authors used section from Apoe^{-/-} which showed less lipid accumulation and inflammation. What about in Ldlr^{-/-} mice? To confirm the activation of PPAR α in VECs, the cytoplasmic and nuclear cellular localization of PPAR α should be analyzed and/or the analysis of PPAR α targeted genes should be examined. This should also be performed for VIC and macrophages since this pathway is also activated in these cells based on the single cell RNA sequencing.
6. The authors propose that PPAR α pathway regulates the recruitment of monocyte into the valve. Do the activation or inhibition of PPAR α also affect the frequency and number of circulating blood monocyte which are known to be modulated in hyperlipidemic mice. Can the authors pinpoint to candidate gene/protein regulated by PPAR α expressed by pro-inflammatory VECs involved in the control of monocyte recruitment? Such as chemokine, adhesion molecules.

Reviewer #3 (Remarks to the Author):

In this paper, the authors characterize the cellular content of valvular inflammation under hyperlipidemia. They confirm an association between increased plasma LDL levels and increased valvular lipids and macrophage inflammation. They performed scRNAseq analysis to study the cellular heterogeneity of aortic valves under hyperlipidemia in 2 mouse models: ApoE and LDLR knockout. They confirm that PPAR γ pathway activation reduces inflammation thus putting this pathway forward as a potential drug target for aortic valve disease.

The introduction gives a good and generalized background of the topic and clearly states the motivations to undergo this research. The methods are well developed and explained and provide enough details to allow reproducibility, they also comply with the requested standards in the field. The results are well presented, analyzed, and interpreted. However, the discussion lacks in-depth analysis.

In the current state, the results presented in this paper lack originality and do not bring a major contribution to the field without further investigation. Of note, the scRNAseq analysis is the most noteworthy information but it has not been taken deep enough to provide compelling and significant contributions to the field. Despite a lack of novelty, a good and reliable amount of work leading to convincing results has been performed. It would be unfortunate not to take advantage of the author's expertise to further decipher the mechanisms involved in this disease affecting a lot of individuals.

Additional comments:

- It would be instrumental in adding ApoE $^{-/-}$ and LDLR $^{-/-}$ without high-fat diet as controls.
- Regarding the monocyte migration assay, LDL and ox-LDL alone (without interstitial cells) should be tested.
- In supplementary Fig. 3, it is mentioned that Clec3b $^{+}$ is not located within the aortic valves. Does it mean that there is possible contamination by aortic cells?
- To strengthen the scRNAseq results and PPAR γ pathway implication (as mentioned in the discussion) specific deletion of PPAR γ in vascular endothelial cells is mandatory and would provide the novelty lacking in this paper.

Reviewer #4 (Remarks to the Author):

Manuscript NCOOMS-21-49026

Single-cell transcriptomics reveal cellular diversity in the aortic valve and immunomodulatory role of PPAR γ during hyperlipemia

The authors demonstrated that lipid accumulation and inflammation in the aortic valve is triggered by LDL in different hyperlipidemic mouse models (WD fed Ldlr $^{-/-}$ and Apoe $^{-/-}$). They performed a comprehensive scRNAseq analysis of diseased vs. normal aortic valves that revealed two main Mo/Ma

populations (monocyte-derived population with pro-inflammatory profile and resident Lyve+ with anti-inflammatory profile. They compared the kinetics of lipid deposits, inflammation, and monocyte/macrophage infiltration in hyperlipidemic mouse models with different genetic manipulation.

The manuscript is well written, and the study is very well designed. The introduction is concise and covers all the topics developed in the paper. However, the novelty of the study is not very clear. The authors describe the accumulation of lipids, in particular LDL, and macrophages in the aortic valve of well-studied and standardized mouse models of hyperlipidemia and atherosclerosis. Comparing two different genetic deletions (Ldl^{-/-} and Apoe^{-/-}) for the development of aortic valve disease might not provide sufficient insight regarding the mechanisms of the disease and how lipid-lowering drugs could be useful in preventing the initiation of the disease. It might be worthwhile to compare either WD-fed Apoe^{-/-} and Ldl^{-/-} with chow-diet fed Apoe^{-/-} and Ldlr^{-/-} mice to gain insight in the diet role in the development of the disease or compare WD-Apoe^{-/-} and Ldlr^{-/-} with WD-C67BL6 mice to elucidate genetic background role. The findings that the PPAR γ pathway is activated in hyperlipidemic mice and the corroboration of these findings in human samples make PPAR γ a possible target to be further evaluated for aortic valve disease. I recommend the authors restructure the figures to better emphasize these findings.

Minor comments

- 1) Figure 2 and 8 legends please change 'infected' for 'injected'.
- 2) Please add statistical analysis to each figure legend and the times that the independent experiment has been performed.
- 3) According to animal experiments section in M&M, the PCSK9-AAV injection was also performed in WT mice, please describe the findings in Figure 2.
- 4) Please provide a full gating strategy for flow cytometry experiments in Figure 3 and Figure 8.
- 5) Indicate time point in Fig 4 (i.e. 16 weeks)
- 6) Please quantify in situ hybridization experiment in Fig. 6E to confirm the increase in CD36⁺ VEC cells in diseased Ldlr^{-/-} valve compared to normal C57BL6 valves.
- 7) Please quantify in situ hybridization experiment in Fig. 7e to state "The valves from aortic hyperlipidemic mice showed much higher PPAR γ expression in nuclei, especially (in particular) in VECs, compared to normal valves (page 16).
- 8) In Fig. 7, the authors describe that PPAR γ pathway in VECs is activated by increased LDL plasma levels, although LDL and cholesterol levels are only increased (above the reference values for humans) in 2 patients. Could the authors comment on this correlation and the fact that PPAR γ ⁺ is highly increased in non-calcified vs. calcified lesion?

NCOMMS-21-49026

Point-by-point response

Reviewer #1

This manuscript describes the regulation of valvular disease in mice models. Using sophisticated analysis of cellular transcriptomes, the authors report that a detailed analysis of changes in compositions of the valve during disease onset. Among the regulated genes, the PPAR γ pathway was predominantly regulated in one cluster of VEC of diseased mice. While inhibition of PPAR γ further increased valve inflammation, pharmacological activation of PPAR γ reduced valve disease. The study comprises extensive data and is generally well performed. However, the findings are not very surprising and the role of PPARs in valve disease has been previously reported (see review www.ncbi.nlm.nih.gov/pmc/articles/PMC6719701/).

- Thank you so much for your valuable comments. As you pointed out, our study is in line with previous reports that support the role of PPAR γ in aortic valve disease. However, previous reports have dealt with late-stage aortic valve disease (involving calcification). In contrast, our study deals with the early stage of aortic valve disease. We showed a protective role of PPAR γ in the early stage of aortic valve disease, via scRNA-seq. In the revised manuscript, we further suggested that the anti-inflammatory effect of PPAR γ is also exerted in a VEC-specific manner, by RNA-seq analysis, using *PPARG* gene silencing of human aortic VEC (**Figure 7f-g**) [page 19, lines 385-392]. We believe that the improvement in our study's content will address the issue of novelty, and hope our revised manuscript and the point-by-point responses will fulfill your expectations.

Fig. 7f-g. Upregulation of Pro-inflammatory genes and cell adhesion molecules in the human VEC with *PPARG* knockdown.

f. Pro-inflammatory (top) and cell adhesion molecule (bottom) scores in non-targeting siRNA (NC) and *PPARG* targeting siRNA-treated (*PPARG* knockdown, KD) human VECs under no (NT) and oxLDL treatment conditions. Each score represents the average expression level of genes, as shown in Fig. 7g. p, two-tailed T-test p-value. **g.** Expression map of pro-inflammatory genes (top) and cell adhesion molecules (bottom).

Specific comments

1. Previous studies showed a down-regulation of CD36 in diseased human aortic valves (DOI:<https://doi.org/10.1016/j.atherosclerosis.2014.05.933>). In Figure 6e, the authors showed a representative image of increased CD36 expression. However, these data are not quantified. An analysis of CD36 in early and later stages of the disease should be done to determine if the induction is indeed an early protective mechanism, which may be lost at later stages of the disease.

- We appreciate your valuable comment. As requested, we have added the quantification of *Cd36* *in situ* hybridization (**Figure 6f**). Owing to difficulties in obtaining early stenosis samples in humans, or late samples in mice, directly comparing early and late stenosis data is difficult. We would like to emphasize that Syvaranta et al. reported two observations of *CD36* expression¹. First, valve tissues expressed lower levels of *CD36* in stenosis than in controls. Second, myofibroblasts isolated from late-stage stenosis expressed higher levels of *CD36* in culture than in those isolated from controls. We also checked the previously reported scRNA-seq data by Xu et al.², and found that the calcified group expressed higher levels of *CD36* than the non-calcified group (**Extra figure 1**). Unlike previous reports, our experiments focused on early-stage disease during which lipid accumulation and inflammation mainly occur. We compared the aortic valves of hyperlipidemic mice with those of normal mice, and found that *CD36* was upregulated in aortic valve hyperlipidemic conditions (**Figure 6f**). Upregulated *CD36* may play a role in valvular lipid accumulation in early-stage disease, supported by the decrease in LDL accumulation in the aortic valve by *CD36* antagonist (salvianolic acid B; SAB) treatment (**Figure 1j**). Overall, these observations suggest that different stimuli can cause complex effects on *CD36* expression. It is likely that cell populations in early- and late-stage stenosis are exposed to different levels of external stimuli. Molecular alterations during the progression of stenosis require further investigation.

Part of Fig. 6f. Localization of *Cd36*⁺ VEC in the aortic valve with quantification.

Identification of the localization of *Cd36*⁺ VEC subclusters using RNA in situ hybridization. The graph indicates the quantification of the *in situ* hybridization using the *CD36* probe (n = 5). Scale bar: 50 μ m.

Extra fig. 1. Expression level of *CD36* in human aortic valve.

a. UMAP projections on single-cells derived from human aortic valve as in Fig. 7d (top), feature (bottom, left) and violin plot (bottom, right) of *CD36* expression in each cluster. **b.** Comparison of *CD36* expression in calcified versus non-calcified samples for aortic valvular cells (top, left), VICs and stromal cells (top, right), macrophages (bottom, left), and VECs (bottom, right). scRNA-seq data of Xu et al.² was used. p, two-tailed T-test p-value.

Fig. 1j. Flow cytometric analysis of DiI-lipoproteins (LDL or VLDL) uptake of aortic valves from C57BL/6J mice, cultured with/without BLT-1 (SR-BI inhibitor) or SAB (*CD36* inhibitor) (n = 4).

2. Please provide more controls for the single cell seq data sets (e.g. number of reads, mapped reads, genes/cell, UMI/cell, total genes per sample) to make sure that the quality is similar among all studied samples.

- The requested information is provided in **Supplementary Table 3** (“Summary of scRNA sequencing parameters”).

		C57BL/6J	Apoe -/-	Ldlr -/-
Sequencing	Number of Reads	46,539,591	151,825,684	181,607,997
	Q30 Bases in Barcode	96.2%	97.5%	97.5%
	Q30 Bases in RNA Read	84.9%	89.0%	89.1%
	Q30 Bases in Sample Index	94.7%	96.3%	95.3%
	Q30 Bases in UMI	96.3%	97.6%	97.6%
Mapping	Reads Mapped to Genome	95.7%	96.8%	96.8%
	Reads Mapped Confidently to Genome	92.2%	92.2%	91.9%
	Reads Mapped Confidently to Intergenic Regions	2.6%	2.1%	2.0%
	Reads Mapped Confidently to Intronic Regions	13.8%	9.4%	9.1%
	Reads Mapped Confidently to Exonic Regions	75.8%	80.8%	80.8%
	Reads Mapped Confidently to Transcriptome	73.6%	78.6%	78.7%
	Reads Mapped Antisense to Gene	0.9%	0.7%	0.7%
Cells	Mean Reads per Cell	14,575	94,477	85,502
	Median UMI Counts per Cell	3,351	6,708	5,514
	Median Genes per Cell	1,380	2,114	1,760
	Total Genes Detected	16,933	16,822	16,705
	Estimated Number of Cells	3,193	1,607	2,124
	Number of Quality Control Pass Cells	3,109	1,477	1,988

Supplementary Table 3. Summary of scRNA sequencing parameters.

3. Mean values and group comparisons should be provided for Suppl. Table 1.

- We have provided mean \pm SD values and group comparisons in **Supplementary**

Table 1.

Group comparisons on (p-value)	Calcified						Non-calcified						Group												
	Mean \pm SD	#1	#2	#3	#4	#5	Mean \pm SD	#1	#2	#3	#4	#5	#6	#7	ID	Age	Sex	Height (cm)	Weight (kg)	Comorbidity		Serum lipid profiles (mg / dL)			
																				Hypertension	Diabetes mellitus	Total cholesterol	Triglyceride	HDL	LDL
0.043	74.4 \pm 8.591	61	75	85	75	76	63.86 \pm 8.295	52	53	64	71	73	67	67	M	M	M	M	M	Yes	No	208	247	38	122
N/A	N/A	F	F	M	M	M	N/A	M	M	M	M	M	M	M	F	M	M	M	M	Yes	No	126	153	45	90
0.508	160 \pm 2.924	156.8	157.4	163.9	161	161	163.3 \pm 8.098	170	176	157	166	162	152	160	M	M	M	M	M	No	Yes	132	73	85	91
0.639	62.92 \pm 2.858	60.3	67.5	60.8	63.4	62.6	67.5 \pm 11.63	69.4	91.9	59.9	64.6	67.9	62.1	56.7	M	M	M	M	M	Yes	No	118	220	40	63
N/A	N/A	Yes	Yes	Yes	Yes	Yes	N/A	No	Yes	No	Yes	Yes	Yes	Yes	No	No	No	No	No	Yes	No	208	247	38	122
N/A	N/A	No	No	No	No	No	N/A	No	Yes	No	Yes	Yes	Yes	Yes	No	No	No	No	No	Yes	No	126	153	45	90
0.755	150.8 \pm 12.28	148	138	171	146	151	165.3 \pm 61.69	105	118	132	195	126	273	208	M	M	M	M	M	No	Yes	195	73	85	91
0.876	144.8 \pm 84.83	277	96	78	183	90	169.1 \pm 98.95	56	220	73	153	110	325	247	M	M	M	M	M	Yes	No	195	73	85	91
0.845	44.8 \pm 9.757	33	58	51	41	41	50 \pm 9.6	25	40	85	63	54	54	38	M	M	M	M	M	Yes	No	132	73	85	91
> 0.999	86.2 \pm 19.24	70	64	102	87	108	89.29 \pm 39.63	75	63	91	90	31	153	122	M	M	M	M	M	Yes	No	132	73	85	91

Supplementary Table 1. Patient clinical information of human sample used in histological analysis.

Reviewer #2

The study by Lee SH et al., deciphers the cellular mechanisms underlying aortic valvular inflammation, the initial process of aortic valve diseases, in hyperlipidemic mouse models. First, the authors showed an association between LDL and increased valvular inflammation in *Apoe*^{-/-} and *Ldlr*^{-/-} mice. Next, single cell RNA sequencing analysis of aortic valves at steady state and under hyperlipidemic conditions revealed a marked cell heterogeneity among valvular interstitial cells (VIC), valvular endothelial cells (VEC) and leukocytes. Moreover, the authors showed that VECs control the recruitment of monocyte-derived macrophage through a PPAR γ dependent pathway. Based on these findings, the authors propose PPAR γ as a target to control valvular inflammation in the pre-calcified stage of aortic valve disease.

The study is well designed and comprehensive. It provides some mechanistic insight in aortic valvular inflammation and thus improves our knowledge on the pathophysiology of aortic valve disease and may offer potential therapeutics target. Nevertheless, some points should be addressed to further strengthen the conclusions of this study.

- Thank you for your helpful review of our work. Thanks to your crucial and detailed comments, we performed additional experiments and revised the manuscript accordingly. We have attempted to answer as many points as possible, and hope our revised manuscript and the point-by-point responses will fulfill your expectations.

1-1. In figure 1, the authors analyzed lipid accumulation in blood and aortic valves in WT, *Apoe*^{-/-} and *Ldlr*^{-/-} mice. Despite similar plasma levels of total cholesterol and LDL in *Apoe*^{-/-} and *Ldlr*^{-/-}, *Ldlr*^{-/-} showed increased accumulation of lipids and LDL in aortic valve. What about macrophage accumulation, is it also different in aortic valve from *Apoe*^{-/-} and *Ldlr*^{-/-} mice? This should be addressed since the authors use this parameter to evaluate inflammation.

- We appreciate the reviewer's valuable comments. Our previous Figure 3e-f of the initially submitted manuscript (the figure comparing immune cells with subsets among C57BL/6J, *Apoe*^{-/-} and *Ldlr*^{-/-}) was insufficient for proper comparisons because of the lack of suitable control groups for diet. Therefore, we performed new flow cytometry analyses of aortic valve cells using six different groups: three different mouse models (C57BL/6J mice,

Apo^e^{-/-} mice, and *Ldlr^{-/-}* mice) with two different dietary conditions (normal chow diet [chow] or western diet [WD]). We compared the chow diet versus the WD groups in terms of the percentage of leukocytes, macrophages, and macrophage subsets in single cells, for each set of mouse models (**Figure 2a-c**). Among them, *Apo^e^{-/-}* and *Ldlr^{-/-}* mice showed an increased percentage of macrophages (particularly MHC-II^{hi} macrophages) when fed with WD compared to chow, whereas C57BL/6J mice showed no difference when fed with WD versus chow. We also observed that *Ldlr^{-/-}* mice showed a higher fold-change in the percentage of leukocytes, macrophages, and MHC-II^{hi} macrophages, where the percentage of MHC-II^{hi} macrophages positively correlated with LDL levels in WD-fed hyperlipidemic mice (*Apo^e^{-/-}* WD and *Ldlr^{-/-}* WD). We have replaced previous Figure 3e-f with these new data, and moved this information into **Figure 2a-c** and **Supplementary figure 5a-c** (for a more concise manuscript storyline) [page 10, lines 171-187].

Fig. 2a-c. Flow cytometry analysis of aortic valves of C57BL/6J, *Apo^e^{-/-}*, and *Ldlr^{-/-}* mice fed a chow diet (chow) or western diet (WD) for 10 weeks (n = 5).

a. Representative plot of leukocytes. **b.** Percentage of leukocytes, macrophages, and macrophage subsets. **c.** Correlations between percentage of MHC-II^{hi} CD11c⁺ CD206⁻ macrophages and blood lipid profiles of WD-fed hyperlipidemic mice.

Supplementary Fig. 5a-c. The relationship between lipid profiles and proportional changes in aortic valvular cells.

a. Percentage of VECs, VICs, and DCs in aortic valvular cells of chow-fed or WD-fed (for 10 weeks) C57BL/6J, *Apoe*^{-/-}, and *Ldlr*^{-/-} mice. **b.** Total cholesterol and LDL levels in the blood plasma of chow-fed or WD-fed (for 10 weeks) C57BL/6J, *Apoe*^{-/-}, and *Ldlr*^{-/-} mice. **c.** Correlations of total cholesterol and LDL levels with the percentage of leukocytes and macrophages in aortic valvular cells of WD-fed (for 10 weeks) hyperlipidemic mice (*Apoe*^{-/-} and *Ldlr*^{-/-}).

1-2. Explanation for the difference in LDL accumulation in aortic valves between *Apoe*^{-/-} and *Ldlr*^{-/-} mice should be provided. Is this decreased in lipid accumulation in aortic valve from *Apoe*^{-/-} mice explained by differences in expression of LDL scavenger receptors.

- We appreciate the reviewer's point of the possibility of intrinsic differences between *Apoe*^{-/-} and *Ldlr*^{-/-} mice, and we think this observation creates a good opportunity for a more precise interpretation of the data we have presented. Thus, we have added newly generated bulk tissue RNA-seq data using aortic valves of C57BL/6J (WT) mice, *Apoe*^{-/-} & *Ldlr*^{-/-} mice

fed with chow, and *Apoe*^{-/-} & *Ldlr*^{-/-} mice fed with WD (GSE205587, token: wfknwwoojxkjpax). We then compared the expression of overall scavenger receptors and each of the LDL scavenger receptors: *Cd36*, *Msr1*, *Olr1*, and *Scarb1* (**Supplementary Figure 3**). No significant difference in overall and LDL scavenger receptor expression was found regardless of whether WT, *Apoe*^{-/-} or *Ldlr*^{-/-} mice were involved or whether the diet was chow or WD. Thus, we concluded that blood LDL cholesterol levels cause a difference in LDL accumulation in the aortic valves between *Apoe*^{-/-} and *Ldlr*^{-/-} mice (**Figure 1a-d**). We have presented this new data in **Supplementary Figure 3** and written the related text in the Results [page 9, lines 143-149] and the Methods [page 35, lines 750-760], [page 37, lines 792-797].

Supplementary Fig. 3. Expression levels of scavenger receptors in the aortic valve tissue of C57BL/6J, *Apoe*^{-/-} and *Ldlr*^{-/-} mice.

a. Boxplot of the average expression levels of scavenger receptor genes in the aortic valve tissue of C57BL/6J, *Apoe*^{-/-}, and *Ldlr*^{-/-} mice (*Ager*, *Cd14*, *Cd163*, *Cd207*, *Cd36*, *Cd68*, *Clec7a*, *Colec12*, *Cxcl16*, *Ly75*, *Marco*, *Megf10*, *Mrc1*, *Msr1*, *Olr1*, *Scara3*, *Scara5*, *Scarb1*, *Scarb2*, *Scarf1*, *Scarf2*, *Ssc4d*, *Ssc5d*, *Stab1*, *Stab2*). **b.** Boxplots of the expression levels of each LDL/oxLDL scavenger

receptor genes (*Msr1*, *Scarb1*, *Olr1*, and *Cd36*). Chow: chow diet-fed, WD: western diet-fed (for 8 weeks). p, two-tailed T-test p-value.

Fig. 1a-d. Comparison of the lipid accumulation in the aortic valves and blood lipid profiles.

Mice (*C57BL/6J*, *Apoe*^{-/-}, and *Ldlr*^{-/-}) fed a chow diet (chow) versus a western diet (WD) for 10 weeks (n = 5). **a.** Representative lipid stain images and measurement of valvular lipid deposition. Scale bar: 150 μm (left), 30 μm (right). **b.** Total cholesterol and LDL levels in the blood plasma. **c.** Correlations between valvular lipid deposition and blood lipid profiles of WD-fed hyperlipidemic mice. **d.** Correlation between aortic valvular lipid accumulation and aortic sinus lesion in WD-fed hyperlipidemic mice.

1-3. It is not clear which mice were used in Fig1h and j, the information is missing from the result and figure legend.

- We have added information about the mice (*C57BL/6J*) in the legends of Figure 1h-j, and their corresponding Results section [pages 9-10, lines 162-169].

2. The authors showed that increasing plasma LDL levels in *Apoe*^{-/-} mice enhanced lipid and macrophage accumulation in aortic valves. To further support this causal relationship between lipid accumulation and aortic valvular inflammation, the authors

should consider analyzing the effect of reducing LDL levels in *Ldlr*^{-/-} mice on aortic valve inflammation.

- We performed a lipid-lowering experiment using ezetimibe (Ezetrol, MSD) in *Ldlr*^{-/-} mice. For 10 weeks, we fed the experimental *Ldlr*^{-/-} group with a WD that contained 0.015% ezetimibe, and the control *Ldlr*^{-/-} group with ordinary WD. We then analyzed blood lipid profiles (total cholesterol and LDL), quantified valvular lipid accumulation, and performed a flow cytometric analysis of immune cells. Blood cholesterol-lowering by ezetimibe (Supplementary figure 5d) significantly decreased valvular lipid accumulation and alleviated inflammatory parameters (such as the percentage of leukocytes, macrophages, and MHC-II macrophages; Figure 2j-m). We have described these Results [page 12, lines 212-224], and the related Methods [page 27, lines 560-562], in the revised manuscript.

Supplementary Fig. 5d. Total cholesterol and LDL levels in the blood plasma of normal WD-fed or ezetimibe containing WD-fed (for 10 weeks) *Ldlr*^{-/-} mice.

Fig. 2j-m. Effects of lipid-lowering by ezetimibe on lipid accumulation and inflammation in aortic valves.

j, k. Lipid-lowering effects of ezetimibe on lipid accumulation in the aortic valves of WD-fed (for 10 weeks) *Ldlr*^{-/-} mice (n = 4). Representative lipid stain images. Scale bar: 150 μm (top), 20 μm

(bottom). **(j)** and quantification of valvular lipid deposition with blood lipid profiles. **(k)**. l, m. Effects of lipid-lowering by ezetimibe on the proportion of immune cells in aortic valves of WD-fed (for 10 weeks) *Ldlr*^{-/-} mice, through flow cytometry (n = 5). Representative plot of leukocytes **(l)** and percentages of each cell population in single cells **(m)**.

3-1. Single cell RNA sequencing analysis revealed heterogeneity in VIC, VEC and leukocytes. VICs show pro-inflammatory features. What about the VECs?

- We plotted the pro-inflammatory and anti-inflammatory gene expression with a heatmap and boxplot (**Figure 6d**), and found that hyperlipidemic mouse models (*Apoe*^{-/-} and *Ldlr*^{-/-}) experienced more inflammation than did the chow diet controls (C57BL/6J). In addition, gene set enrichment analysis (**Figure 6e**) indicated the enriched monocyte chemotaxis in the hyperlipidemic models (*Apoe*^{-/-} and *Ldlr*^{-/-} mice) than in the control model (C57BL/6J mice). Monocyte chemotaxis is representative of the inflammatory change of tissues. In summary, VECs also showed pro-inflammatory changes during hyperlipidemia [pages 16-17, lines 324-327].

Fig. 6d-e. Pro-Inflammatory features of VECs in aortic valves of hyperlipidemic mice

d. Expression of proinflammatory genes in VECs. Heatmap (left) and boxplot (right) of average expression level of genes listed in the heatmap. Heatmap are displayed as expression values scaled by z-transformation on a scale of at least -2.5. p, two-tailed T-test p-value. **e.** Enrichment plot for significant Gene Ontology (GO) terms related to monocyte chemotaxis. Genes were ranked by the fold changes between knockout and wild type models for cells in VEC clusters (VEC_C0, C1, and C2).

3-2. Among the VEC subpopulations, one express Prox-1. Are these cells lymphatic endothelial cells?

- VEC_C2 (*Prox1*⁺ EC) cells were localized in the endothelial lining of the fibrosa, and they did not form structures within lymphatic vessels (**Figure 6f**). Except for the localization displayed in Figure 6f, none of the other regions had PROX1⁺ VECs. Lymphatic endothelial cells expressed *Lyve1* as a marker gene^{3, 4}, but *Lyve1* was not expressed in VEC_C2 (*Prox1*⁺ EC) cells (**Extra figure 2**). In this regard, we conclude that VEC_C2 (*Prox1*⁺ EC) cells are not a type of lymphatic EC.

Part of Fig. 6f. Localization of PROX1⁺ VEC in aortic side of the aortic valve. Scale bar: 50 µm.

Extra fig. 2. Expression of *Lyve1* and *Prox1* in VECs.

UMAP projections on VECs as in Fig. 6a (top) and feature plot of *Lyve1* (bottom, left) and of *Prox1* (bottom, right) color-coded by expression (gray to blue). Note that VEC_C2 (*Prox1*⁺ EC) does not express *Lyve1*.

4. Single cell RNA sequencing analysis also reveal the activation of PPAR-g pathway in VECs. Is this also observed in VICs and macrophages?

- We examined the level of PPAR γ pathway activation in the VECs, VICs, and macrophages via scRNA-seq, and analyzed the expression of PPAR γ target genes (gene list from PPARgene database⁵; **Supplementary Figure 12c-d**). We found that the expression levels of PPAR γ target genes in VICs and macrophages were much lower than those in VECs in all mouse models (C57BL/6J, *Apoe*^{-/-}, *Ldlr*^{-/-} mice; **Supplementary Figure 12c-d**) [pages 18-19, lines 370-375].

Supplementary Fig. 12c-d. Analysis of PPAR γ target genes.

c. Average expression level of 49 PPAR γ target genes (<http://www.ppargene.org/>) in VECs (VEC_C0, C1, and C2), VICs, and macrophages for each mouse model. **d.** Relative expression levels of PPAR γ target genes in VECs (VEC_C0, C1, and C2), VICs, and macrophages of *Ldlr*^{-/-} and *Apoe*^{-/-} mice, compared with C57BL/6J mice, are shown as log fold changes.

5-1. The staining of PPAR γ is not obvious and costaining with a marker of VECs should be performed in Fig 7e. For this experiment, the authors used section from *Apoe*^{-/-} which showed less lipid accumulation and inflammation. What about in *Ldlr*^{-/-} mice?

- We replaced the previous DAB-based IHC data from Figure 7e with the new immunofluorescence data of C57BL/6J, *Apoe*^{-/-}, and *Ldlr*^{-/-} mice, co-stained with PPAR γ (red), endomucin (EC marker, EMCN, green), and DAPI nuclei stain (**Figure 7c** and **Supplementary Figure 13**). We also quantified nuclear PPAR γ signals, per EC (**Figure 7c**) [page 19, lines 375-378].

Fig. 7c

Supplementary Fig. 13

Fig. 7c. and Supplementary Fig. 13. Immunostaining of PPAR γ in mouse aortic valve.

Immunostaining of PPAR γ (red) and endomucin (EMCN, EC marker, green) in aortic valve with sinus from normal (chow diet) and hyperlipidemic mice (*Apoe*^{-/-} and *Ldlr*^{-/-} mice, WD for 16 weeks) (n = 4). DAPI (blue) was used to stain nuclei. The graph represents the mean fluorescence intensity (MFI) of PPAR γ in VECs. Kruskal-Wallis test with post-hoc Dunn's test was used. Scale bar: 30 μ m.

5-2. To confirm the activation of PPAR γ in VECs, the cytoplasmic and nuclear cellular localization of PPAR γ should be analyzed and/or the analysis of PPAR γ targeted genes should be examined. This should also be performed for VIC and macrophages since this pathway is also activated in these cells based on the single cell RNA sequencing.

- To confirm the activation of PPAR γ in VECs, we analyzed the relative expression of PPAR γ targeted genes (the gene list from PPARgene database)⁵ in the VECs, VICs, and macrophages of C57BL/6J, *Apoe*^{-/-} and *Ldlr*^{-/-} mice, using scRNA-seq (**Supplementary Figure 12c-d**). The expression levels of PPARG target genes were higher in VECs than in VICs and macrophages (**Supplementary Figure 12c-d**). The relative expression map showed that the expression of PPARG-targeted genes was much higher in the VECs of *Apoe*^{-/-} and *Ldlr*^{-/-} mice than in the VECs of C57BL/6J mice (**Supplementary Figure 12c-d**) [pages 18-19, lines 370-375].

Supplementary Fig. 12c-d. Analysis of PPAR γ target genes.

c. Average expression level of 49 PPAR γ target genes (<http://www.ppargene.org/>) in VECs (VEC_C0, C1, and C2), VICs, and macrophages for each mouse model. **d.** Relative expression levels of PPAR γ target genes in VECs (VEC_C0, C1, and C2), VICs, and macrophages of *Ldlr*^{-/-} and *Apoe*^{-/-} mice, compared with C57BL/6J mice, are shown as log fold changes.

6-1. The authors propose that PPAR γ pathway regulates the recruitment of monocyte into the valve. Do the activation or inhibition of PPAR γ also affect the frequency and number of circulating blood monocyte which are known to be modulated in hyperlipidemic mice.

- We fed pioglitazone-WD (experimental group) or normal WD (control group) to C57BL/6J mice injected with PCSK9-AAV, using the same method as that shown in Figure 8d. We compared the number of blood monocytes and subsets (Ly6C-high/low), for 6 weeks, by flow cytometric analyses once a week (**Supplementary Figure 15c-d**). No difference in the number of monocytes or subsets was found at each of the checkpoints. We present these data in **Supplementary Figure 15c-d** and related texts on [page 20, lines 416-417], [page 28, lines 573-576].

Supplementary Fig. 15c-d. Effects of pioglitazone in the number of circulating monocytes.

Flow cytometry analysis of circulating blood monocytes, for 6 weeks. PCSK9-AAV-injected C57BL/6J mice were fed with pioglitazone-containing WD or normal WD for 6 weeks, and the number of monocytes and subsets in 10 μ L of blood was examined once a week ($n = 5$). **c.** The gating strategy of blood monocytes and subsets. **d.** Cell counts of monocytes, Ly6C high monocytes, and Ly6C low monocyte in 10 μ L of blood, from week 1 to 6.

6-2. Can the authors pinpoint to candidate gene/protein regulated by PPAR γ expressed by pro-inflammatory VECs involved in the control of monocyte recruitment? Such as chemokine, adhesion molecules.

- We found that knockdown of *PPARG* upregulated *CXCL1*, *CCL2*, *CXCL16*, *IL6*, *ICAM1*, and *ICAM2* expression in oxLDL-treated human VECs (Figure 7f-g) [page 19, lines 385-392]. We pinpoint these genes in Figure 8e.

Fig. 7f-g. Upregulation of Pro-inflammatory genes and cell adhesion molecules in the human VEC with *PPARG* knockdown.

f. Pro-inflammatory (top) and cell adhesion molecule (bottom) scores in non-targeting siRNA (NC) and *PPARG* targeting siRNA-treated (*PPARG* knockdown, KD) human VECs under no (NT) and oxLDL treatment conditions. Each score represents the average expression level of genes, as shown in Fig. 7g. p, two-tailed T-test p-value. **g.** Expression map of pro-inflammatory genes (top) and cell adhesion molecules (bottom).

Fig. 8e. Proposed pathogenesis model of the early-stage aortic valve disease induced by hyperlipidemia.

In hyperlipidemic states, oxidized LDL triggers aortic valvular inflammation by enhancing the production of various cytokines and chemokines, leading to the recruitment of monocyte-derived MHC-II^{hi} macrophages. Meanwhile, PPAR γ activation during hyperlipidemia inhibits the accumulation of monocytes and macrophages in the aortic valve. Top view (left, top), side view (left, bottom), and legends (right).

Reviewer #3

In this paper, the authors characterize the cellular content of valvular inflammation under hyperlipidemia. They confirm an association between increased plasma LDL levels and increased valvular lipids and macrophage inflammation. They performed scRNAseq analysis to study the cellular heterogeneity of aortic valves under hyperlipidemia in 2 mouse models: ApoE and LDLR knockout. They confirm that PPAR γ pathway activation reduces inflammation thus putting this pathway forward as a potential drug target for aortic valve disease.

The introduction gives a good and generalized background of the topic and clearly states the motivations to undergo this research. The methods are well developed and explained and provide enough details to allow reproducibility, they also comply with the requested standards in the field. The results are well presented, analyzed, and interpreted. However, the discussion lacks in-depth analysis.

In the current state, the results presented in this paper lack originality and do not bring a major contribution to the field without further investigation. Of note, the scRNAseq analysis is the most noteworthy information but it has not been taken deep enough to provide compelling and significant contributions to the field. Despite a lack of novelty, a good and reliable amount of work leading to convincing results has been performed. It would be unfortunate not to take advantage of the author's expertise to further decipher the mechanisms involved in this disease affecting a lot of individuals.

- Thank you so much for your valuable comments. In this study, we present the first scRNA-seq analysis of the inflammatory aortic valve under hyperlipidemia. Taking a clue from our scRNA-seq data on the PPAR γ , we performed additional knockdown experiments showing the anti-inflammatory role of PPAR γ in VEC-specific manners (revised **Figure 7f-g**). With the captivating results, we further discuss the implications, especially from the therapeutic angle [**pages 22-24, lines 450-492**]. We believe the revision provides the novelty and in-depth discussion that you requested, and hope our revised manuscript and the point-by-point responses will fulfill your expectations.

.

Additional comments:

1. It would be instrumental in adding ApoE^{-/-} and LDLR^{-/-} without high-fat diet as controls.

- Thank you for your suggestion. For a more precise comparison, we examined valvular lipid accumulation and performed flow cytometric analyses of leukocytes and their associated subsets, using six different groups: three distinct mice (C57BL/6J, ApoE^{-/-} and Ldlr^{-/-}), along with two dietary conditions (normal chow versus western diet). Similar to the previous Figures 1a-d (Supplementary Figure 2 in the revised version) and previous Figures 3e-f, when comparing WD-fed to chow-fed mice, Ldlr^{-/-} mice showed a greater increase in lipid accumulation (Figure 1a-d) [page 8, lines 119-137], and immune cell proportions (leukocytes, macrophages, and MHC-II^{hi} macrophages) than did ApoE^{-/-} mice (Figure 2a-c and Supplementary Figure 5a-c) [page 10, lines 171-187]. In contrast, C57BL/6J mice showed no difference in lipid accumulation and immune cell proportions when fed with WD versus chow (Figure 1a-d and Figure 2a-c). These results may be derived from differences in the blood lipid profiles of mice. Unlike that which was observed in hyperlipidemic models (ApoE^{-/-} and Ldlr^{-/-} mice), feeding WD to wild-type (C57BL/6J) mice did not change blood total cholesterol and LDL levels (Figure 1b and Supplementary Figure 5b). We have replaced the previous Figure 3e-f with these new data and moved to Figure 2a-c for a better description of the results.

Fig. 1a-d. Comparison of the lipid accumulation in the aortic valves and blood lipid profiles.

Mice (*C57BL/6J*, *Apoe*^{-/-}, and *Ldlr*^{-/-}) fed a chow diet (chow) versus a western diet (WD) for 10 weeks (n = 5). **a.** Representative lipid stain images and measurement of valvular lipid deposition. Scale bar: 150 μ m (left), 30 μ m (right). **b.** Total cholesterol and LDL levels in the blood plasma. **c.** Correlations between valvular lipid deposition and blood lipid profiles of WD-fed hyperlipidemic mice. **d.** Correlation between aortic valvular lipid accumulation and aortic sinus lesion in WD-fed hyperlipidemic mice.

Fig. 2a-c. Flow cytometry analysis of aortic valves of *C57BL/6J*, *Apoe*^{-/-}, and *Ldlr*^{-/-} mice fed a chow diet (chow) or western diet (WD) for 10 weeks (n = 5).

a. Representative plot of leukocytes. **b.** Percentage of leukocytes, macrophages, and macrophage subsets. **c.** Correlations between percentage of MHC-II^{hi} CD11c⁺ CD206⁻ macrophages and blood lipid profiles of WD-fed hyperlipidemic mice.

Supplementary Fig. 5a-c. The relationship between lipid profiles and proportional changes in aortic valvular cells

a. Percentage of VECs, VICs, and DCs in aortic valvular cells of chow-fed or WD-fed (for 10 weeks) C57BL/6J, *Apoe*^{-/-}, and *Ldlr*^{-/-} mice. **b.** Total cholesterol and LDL levels in the blood plasma of chow-fed or WD-fed (for 10 weeks) C57BL/6J, *Apoe*^{-/-}, and *Ldlr*^{-/-} mice. **c.** Correlations of total cholesterol and LDL levels with the percentage of leukocytes and macrophages in aortic valvular cells of WD-fed (for 10 weeks) hyperlipidemic mice (*Apoe*^{-/-} and *Ldlr*^{-/-}).

2. Regarding the monocyte migration assay, LDL and ox-LDL alone (without interstitial cells) should be tested.

- We added data for non-treated, LDL, and oxLDL-alone (without VICs) treated mice to **Figure 5J**. These data were obtained simultaneously with pre-existing data (with VICs), within the same experimental batch, but were not included in the initially submitted manuscript.

Fig. 5j. Transwell migration assay for evaluating monocyte chemotactic levels of VICs cultured with/without LDL or oxLDL. (n = 4). Samples without VIC were used for control. Size of field: 1272.79 μm^2 . Scale bar: 150 μm .

3. In supplementary Fig. 3, it is mentioned that *Clec3b*⁺ is not located within the aortic valves. Does it mean that there is possible contamination by aortic cells?

- First, based on our anatomical criteria (**Supplementary Figure 1b**), we isolated the aortic valve leaflet only; therefore, the possibility of aortic cell contamination was very low. If contamination had occurred, smooth muscle cells (SMCs), the main cellular components of the aortic wall, should have been present as much as VICs. However, in our scRNA-seq data, SMCs were not a major population. Nevertheless, we considered the previous Supplementary Figure 3b (image of RNA in situ hybridization of *Clec3b* and *Dpep1* in the myocardium) might be confusing, so we now presented the new image containing *Clec3b*⁺ *Dpep1*⁺ fibroblasts located in the vicinity of the hinge region (**Supplementary Figure 8b**), and changed the title of Supplementary Figure 8 to “Location of *Clec3b*⁺ *Dpep1*⁺ fibroblasts”.

Supplementary Fig. 1b. Criteria of anatomy for aortic valve prepared for scRNA-seq and flow cytometric analysis.

Aortic valves were carefully isolated from aortic sinus without contamination of other tissues. Whole-mount brightfield (left) and H&E images (right) of the isolated aortic valve. Scale bar: 100 μm (left), 50 μm (right).

Supplementary Fig. 8b. Location of *Clec3b*⁺ *Dpep1*⁺ fibroblasts.

Representative image of RNA in situ hybridization of marker genes (*Clec3b*, *Dpep1*) of Cluster_3 (*Clec3b*⁺ *Dpep1*⁺ fibroblasts). Scale bar: 50 μm (left), 10 μm (right). Cluster_3 cells (*Clec3b*⁺ *Dpep1*⁺ fibroblasts) were mainly located on the connective tissue adjacent to the hinge of aortic valve. Data are representative of at least three independent experiments.

4. To strengthen the scRNAseq results and PPAR γ pathway implication (as mentioned in the discussion) specific deletion of PPAR γ in vascular endothelial cells is mandatory and would provide the novelty lacking in this paper.

- We appreciate your helpful comment. To reinforce our hypothesis, we performed new bulk RNA-seq using human aortic valvular endothelial cells that were isolated from patients who underwent aortic valve replacement and had been cultured with oxLDL treatment and siRNA-based *PPARG* knockdown (**Figure 7f-g, Supplementary Figure 14, and Supplementary Data 4**). Upon treatment with oxLDL, knockdown of *PPARG* in VECs induced upregulation of pro-inflammatory genes, such as *CXCL1*, *CCL2*, *CXCL16*, and *IL6*,

and cell adhesion molecules, such as *ICAM1* and *ICAM2* (Figure 7f-g). We have added this data to Figure 7f-g, Supplementary Figure 14, Supplementary Data 4, and related text to the Results [page 19, lines 385-392], and the Methods [page 26, lines 534-539], [pages 35-37, lines 762-804], [pages 41-42, lines 902-910], [pages 42-43, lines 916-933]. As the reviewer suggested, the additional experiments strengthen our claim and the novelty of the study.

Fig. 7f-g. Upregulation of Pro-inflammatory genes and cell adhesion molecules in the human VEC with *PPARG* knockdown.

f. Pro-inflammatory (top) and cell adhesion molecule (bottom) scores in non-targeting siRNA (NC) and *PPARG* targeting siRNA-treated (*PPARG* knockdown, KD) human VECs under no (NT) and oxLDL treatment conditions. Each score represents the average expression level of genes, as shown in Fig. 7g. p, two-tailed T-test p-value. **g.** Expression map of pro-inflammatory genes (top) and cell adhesion molecules (bottom).

Supplementary Fig. 14. *PPARG* gene silencing mediated by siRNA in human aortic VECs.

a. Phase contrast microscopy image of cultured human aortic VECs at passage 3. Scale bar: 100 μ m. **b.** Immunocytochemistry of CD31 (green) in human aortic VECs. DAPI (blue) was used for nuclei staining. Scale bar: 50 μ m. **c.** *PPARG* gene knockdown by siRNA was validated by western blot (n = 3). The graph indicates the relative intensity of PPAR γ to GAPDH. si-NC: negative control siRNA, si-*PPARG*: siRNA targeting *PPARG*. Image data are representative of at least three independent experiments.

Reviewer #4

The authors demonstrated that lipid accumulation and inflammation in the aortic valve is triggered by LDL in different hyperlipidemic mouse models (WD fed *Ldlr*^{-/-} and *Apoe*^{-/-}). They performed a comprehensive scRNAseq analysis of diseased vs. normal aortic valves that revealed two main Mo/Ma populations (monocyte-derived population with pro-inflammatory profile and resident Lyve⁺ with anti-inflammatory profile. They compared the kinetics of lipid deposits, inflammation, and monocyte/macrophage infiltration in hyperlipidemic mouse models with different genetic manipulation.

The manuscript is well written, and the study is very well designed. The introduction is concise and covers all the topics developed in the paper. However, the novelty of the study is not very clear.

- Thank you so much for your valuable comments. In the revised manuscript, we have added new data, such as RNA-seq analysis using *PPARG* gene knockdown of human aortic VECs (Figure 7f-g). Therefore, we have further supported our findings and improved the premise of the manuscript. We believe that our revised manuscript highlights the novelty of the study, and hope our revised manuscript and the point-by-point responses will fulfill your expectations.

I-1: The authors describe the accumulation of lipids, in particular LDL, and macrophages in the aortic valve of well-studied and standardized mouse models of hyperlipidemia and atherosclerosis. Comparing two different genetic deletions (*Ldl*^{-/-} and *Apoe*^{-/-}) for the development of aortic valve disease might not provide sufficient insight regarding the mechanisms of the disease and how lipid-lowering drugs could be useful in preventing the initiation of the disease. It might be worthwhile to compare either WD-fed *Apoe*^{-/-} and *Ldl*^{-/-} with chow-diet fed *Apoe*^{-/-} and *Ldlr*^{-/-} mice to gain insight in the diet role in the development of the disease or compare WD-*Apoe*^{-/-} and *Ldlr*^{-/-} with WD-C67BL6 mice to elucidate genetic background role.

- We appreciate and fully agree with your valuable comment. To provide mechanistic insights, we examined lipid accumulation and flow cytometric analyses of leukocytes and their associated subsets, comparing western diet (WD)-fed C57BL/6J, *Apoe*^{-/-}, and *Ldlr*^{-/-} mice with chow diet-fed C57BL/6J, *Apoe*^{-/-}, and *Ldlr*^{-/-} mice (**Figure 1a-d** for lipid accumulation and **Figure 2a-c** for flow cytometry). The new data showed an increase in lipid accumulation and immune cell proportions in WD-fed *Apoe*^{-/-} and *Ldlr*^{-/-} mice, but not in C57BL/6J mice, compared to chow-fed mice (**Figure 1a-d**, **Figure 2a-c**, and **Supplementary Figure 5a-c**). In addition, the fold change in WD-fed mice versus chow-fed mice was much higher in *Ldlr*^{-/-} than in *Apoe*^{-/-} mice, which corresponds with our previous analysis that revealed that *Ldlr*^{-/-} mice had a higher lipid accumulation and larger portion of valvular immune cells (especially MHC-II^{hi} macrophages) than did *Apoe*^{-/-} mice among WD-fed mice. This observation may be caused by a higher blood LDL level in *Ldlr*^{-/-} mice than in *Apoe*^{-/-} mice (**Figure 1a-d**, **Figure 2a-c**). We have replaced the previous Figure 1a-c and previous Figure 3e-f with new data (**Figure 1a-d**, **Figure 2a-c**, and **Supplementary Figure 5a-c**), and described related texts [page 8, lines 119-137], [page 10, lines 171-187], in the revised manuscript. In addition, to identify the effect of lipid-lowering drug treatment, we fed *Ldlr*^{-/-} mice either with WD containing 0.015% ezetimibe (experimental group) or normal WD (control group), and analyzed blood lipid profiles (total cholesterol and LDL), valvular lipid accumulation, and the proportion of immune cells and their subsets via flow cytometry (**Figure 2j-m** and **Supplementary Figure 5d**). Lipid-lowering by ezetimibe significantly decreased lipid accumulation and the proportion of immune cells — particularly MHC-II^{hi} macrophages (**Figure 2j-m** and **Supplementary Figure 5d**) [page 12, lines 212-224], [page 27, lines 560-562]. Collectively, blood cholesterol level (especially LDL) is associated with the development of early aortic valve disease caused by hyperlipidemia, where lipid-lowering drugs, such as ezetimibe, may be effective in reducing disease progression.

Fig. 1a-d. Comparison of the lipid accumulation in the aortic valves and blood lipid profiles.

Mice (*C57BL/6J*, *Apoe*^{-/-}, and *Ldlr*^{-/-}) fed a chow diet (chow) versus a western diet (WD) for 10 weeks (n = 5). **a.** Representative lipid stain images and measurement of valvular lipid deposition. Scale bar: 150 μm (left), 30 μm (right). **b.** Total cholesterol and LDL levels in the blood plasma. **c.** Correlations between valvular lipid deposition and blood lipid profiles of WD-fed hyperlipidemic mice. **d.** Correlation between aortic valvular lipid accumulation and aortic sinus lesion in WD-fed hyperlipidemic mice.

Fig. 2a-c. Flow cytometry analysis of aortic valves of *C57BL/6J*, *Apoe*^{-/-}, and *Ldlr*^{-/-} mice fed a chow diet (chow) or western diet (WD) for 10 weeks (n = 5).

a. Representative plot of leukocytes. **b.** Percentage of leukocytes, macrophages, and macrophage subsets. **c.** Correlations between percentage of MHC-II^{hi} CD11c⁺ CD206⁻ macrophages and blood lipid profiles of WD-fed hyperlipidemic mice.

Supplementary Fig. 5. The relationship between lipid profiles and proportional changes in aortic valvular cells.

a. Percentage of VECs, VICs, and DCs in aortic valvular cells of chow-fed or WD-fed (for 10 weeks) C57BL/6J, *Apoe*^{-/-}, and *Ldlr*^{-/-} mice. **b.** Total cholesterol and LDL levels in the blood plasma of chow-fed or WD-fed (for 10 weeks) C57BL/6J, *Apoe*^{-/-}, and *Ldlr*^{-/-} mice. **c.** Correlations of total cholesterol and LDL levels with the percentage of leukocytes and macrophages in aortic valvular cells of WD-fed (for 10 weeks) hyperlipidemic mice (*Apoe*^{-/-} and *Ldlr*^{-/-}). **d.** Total cholesterol and LDL levels in the blood plasma of normal WD-fed or ezetimibe containing WD-fed (for 10 weeks) *Ldlr*^{-/-} mice.

Fig. 2j-m. Effects of lipid-lowering by ezetimibe on lipid accumulation and inflammation in aortic valves.

j, k. Lipid-lowering effects of ezetimibe on lipid accumulation in the aortic valves of WD-fed (for 10 weeks) *Ldlr*^{-/-} mice (n = 4). Representative lipid stain images. Scale bar: 150 μ m (top), 20 μ m (bottom). **(j)** and quantification of valvular lipid deposition with blood lipid profiles. **(k)**. **l, m.** Effects of lipid-lowering by ezetimibe on the proportion of immune cells in aortic valves of WD-fed (for 10 weeks) *Ldlr*^{-/-} mice, through flow cytometry (n = 5). Representative plot of leukocytes **(l)** and percentages of each cell population in single cells **(m)**.

I-2. The findings that the PPAR γ pathway is activated in hyperlipidemic mice and the corroboration of these findings in human samples make PPAR γ a possible target to be further evaluated for aortic valve disease. I recommend the authors restructure the figures to better emphasize these findings.

- Thank you for your suggestion. In the revised manuscript, we have added new data and reconstructed the figures to strengthen our statement of the protective role of PPAR γ in aortic VECs, in a hyperlipidemic condition (**Figure 7f-g, Supplementary Figure 14 and 12c-d**) [page 19, lines 385-392]. We believe that these newly included data will adequately highlight our main findings.

Fig. 7f-g. Upregulation of Pro-inflammatory genes and cell adhesion molecules in the human VEC with *PPARG* knockdown.

f. Pro-inflammatory (top) and cell adhesion molecule (bottom) scores in non-targeting siRNA (NC) and *PPARG* targeting siRNA-treated (*PPARG* knockdown, KD) human VECs under no (NT) and oxLDL treatment conditions. Each score represents the average expression level of genes, as shown in Fig. 7g. p, two-tailed T-test p-value. **g.** Expression map of pro-inflammatory genes (top) and cell adhesion molecules (bottom).

Supplementary Fig. 14. *PPARG* gene silencing mediated by siRNA in human aortic VECs.

a. Phase contrast microscopy image of cultured human aortic VECs at passage 3. Scale bar: 100 μ m. **b.** Immunocytochemistry of CD31 (green) in human aortic VECs. DAPI (blue) was used for nuclei staining. Scale bar: 50 μ m. **c.** *PPARG* gene knockdown by siRNA was validated by western blot (n = 3). The graph indicates the relative intensity of PPAR γ to GAPDH. si-NC: negative control siRNA, si-*PPARG*: siRNA targeting *PPARG*. Image data are representative of at least three independent experiments.

Supplementary Fig. 12c-d. Analysis of PPAR γ target genes.

c. Average expression level of 49 PPAR γ target genes (<http://www.ppargene.org/>) in VECs (VEC_C0, C1, and C2), VICs, and macrophages for each mouse model. **d.** Relative expression levels of PPAR γ target genes in VECs (VEC_C0, C1, and C2), VICs, and macrophages of *Ldlr*^{-/-} and *Apoe*^{-/-} mice, compared with C57BL/6J mice, are shown as log fold changes.

Minor comments

1) Figure 2 and 8 legends please change ‘infected’ for ‘injected’.

- We changed ‘infected’ into ‘injected’ in the legends of Figure 2 and 8.

2) Please add statistical analysis to each figure legend and the times that the independent experiment has been performed.

- We have added an explanation of the statistics as well as the information on the number of times for each independent experiment to each figure legend.

3) According to animal experiments section in M&M, the PCSK9-AAV injection was also performed in WT mice, please describe the findings in Figure 2.

- Confusion may occur from integration of the methods of two different experiments, using PCSK9-AAV in the “Mice” section. In the experiment in Figure 2d-i, PCSK9-AAV-injected *ApoE*^{-/-} mice were used in the experimental group, and non-injected *ApoE*^{-/-} mice were used in the control group. C57BL/6J (wild-type) mice were not used in this experiment. C57BL/6J (wildtype, *Ccr2*^{+/+}) mice were used for the experiment Figure 3f-g as the control group, compared with the experimental group which used *Ccr2*^{-/-} mice. In the revised manuscript, we have altered the descriptions to distinguish both [page 27, lines 553-555].

4) Please provide a full gating strategy for flow cytometry experiments in Figure 3 and Figure 8.

- We have provided the full gating strategy for the flow cytometry experiments outlined in **Supplementary Figure 4a**. The plots in Figures 3 and 8 seem slightly different, but we assure you that we used the same leukocyte gating strategy for Figures 3 and 8. The seeming differences may be because these two procedures formed part of different experiments; however, the compensation and voltage options were adjusted according to each experiment.

Supplementary Fig. 4a. Gating strategy for flow cytometry analysis of aortic valvular cells.

Full gating strategy of flow cytometry analysis of aortic valves. Alx488: Alexa Fluor 488. Alx647: Alexa Fluor 647. Mac: Macrophage.

5) Indicate time point in Fig 4 (i.e. 16 weeks)

- We have inscribed the time point in the legend of Figure 4.

6) Please quantify in situ hybridization experiment in Fig. 6E to confirm the increase in CD36+ VEC cells in diseased *Ldlr*^{-/-} valve compared to normal C57BL/6 valves.

- We have quantified RNA *in situ* hybridization of *Cd36* (Figure 6f).

Part of Fig. 6f. Localization of *Cd36*⁺ VEC in the aortic valve with quantification.

Identification of the localization of $Cd36^+$ VEC subclusters using RNA in situ hybridization. The graph indicates the quantification of the *in situ* hybridization using the $CD36$ probe (n = 5). Scale bar: 50 μ m.

7) Please quantify *in situ* hybridization experiment in Fig. 7e to state “The valves from aortic hyperlipidemic mice showed much higher PPAR γ expression in nuclei, especially (in particular) in VECs, compared to normal valves (page 16).

- We have replaced the previous immunohistochemistry data of Figure 7e with new immunofluorescence data of C57BL/6J, $Apoe^{-/-}$, and $Ldlr^{-/-}$ mice: PPAR γ (red) and endomucin (EC marker, EMCN, green) with DAPI (nuclei, blue) (Figure 7c and Supplementary Figure 13). We have also quantified the new data (Figure 7c) [page 19, lines 375-378].

Fig. 7c

Supplementary Fig. 13

Fig. 7c. and Supplementary Fig. 13. Immunostaining of PPAR γ in mouse aortic valve.

Immunostaining of PPAR γ (red) and endomucin (EMCN, EC marker, green) in aortic valve with sinus from normal (chow diet) and hyperlipidemic mice ($Apoe^{-/-}$ and $Ldlr^{-/-}$ mice, WD for 16 weeks) (n = 4). DAPI (blue) was used to stain nuclei. The graph represents the mean fluorescence intensity (MFI) of PPAR γ in VECs. Kruskal-Wallis test with post-hoc Dunn’s test was used. Scale bar: 30 μ m.

8) In Fig. 7, the authors describe that PPAR γ pathway in VECs is activated by increased LDL plasma levels, although LDL and cholesterol levels are only increased (above the reference values for humans) in 2 patients. Could the authors comment on this correlation and the fact that PPAR γ ⁺ is highly increased in non-calcified vs. calcified lesion?

- Thank you for your comments. As you mentioned, only two patients in the non-calcified group showed hypercholesterolemia (total cholesterol > 200 mg/dL; Supplementary Table 1). However, PPAR γ ⁺ VECs showed positive correlations with both total cholesterol and LDL levels (**Figure 7j**), which may be due to the influence of blood total cholesterol and LDL levels on PPAR γ activity — even under conditions of normal cholesterol levels. This may be explained by mouse aortic valve scRNA-seq analyses (**Supplementary Figure 12c**). Overall, VECs showed higher expression of PPAR γ target genes than did VICs or macrophages, not only in hyperlipidemic (*ApoE*^{-/-} and *Ldlr*^{-/-}, WD-fed) mice, but also in normal controls (C57BL/6J, chow-fed; **Supplementary Figure 12c**). It seems that, to some extent, PPAR γ is basally activated in VECs, and this activation may be enhanced as blood total cholesterol and LDL levels increase. We think this tendency seems to be conserved in humans.

In our IHC results, PPAR γ ⁺ valvular cells and PPAR γ ⁺ VECs were higher in non-calcified lesions than in calcified lesions (**Figure 7h-i**), and this effect was recapitulated by the expression level of genes in the PPAR γ regulon in the human aortic valve, observed via scRNA-seq (**Figure 7d-e**). During calcification, the structure and matrix of the aortic valve becomes irreversibly deformed. This deformation results in a significant change in the tissue microenvironment, which likely affects the loss of PPAR γ activity in valvular cells — including VECs. However, further studies are needed to verify this assumption.

Supplementary Fig. 12c. Analysis of PPAR γ target genes. Average expression level of 49 PPAR γ target genes (<http://www.ppargene.org/>) in VECs (VEC_C0, C1, and C2), VICs, and macrophages for each mouse model.

Fig. 7h-i. PPAR γ IHC in human aortic valves.

h. Representative image of PPAR γ IHC (top) and H&E stain (bottom). **i.** measurement of PPAR γ ⁺ cellular proportion in valvular cells (left) and VECs (right). Scale bar: 40 μ m (left), 20 μ m (right).

Fig. 7d-e. Expression level of genes of PPAR γ regulon in scRNA-seq of human aortic valves.

d. UMAP plot of 41,336 single-cells derived from human aortic valve, colored by the clusters (left) and samples (right). **e.** Average expression map of genes in PPAR γ regulon for each cell cluster from human aortic valve.

References

1. Syvaranta S, *et al.* Potential pathological roles for oxidized low-density lipoprotein and scavenger receptors SR-AI, CD36, and LOX-1 in aortic valve stenosis. *Atherosclerosis* **235**, 398-407 (2014).
2. Xu K, *et al.* Cell-Type Transcriptome Atlas of Human Aortic Valves Reveal Cell Heterogeneity and Endothelial to Mesenchymal Transition Involved in Calcific Aortic Valve Disease. *Arterioscler Thromb Vasc Biol* **40**, 2910-2921 (2020).
3. Banerji S, *et al.* LYVE-1, a new homologue of the CD44 glycoprotein, is a lymph-specific receptor for hyaluronan. *J Cell Biol* **144**, 789-801 (1999).
4. Jackson DG, Prevo R, Clasper S, Banerji S. LYVE-1, the lymphatic system and tumor lymphangiogenesis. *Trends Immunol* **22**, 317-321 (2001).
5. Fang L, Zhang M, Li Y, Liu Y, Cui Q, Wang N. PPARgene: A Database of Experimentally Verified and Computationally Predicted PPAR Target Genes. *PPAR Res* **2016**, 6042162 (2016).

REVIEWERS' COMMENTS

Reviewer #1 (Remarks to the Author):

N/A

Reviewer #2 (Remarks to the Author):

The authors have addressed all the reviewer's comments. The revised manuscript has been improved. No further comments.

Reviewer #3 (Remarks to the Author):

In this revised paper, the authors performed convincing additional experiments, addressing my comments. Despite well-designed experiments and cutting-edge methodology, the authors did not properly extract the novelty of their results from the current literature in the field. In particular, it appears that their results are mostly related to early stages of aortic valve disease. This needs to be stressed more in the discussion and highlighted in the abstract and title.

Minor comments:

- In Figure 1a, in C5BL/6J mice Western diet, it seems that the oil-red-o staining is strong in the valves, which are not highlighted. Please clarify.
- In figure 7d and 7e, the authors could check CD36 expression between calcified and non-calcified human samples. This could be instrumental in clarifying CD36 expression in early and late stages of valvular disease.

Reviewer #4 (Remarks to the Author):

Single-cell transcriptomics reveal cellular diversity in the aortic valve and the immunomodulatory role of PPAR γ during hyperlipidemia

The manuscript describes the heterogeneity of cells in the early aortic valve disease and emphasizes the role of PPAR γ as a key anti-inflammatory regulator in the pre-calcified stage of aortic valve disease. The authors have effectively clarified all the suggestions and concerns from Revision 1 providing a better background in the premise of the manuscript. Overall, the study is well designed and comprehensive, providing mechanistic insights of early aortic valve disease under hyperlipidemic conditions and scRNAseq and FACS, IHC complementary data has shown possible pathways to be targeted as therapeutic options. The methodology is appropriate and meets the expected standards in the field, providing enough details for the work to be reproduced. Largely, the manuscript has a reliable number of results backing the conclusions of the study. Nevertheless, other points should be addressed to strengthen the manuscript.

Minor comments that remain to be addressed:

1) Please add statistical analysis of Ldlr $^{-/-}$ vs. Apoe $^{-/-}$ in Fig. 1b to claim “LDL, but not total cholesterol, was higher in Ldlr $^{-/-}$ than in Apoe $^{-/-}$ mice”.

2) Please clarify the scientific reason of why the PCSK9 AAV injection experiment was only performed in Apoe $^{-/-}$ mice and, in the next figure ezetimibe lipid-lowering treatment was only performed in LDLr $^{-/-}$ but not in Apoe $^{-/-}$ mice. Consistently evaluating both LDLr $^{-/-}$ and Apoe $^{-/-}$ backgrounds in the manuscript will provide more strength to the conclusions of the study.

3) Supplementary data 1: please provide insights in the cellular pathways that suggest a phenotypic and functional heterogeneity in macrophages subclusters. Although the results are shown in Supplementary Data 1, the description of Sup. Data 1 is lacking in the results section.

4) Lastly, PPAR γ agonist + lipid lowering treatment combination could provide a more suitable treatment for early aortic valve disease? Can the authors speculate or test synergism between these two treatments in decreasing lipid accumulation and leukocyte infiltration or inflammation and aortic valve disease progression?

Overall, the authors have extensively discussed and addressed all valuable comments from the reviewers in the revised manuscript. The addition of siRNA-based PPAR γ knockdown in vitro experiment has provided a valuable mechanistic insight in human disease. The authors could discuss about the therapeutic perspectives of the modulation of PPAR γ pathway and associate with different vascular

diseases and disease stage. In the current state, the results are adequate and present a comprehensive characterization of valvular inflammation and hyperlipidemia. However, addressing minor comments would definitely enhance the strength of the current version.

NCOMMS-21-49026

2nd revision

Point-by-point response

Reviewer #1

N/A

- We really appreciate your positive review of our revised manuscript.

Reviewer #2

The authors have addressed all the reviewer's comments. The revised manuscript has been improved. No further comments.

- We really appreciate your positive review of our revised manuscript.

Reviewer #3

In this revised paper, the authors performed convincing additional experiments, addressing my comments. Despite well-designed experiments and cutting-edge methodology, the authors did not properly extract the novelty of their results from the current literature in the field. In particular, it appears that their results are mostly related to early stages of aortic valve disease. This needs to be stressed more in the discussion and highlighted in the abstract and title.

- Thank you so much for your valuable comments. As you pointed out, our study is mainly about the very early stage of aortic valve disease having local inflammation, and we agree that it is the crucial novelty of the study. Thus, we revised the statements of abstract [page 4, line 54-56 and 62-65] and discussion [page 22, line 442-447, and page 26, line 547-550] in the 2nd revised version of the manuscript, to better highlight the novelty of our study.

Minor comments:

• In Figure 1a, in C57BL/6J mice Western diet, it seems that the oil-red-o staining is strong in the valves, which are not highlighted. Please clarify.

- In C57BL/6J-WD of Fig. 1a, we can see black (or dark brown) pigments in the valve leaflet. These are the melanin pigments of aortic valves. There are plenty of melanin pigments in the mouse aortic valve, as shown in the left panels of Supplementary Fig. 1a-b. Surely, the valvular oil red o positive area of C57BL/6J mice was extremely low in both chow-fed and WD-fed groups. We added arrowheads in Fig. 1a indicating accumulated lipids (oil red o positive area) and the following statement in the figure legend to avoid confusion: Black or dark brown spots are melanin pigments.

Fig. 1a-d. Comparison of the lipid accumulation in the aortic valves and blood lipid profiles.

a-d. Comparison of lipid accumulation in the aortic valves and blood lipid profiles of mice (C57BL/6J, *Apoe*^{-/-}, and *Ldlr*^{-/-}) fed a chow diet (chow) versus a western diet (WD) for 10 weeks ($n = 5$). Representative lipid stain images and measurement of valvular lipid deposition. Black or dark brown spots are melanin pigments. Arrowhead: accumulated lipids. Scale bar: 150 μm (left), 30 μm (right). **(a)**, total cholesterol and LDL levels in the blood plasma. **(b)**, correlations between aortic valvular lipid deposition and blood lipid profiles of WD-fed hyperlipidemic mice. **(c)**, and correlation between aortic valvular lipid accumulation and aortic sinus lesions in WD-fed hyperlipidemic mice **(d)**.

Supplementary Fig. 1. Histological description of mouse aortic valve.

a. Gross and histological features of mouse aortic valve. Whole-mount brightfield images (left), and H&E images (right). Black or dark brown colors are melanin pigments. Scale bar: 100µm. **b.** Gross and histology of the aortic valve prepared for scRNA-seq and flow cytometric analysis. Aortic valves were carefully isolated from aortic sinus without contamination of other tissues. Whole-mount brightfield (left) and H&E images (right) of the isolated aortic valve. Black or dark brown spots are melanin pigments. Scale bar: 100 µm (left), 50 µm (right).

• **In figure 7d and 7e, the authors could check CD36 expression between calcified and non-calcified human samples. This could be instrumental in clarifying CD36 expression in early and late stages of valvular disease.**

- In our first revision, we showed the *CD36* expression between calcified and non-calcified human samples as Extra fig. 1 to respond to the comments of reviewer #1. In total valvular cells and VICs, it seemed that expression of *CD36* was relatively higher in calcified compared to non-calcified, but as shown in the feature plot, the overall expression level of *CD36* was extremely low (**Supplementary Fig. 14**). Now we presented these data (previous Extra fig. 1) as the new supplementary figure (**Supplementary Fig. 14**) [page 20, line 401-404].

Supplementary Fig. 14. Expression level of *CD36* in scRNA-seq of human aortic valves.

a. Feature (left) and violin (right) plot of *CD36* expression in each cluster of single-cells derived from human aortic valve. **b.** Comparison of *CD36* expression in calcified versus non-calcified groups for aortic valvular cells (top, left), VICs and stromal cells (top, right), macrophages (bottom, left), and VECs (bottom, right). Cells having no expression for *CD36* were excluded (n=14 and 35 cells for aortic valvular cells; 4 and 5 cells for VICs and stromal cells; 8 and 5 cells for macrophages; 2 and 24 cells for VECs in calcified and non-calcified groups). Each box depicts the IQR and median of each score, whiskers indicate 1.5 times the IQR. p, two-sided T-test p-value.

Reviewer #4

Single-cell transcriptomics reveal cellular diversity in the aortic valve and the immunomodulatory role of PPARG during hyperlipidemia

The manuscript describe the heterogeneity of cells in the early aortic valve disease and empathize the role of PPARG as a key anti-inflammatory regulator in the pre-calcified stage of aortic valve disease. The authors have effectively clarified all the suggestions and concerns from Revision 1 providing the a better background in premise of the manuscript. Overall, the study is well design and comprehensive, providing mechanistic insights of early aortic valve disease under hyperlipidemic condition and scRNAseq and FACS, IHC

complementary data has shown possible pathways to be targeted as therapeutic options. The methodology is appropriate and meets the expected standards in the field, providing enough details for the work to be reproduced. Largely, the manuscript has a reliable number of results backing the conclusions of the study. Nevertheless, other points should be addressed to strengthen the manuscript.

- Thank you so much for your helpful comments. We faithfully responded to all comments as much as possible, and hope our responses will suffice to answer the questions.

Minor comments that remain to be addressed:

1) Please add statistical analysis of *Ldlr*^{-/-} vs. *Apoe*^{-/-} in Fig. 1b to claim “LDL, but not total cholesterol, was higher in *Ldlr*^{-/-} than in *Apoe*^{-/-} mice”.

- We added the statistical analysis of *Ldlr*^{-/-} WD versus *Apoe*^{-/-} WD for total cholesterol and LDL level in Fig. 1b.

Fig. 1a-d. Comparison of the lipid accumulation in the aortic valves and blood lipid profiles.

a-d. Comparison of lipid accumulation in the aortic valves and blood lipid profiles of mice (*C57BL/6J*, *Apoe*^{-/-}, and *Ldlr*^{-/-}) fed a chow diet (chow) versus a western diet (WD) for 10 weeks (*n* = 5). Representative lipid stain images and measurement of valvular lipid deposition. Black or dark brown spots are melanin pigments. Arrowhead: accumulated lipids. Scale bar: 150 μm (left), 30 μm (right).

(a), total cholesterol and LDL levels in the blood plasma. (b), correlations between aortic valvular lipid deposition and blood lipid profiles of WD-fed hyperlipidemic mice. (c), and correlation between aortic valvular lipid accumulation and aortic sinus lesions in WD-fed hyperlipidemic mice (d).

2) Please clarify the scientific reason of why the PCSK9 \square AV injection experiment was only performed in \square poe^{-/-} mice and, in the next figure ezetimibe lipid-lowering treatment was only performed in LDLr^{-/-} but not in \square poe^{-/-} mice. Consistently evaluating both LDLr^{-/-} and \square poe^{-/-} backgrounds in the manuscript will provide more strength to the conclusions of the study.

- The molecular mechanism of PCSK9 is binding and preventing LDLR recycling [page 5-6, line 88-91]. Therefore, the PCSK-AAV injection cannot increase plasma LDL in *Ldlr*^{-/-} mice because they do not have LDLR. Also, the objective of the PCSK9-AAV injection experiment was to identify whether the increase in LDL level would aggravate the valvular lipid accumulation and inflammation. In this case, *Apoe*^{-/-} mice are the adequate model to utilize the PCSK-AAV injection because WD-fed *Apoe*^{-/-} mice showed low amounts of valvular lipid accumulation, due to the relatively low level of LDL, compared to their high total cholesterol level.

The objective of the experiment using ezetimibe was to confirm the alleviating effect in early aortic valvular disease by lipid-lowering treatment. In this case, to investigate the amount of change in valvular lipid accumulation and inflammation by ezetimibe treatment, *Ldlr*^{-/-} mice are more suitable than *Apoe*^{-/-} mice, because in **Fig. 1a**, WD-fed *Ldlr*^{-/-} mice showed a much higher level of valvular lipid accumulation and inflammation than WD-fed *Apoe*^{-/-} mice [page 12, line 215-218].

3) Supplementary data 1: please provide insights in the cellular pathways that suggest a phenotypic and functional heterogeneity in macrophages subclusters. Although the results are shown in Supplementary Data 1, the description of Sup. Data 1 is lacking in the results section.

- We described the gene expression and pathway analysis to explain phenotypic and functional heterogeneity of macrophage subclusters in Result section, using **Supplementary**

Data 1 [page 14-15, line 278-292].

4) Lastly, PPAR γ agonist + lipid lowering treatment combination could provide a more suitable treatment for early aortic valve disease? Can the authors speculate or test synergism between these two treatments in decreasing lipid accumulation and leukocyte infiltration or inflammation and aortic valve disease progression?

- In our experiments, lipid-lowering by ezetimibe showed alleviating lipid accumulation and inflammation. And PPAR γ activation by pioglitazone showed an anti-inflammatory effect independent of the blood total cholesterol and LDL levels. Accordingly, it seems that the co-administration of lipid-lowering drugs and PPAR γ agonist drugs would have a synergy effect on the early aortic valve disease. Previous studies suggested the synergistic beneficial effects of lipid-lowering drugs with pioglitazone in atherosclerosis¹, and cardiovascular disease². Therefore, it would be an important approach to investigate the effect of co-treatment of lipid-lowering drugs and PPAR γ agonists in the early-stage of aortic valve disease [page 25, line 519-525].

Overall, the authors have extensively discussed and addressed all valuable comments from the reviewers in the revised manuscript. The addition of siRNA-based PPAR γ knockdown in vitro experiment has provide a valuable mechanism insight in human disease. The authors could discuss about the therapeutic perspectives of the modulation of PPAR γ pathway and associate with different vascular diseases and disease stage. In the current state, the results are adequate and present a comprehensive characterization of valvular inflammation and hyperlipidemia. However, addressing minor comments would definitely enhance the strength of the current version.

- Thanks to the reviewer's valuable comments, we were able to strengthen the context of the study. Once again, we appreciate your dedication to reviewing our manuscript.

References

1. Choo EH, *et al.* Effect of Pioglitazone in Combination with Moderate Dose Statin on

Atherosclerotic Inflammation: Randomized Controlled Clinical Trial Using Serial FDG-PET/CT. *Korean Circ J* **48**, 591-601 (2018).

2. Hanefeld M, *et al.* Anti-inflammatory effects of pioglitazone and/or simvastatin in high cardiovascular risk patients with elevated high sensitivity C-reactive protein: the PIOSTAT Study. *J Am Coll Cardiol* **49**, 290-297 (2007).